# Trajectory Alignment: Understanding the Edge of Stability Phenomenon via Bifurcation Theory

**Minhak Song**
KAIST ISysE/Math
minhaksong@kaist.ac.kr

**Chulhee Yun**
KAIST AI
chulhee.yun@kaist.ac.kr

## Abstract

Cohen et al. (2021) empirically study the evolution of the largest eigenvalue of the loss Hessian, also known as sharpness, along the gradient descent (GD) trajectory and observe the *Edge of Stability* (EoS) phenomenon. The sharpness increases at the early phase of training (referred to as *progressive sharpening*), and eventually saturates close to the threshold of 2/(step size). In this paper, we start by demonstrating through empirical studies that when the EoS phenomenon occurs, different GD trajectories (after a proper reparameterization) align on a specific bifurcation diagram independent of initialization. We then rigorously prove this *trajectory alignment* phenomenon for a two-layer fully-connected linear network and a single-neuron nonlinear network trained with a single data point. Our trajectory alignment analysis establishes both progressive sharpening and EoS phenomena, encompassing and extending recent findings in the literature.

## 1 Introduction

It is widely believed that implicit bias or regularization of gradient-based methods plays a key role in generalization of deep learning (Vardi, 2022). There is a growing literature (Gunasekar et al., 2017, 2018; Soudry et al., 2018; Arora et al., 2018, 2019; Ji and Telgarsky, 2019a; Woodworth et al., 2020; Chizat and Bach, 2020; Yun et al., 2021) studying how the choice of optimization methods induces an *implicit bias* towards specific solutions among the many global minima in overparameterized settings.

Cohen et al. (2021) identify a surprising implicit bias of gradient descent (GD) towards global minima with a certain sharpness[1] value depending on the step size $\eta$. Specifically, for reasonable choices of $\eta$, (a) the sharpness of the loss at the GD iterate gradually increases throughout training until it reaches the stability threshold[2] of $2/\eta$ (known as *progressive sharpening*), and then (b) the sharpness saturates close to or above the threshold for the remainder of training (known as *Edge of Stability* (EoS)). These findings have sparked a surge of research aimed at developing a theoretical understanding of the progressive sharpening and EoS phenomena (Arora et al., 2022; Lyu et al., 2022; Wang et al., 2022; Ahn et al., 2023; Chen and Bruna, 2023; Zhu et al., 2023). In this paper, we study these phenomena through the lens of *bifurcation theory*, both empirically and theoretically.

**Motivating observations:** Figure 1 illustrates the GD trajectories with different initializations and fixed step sizes trained on three types of two-dimensional functions: (a) $\log(\cosh(xy))$, (b) $\frac{1}{2}(\tanh(x)y)^2$, and (c) $\frac{1}{2}(\text{ELU}(x)y)^2$, where $x$ and $y$ are scalars. The functions $\mathcal{L} : \mathbb{R}^2 \to \mathbb{R}$ have sharpness $y^2$ at the global minimum $(0, y)$ for all three models. These toy models can be viewed as examples of single-neuron models, where (a) represents a linear network with log-cosh loss, while (b) and (c) represent nonlinear networks with squared loss. These simple models can capture some interesting aspects of neural network training in the EoS regime, which are summarized below:

---

[1]Throughout this paper, the term "sharpness" means the maximum eigenvalue of the training loss Hessian.

[2]For quadratic loss, GD becomes unstable if the sharpness is larger than a threshold of 2/(step size).

37th Conference on Neural Information Processing Systems (NeurIPS 2023).

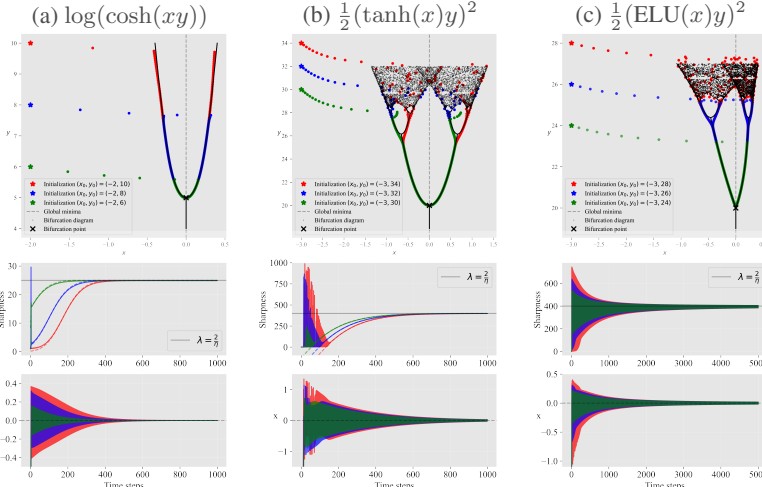

Figure 1: **GD trajectories align on the bifurcation diagram at the Edge of Stability.** We run GD on toy models with step size $\eta = 2/25$ in (a), and $\eta = 2/400$ in (b) and (c). Distinct colors indicate independent runs of GD with varying initializations. **Top row:** GD trajectories closely follow the bifurcation diagram of the map $x \mapsto x - \eta \frac{\partial}{\partial x} \mathcal{L}(x, y)$ and asymptotically reaches the bifurcation point $(0, \sqrt{2/\eta})$. **Bottom row:** the sharpness reaches $2/\eta$, and $x_t$ typically shows 2-period oscillating dynamics. The theoretical prediction $\tilde{\lambda}(q_t)$ (dashed lines, defined in Theorems 4.4 and C.4) in (a) and (b) approximates the sharpness along the GD trajectory and demonstrates progressive sharpening.

- EoS phenomenon: GD converges to a global minimum near the point $(0, \sqrt{2/\eta})$ with sharpness close to $2/\eta$. During the convergence phase, the training dynamics exhibit period-2 oscillations.

- For different initializations, GD trajectories for a given step size *align* on the same curve. For example, Figure 1a shows that GD trajectories with different initializations closely follow a specific U-shaped curve until convergence. We call this phenomenon *trajectory alignment*.

- In Figures 1b and 1c, GD trajectories are aligned on a curve with a fractal structure that qualitatively resembles the bifurcation diagram of a typical polynomial map, such as the logistic map. Particularly, Figure 1c demonstrates a period-halving phase transition in the GD dynamics, shifting from period-4 oscillation to period-2 oscillation.

- Surprisingly, the curve that GD trajectories approach and follow coincides with the *bifurcation diagram* of a one-dimensional map $x \mapsto x - \eta \frac{\partial}{\partial x} \mathcal{L}(x, y)$ with a fixed "control parameter" $y$. The stability of its fixed point $x = 0$ changes at the bifurcation point $(x, y) = (0, \sqrt{2/\eta})$, where period-doubling bifurcation occurs. Note that this point is a global minimum with sharpness $2/\eta$.

Interestingly, such striking behaviors can also be observed in more complex models, up to a proper reparameterization, as we outline in the next subsection.

## 1.1 Our contributions

In this paper, we discover and study the *trajectory alignment* behavior of (reparameterized) GD dynamics in the EoS regime. To our best knowledge, we are the first to identify such an alignment with a specific *bifurcation diagram* independent of initialization. Our empirical findings are rigorously proven for both two-layer fully-connected networks and single-neuron nonlinear networks. Our main contributions are summarized below:

- In Section 2, we introduce a novel *canonical reparameterization* of training parameters, which incorporates the data, network, and GD step size. This reparameterization allows us to study the trajectory alignment phenomenon in a unified framework. Through empirical study, Section 3 demonstrates that the alignment property of GD trajectories is not limited to toy models but also occurs in wide and deep networks trained on real-world dataset. Furthermore, we find that the alignment trend becomes more pronounced as the network width increases.

- In Section 4, we use our canonical reparameterization to establish the trajectory alignment phenomenon for a two-layer fully-connected linear network trained with a single data point. Our theoretical analysis rigorously proves both progressive sharpening and the EoS phenomenon, extending the work of Ahn et al. (2023) to a much broader class of networks and also providing more accurate bounds on the limiting sharpness.
- Our empirical and theoretical analyses up to Section 4 are applicable to convex Lipschitz losses, hence missing the popular squared loss. In Section 5, we take a step towards handling the squared loss. Employing an alternative reparameterization, we prove the same set of theorems as Section 4 for a single-neuron nonlinear network trained with a single data point under squared loss.

## 1.2 Related works

The *Edge of Stability* (EoS) phenomenon has been extensively studied in recent years, with many works seeking to provide a deeper understanding of the evolution of sharpness and the oscillating dynamics of GD. Jastrzębski et al. (2019) and Jastrzebski et al. (2020) observe that step size affects the sharpness along the optimization trajectory. Cohen et al. (2021) first formalize EoS through empirical study, and subsequent works have built on their findings. Ahn et al. (2022) analyze EoS through experiments and identify the relations between the behavior of loss, iterates, and sharpness. Ma et al. (2022) suggest that subquadratic growth of the loss landscape is the key factor of oscillating dynamics. Arora et al. (2022) show that (normalized) GD enters the EoS regime, by verifying the convergence to some limiting flow on the manifold of global minimizers. Wang et al. (2022) divide GD trajectory into four phases and explain progressive sharpening and EoS by using the norm of output layer weight as an indicator of sharpness. Lyu et al. (2022) prove that normalization layers encourage GD to reduce sharpness. Damian et al. (2023) use the third-order Taylor approximation of the loss to theoretically analyze EoS, assuming the existence of progressive sharpening. Lee and Jang (2023) propose a new sharpness measure using batch gradient distribution and characterize EoS for SGD. Concurrent to our work, Wu et al. (2023) study the logistic regression problem with separable dataset and establish that GD exhibits an implicit bias toward the max-margin solution in the EoS regime, extending prior findings in the small step size regime (Soudry et al., 2018; Ji and Telgarsky, 2019b).

Some recent works rigorously analyze the full GD dynamics for some toy cases and prove that the limiting sharpness is close to $2/\eta$. Zhu et al. (2023) study the loss $(x, y) \mapsto \frac{1}{4}(x^2 y^2 - 1)^2$ and prove that the sharpness converges close to $2/\eta$ with a local convergence guarantee. Notably, Ahn et al. (2023) study the function $(x, y) \mapsto \ell(xy)$ where $\ell$ is convex, even, and Lipschitz, and provide a global convergence guarantee. The authors prove that when $\ell$ is log-cosh loss or square root loss, the limiting sharpness in the EoS regime is between $2/\eta - \mathcal{O}(\eta)$ and $2/\eta$. Our theoretical results extend their results on a single-neuron linear network to a two-layer fully-connected linear network and provide an improved characterization on the limiting sharpness, tightening the gap between upper and lower bounds to only $\mathcal{O}(\eta^3)$.

The trajectory alignment phenomenon is closely related to Zhu et al. (2023) which shows empirical evidence of bifurcation-like oscillation in deep neural networks trained on real-world data. However, their empirical results do not show the alignment property of GD trajectory. In comparison, we observe that GD trajectories align on the same bifurcation diagram, independent of initialization.

Very recently, Kreisler et al. (2023) observe a similar trajectory alignment phenomenon for scalar linear networks, employing a reparameterization based on the sharpness of the gradient flow solution. However, their empirical findings on trajectory alignment are confined to scalar linear networks, and do not provide a theoretical explanation. In contrast, our work employs a novel canonical reparameterization and offers empirical evidence for the alignment phenomenon across a wide range of networks. Moreover, we provide theoretical proofs for two-layer linear networks and single-neuron nonlinear networks.

## 2 Preliminaries

**Notations.** For vectors $\boldsymbol{u}$ and $\boldsymbol{v}$, we denote the $\ell_p$ norm of $\boldsymbol{u}$ by $\|\boldsymbol{u}\|_p$, their tensor product as $\boldsymbol{u} \otimes \boldsymbol{v}$, and $\boldsymbol{u} \otimes \boldsymbol{u}$ by $\boldsymbol{u}^{\otimes 2}$. For a matrix $\boldsymbol{A}$, we denote the spectral norm by $\|\boldsymbol{A}\|_2$. Given a function $\mathcal{L}$ and a parameter $\boldsymbol{\Theta}$, we use $\lambda_{\max}(\boldsymbol{\Theta}) \coloneqq \lambda_{\max}(\nabla_{\boldsymbol{\Theta}}^2 \mathcal{L}(\boldsymbol{\Theta}))$ to denote the sharpness (i.e., the maximum eigenvalue of the loss Hessian) at $\boldsymbol{\Theta}$. We use asymptotic notations with subscripts (e.g., $\mathcal{O}_\ell(\cdot)$, $\mathcal{O}_{\delta,\ell}(\cdot)$) in order to hide constants that depend on the parameters or functions written as subscripts.

## 2.1 Problem settings

We study the optimization of neural network $f(\,\cdot\,;\boldsymbol{\Theta}) : \mathbb{R}^d \to \mathbb{R}$ parameterized by $\boldsymbol{\Theta}$. We focus on a simple over-parameterized setting trained on a single data point $\{(\boldsymbol{x}, y)\}$, where $\boldsymbol{x} \in \mathbb{R}^d$ and $y \in \mathbb{R}$. We consider the problem of minimizing the empirical risk

$$\mathcal{L}(\boldsymbol{\Theta}) = \ell(f(\boldsymbol{x};\boldsymbol{\Theta}) - y),$$

where $\ell$ is convex, even, and twice-differentiable with $\ell''(0) = 1$. We minimize $\mathcal{L}$ using GD with step size $\eta$: $\boldsymbol{\Theta}_{t+1} = \boldsymbol{\Theta}_t - \eta\nabla_{\boldsymbol{\Theta}}\mathcal{L}(\boldsymbol{\Theta}_t)$. The gradient and the Hessian of the function are given by

$$\nabla_{\boldsymbol{\Theta}}\mathcal{L}(\boldsymbol{\Theta}) = \ell'(f(\boldsymbol{x};\boldsymbol{\Theta}) - y)\nabla_{\boldsymbol{\Theta}}f(\boldsymbol{x};\boldsymbol{\Theta}),$$
$$\nabla_{\boldsymbol{\Theta}}^2\mathcal{L}(\boldsymbol{\Theta}) = \ell''(f(\boldsymbol{x};\boldsymbol{\Theta}) - y)(\nabla_{\boldsymbol{\Theta}}f(\boldsymbol{x};\boldsymbol{\Theta}))^{\otimes 2} + \ell'(f(\boldsymbol{x};\boldsymbol{\Theta}) - y)\nabla_{\boldsymbol{\Theta}}^2 f(\boldsymbol{x};\boldsymbol{\Theta}).$$

Suppose that $\boldsymbol{\Theta}^*$ be a global minimum of $\mathcal{L}$, i.e., $f(\boldsymbol{x};\boldsymbol{\Theta}^*) = y$. In this case, the loss Hessian and the sharpness at $\boldsymbol{\Theta}^*$ are simply characterized as

$$\nabla_{\boldsymbol{\Theta}}^2\mathcal{L}(\boldsymbol{\Theta}^*) = (\nabla_{\boldsymbol{\Theta}}f(\boldsymbol{x};\boldsymbol{\Theta}^*))^{\otimes 2}, \text{ and } \lambda_{\max}(\boldsymbol{\Theta}^*) = \|\nabla_{\boldsymbol{\Theta}}f(\boldsymbol{x};\boldsymbol{\Theta}^*)\|_2^2. \tag{1}$$

## 2.2 Canonical reparameterization

**Definition 2.1** (canonical reparameterization). *For given step size $\eta$, the canonical reparameterization of $\boldsymbol{\Theta}$ is defined as*

$$(p, q) := \left( f(\boldsymbol{x};\boldsymbol{\Theta}) - y, \frac{2}{\eta\|\nabla_{\boldsymbol{\Theta}}f(\boldsymbol{x};\boldsymbol{\Theta})\|_2^2} \right). \tag{2}$$

Under the canonical reparameterization, $p = 0$ represents global minima, and Eq. (1) implies that the point $(p, q) = (0, 1)$ is a global minimum with sharpness $2/\eta$. Note that $(p, q)$ alone does not, in general, uniquely determine the value of $\boldsymbol{\Theta}$. Rather, the motivation for this reparameterization technique is to effectively analyze the complex GD dynamics in the high-dimensional parameter space by reducing it to a 2-dimensional representation. The update of $p$ can be written as

$$\begin{aligned}
p_{t+1} &= f(\boldsymbol{x};\boldsymbol{\Theta}_{t+1}) - y = f\big(\boldsymbol{x};\boldsymbol{\Theta}_t - \eta\ell'(f(\boldsymbol{x};\boldsymbol{\Theta}_t) - y)\nabla_{\boldsymbol{\Theta}}f(\boldsymbol{x};\boldsymbol{\Theta}_t)\big) - y \\
&\approx f(\boldsymbol{x};\boldsymbol{\Theta}_t) - \nabla_{\boldsymbol{\Theta}}f(\boldsymbol{x};\boldsymbol{\Theta}_t)^\top \left(\eta\ell'(f(\boldsymbol{x};\boldsymbol{\Theta}_t) - y)\nabla_{\boldsymbol{\Theta}}f(\boldsymbol{x};\boldsymbol{\Theta}_t)\right) - y \\
&= (f(\boldsymbol{x};\boldsymbol{\Theta}_t) - y) - \eta\ell'(f(\boldsymbol{x};\boldsymbol{\Theta}_t) - y)\|\nabla_{\boldsymbol{\Theta}}f(\boldsymbol{x};\boldsymbol{\Theta})\|_2^2 \\
&= p_t - \frac{2\ell'(p_t)}{q_t},
\end{aligned} \tag{3}$$

which can be obtained by first-order Taylor approximation on $f$ for small step size $\eta$.[3]

## 2.3 Bifurcation analysis

Motivated from the approximated 1-step update rule given by Eq. (3), we conduct the bifurcation analysis on this one-dimensional map, considering $q_t$ as a control parameter. We first review some basic notions used in bifurcation theory (Strogatz, 1994).

**Definition 2.2** (stability of fixed point). *Let $z_0$ be a fixed point of a differentiable map $f : \mathbb{R} \to \mathbb{R}$, i.e., $f(z_0) = z_0$. We say $z_0$ is a* stable *fixed point of $f$ if $|f'(z)| < 1$, and we say $z_0$ is an* unstable *fixed point of $f$ if $|f'(z)| > 1$.*

**Definition 2.3** (stability of periodic orbit). *A point $z_0$ is called a period-$p$ point of a map $f : \mathbb{R} \to \mathbb{R}$ if $z_0$ is the fixed point of $f^p$ and $f^j(z_0) \neq z_0$ for any $1 \leq j \leq p - 1$. The orbit of $z_0$, given by $\{z_j = f^j(z_0) \mid j = 0, 1, \ldots, p - 1\}$ is called the period-$p$ orbit of $f$. A period-$p$ orbit is stable (unstable) if its elements are stable (unstable) fixed points of $f^p$, i.e., $\prod_{j=0}^{p-1}|f'(z_j)| < 1 \; (> 1)$.*

Now we analyze the bifurcation of the one-parameter family of mappings $f_q : \mathbb{R} \to \mathbb{R}$ given by

$$f_q(p) := p\left(1 - \frac{2r(p)}{q}\right), \tag{4}$$

where $q$ is a control parameter and $r$ is a differentiable function satisfying Assumption 2.4 below.

---

[3]The approximation is used just to motivate Lemma 2.1; in our theorems, we analyze the exact dynamics.

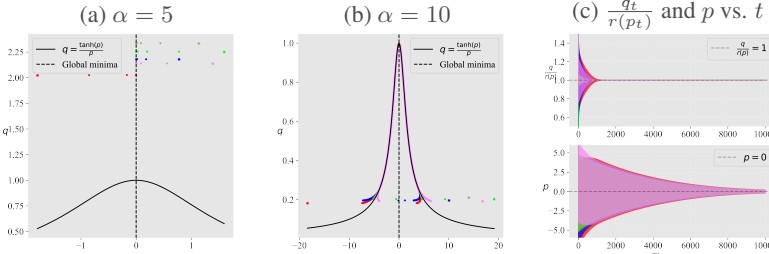

Figure 2: **(a), (b)** GD trajectories of two-layer fully-connected linear networks trained with **different initialization scale** $\alpha$. Each color corresponds to a single run of GD. Smaller initialization scale falls into the gradient flow regime, whereas larger initialization falls into the EoS regime. **(c)** In the EoS regime, $\frac{q_t}{r(p_t)}$ approaches 1 in the early phase of training, whereas $p_t$ converges to 0 relatively slowly.

**Assumption 2.4.** *A function* $r : \mathbb{R} \to \mathbb{R}$ *is even, continuously differentiable,* $r(0) = 1$, $r'(0) = 0$, $r'(p) < 0$ *for any* $p > 0$, *and* $\lim_{p \to \infty} r(p) = \lim_{p \to -\infty} r(p) = 0$. *In other words,* $r$ *is a smooth, symmetric bell-shaped function with the maximum value* $r(0) = 1$.

We note that Eq. (3) can be rewritten as $p_{t+1} = f_{q_t}(p_t)$ if we define $r$ by $r(p) := \frac{\ell'(p)}{p}$ for $p \neq 0$ and $r(0) := 1$. Below are examples of $\ell$ for which the corresponding $r$'s satisfy Assumption 2.4. These loss functions were previously studied by Ahn et al. (2023) to explain EoS for $(x, y) \mapsto \ell(xy)$.

- log-cosh loss: $\ell_{\text{log-cosh}}(p) := \log(\cosh(p))$. Note $\ell'_{\text{log-cosh}}(p) = \tanh(p)$.

- square-root loss: $\ell_{\text{sqrt}}(p) := \sqrt{1 + p^2}$. Note $\ell'_{\text{sqrt}}(p) = \frac{p}{\sqrt{1+p^2}}$.

If $r$ satisfies Assumption 2.4, then for any $0 < q \leq 1$, there exists a nonnegative number $p$ such that $r(p) = q$, and the solution is unique which we denote by $\hat{r}(q)$. In particular, $\hat{r} : (0, 1] \to \mathbb{R}_{\geq 0}$ is a function satisfying $r(\hat{r}(q)) = r(-\hat{r}(q)) = q$ for any $q \in (0, 1]$.

**Lemma 2.1** (period-doubling bifurcation of $f_q$). *Suppose that* $r$ *is a function satisfying Assumption 2.4. Let* $p^* = \sup\{p \geq 0 \mid \frac{xr'(x)}{r(x)} > -1 \text{ for any } |x| \leq p\}$ *and* $c = r(p^*)$. *If* $p^* = \infty$, *we choose* $c = 0$. *Then, the one-parameter family of mappings* $f_q : \mathbb{R} \to \mathbb{R}$ *given by Eq. (4) satisfies*

*(i) If* $q > 1$, $p = 0$ *is the stable fixed point.*

*(ii) If* $q \in (c, 1)$, $p = 0$ *is the unstable fixed point and* $\{\pm\hat{r}(q)\}$ *is the stable period-2 orbit.*

*Proof.* The map $f_q$ has the unique fixed point $p = 0$ for any $q > 0$. Since $|f'_q(0)| = |1 - \frac{2}{q}|$, $p = 0$ is a stable fixed point if $q > 1$ and $p = 0$ is an unstable fixed point if $0 < q < 1$. Now suppose that $q \in (c, 1)$. Then, we have $f_q(\hat{r}(q)) = -\hat{r}(q)$ and $f_q(-\hat{r}(q)) = \hat{r}(q)$, which implies that $\{\pm\hat{r}(q)\}$ is a period-2 orbit of $f_q$. Then, $|f'_q(\hat{r}(q))| = |f'_q(-\hat{r}(q))| = \left|1 + \frac{2\hat{r}(q)r'(\hat{r}(q))}{q}\right| < 1$ implies that $\{\pm\hat{r}(q)\}$ is a stable period-2 orbit. $\square$

According to Lemma 2.1, the stability of the fixed point $p = 0$ undergoes a change at $q = 1$, resulting in the emergence of a stable period-2 orbit. The point $(p, q) = (0, 1)$ is referred to as the *bifurcation point*, where a *period-doubling* bifurcation occurs. A *bifurcation diagram* illustrates the points asymptotically approached by a system as a function of a control parameter. In the case of the map $f_q$, the corresponding bifurcation diagram is represented by $p = 0$ for $q \geq 1$ and $p = \pm\hat{r}(q)$ (or equivalently, $q = r(p)$) for $q \in (c, 1)$.

It is worth noting that the period-2 orbit $\{\pm\hat{r}(p)\}$ becomes unstable for $q \in (0, c)$. If we choose $r$ to be $r(p) = \frac{\ell'(p)}{p}$ for $p \neq 0$ and $r(0) = 1$, then $1 + \frac{pr'(p)}{r(p)} = \frac{\ell''(p)}{r(p)} > 0$ for all $p$, assuming $\ell$ is convex. Consequently, for log-cosh loss and square root loss we have $c = 0$, indicating that the period-2 orbit of $f_q$ remains stable for all $q \in (0, 1)$. However, in Section 5, we will consider $r$ with $c > 0$, which may lead to additional bifurcations.

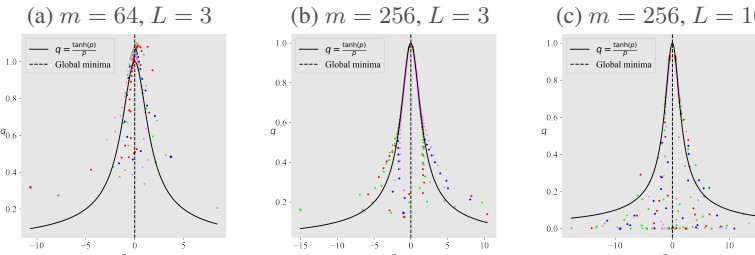

(a) $m = 64$, $L = 3$      (b) $m = 256$, $L = 3$      (c) $m = 256$, $L = 10$

Figure 3: GD trajectories of $\mathtt{tanh}$-activated neural networks with **varying width and depth**. Each color corresponds to a single run of GD. We observe that the wider network ($m = 256$) exhibits a stronger trajectory alignment phenomenon compared to the narrower network ($m = 64$). Figure 3c depicts the trajectories for a deeper network ($L = 10$), which also shows the trajectory alignment phenomenon.

## 3 Trajectory Alignment of GD: An Empirical Study

In this section, we conduct experimental studies on the trajectory alignment phenomenon in GD dynamics under the canonical reparameterization proposed in Section 2.

We consider a fully-connected $L$-layer neural network $f(\,\cdot\,; \boldsymbol{\Theta}) : \mathbb{R}^d \to \mathbb{R}$ written as

$$f(\boldsymbol{x}; \boldsymbol{\Theta}) = \boldsymbol{w}_L^\top \phi(\boldsymbol{W}_{L-1}\phi(\cdots\phi(\boldsymbol{W}_2\phi(\boldsymbol{W}_1\boldsymbol{x}))\cdots)),$$

where $\phi$ is an activation function, $\boldsymbol{W}_1 \in \mathbb{R}^{m \times d}$, $\boldsymbol{W}_l \in \mathbb{R}^{m \times m}$ for $2 \leq l \leq L - 1$, and $\boldsymbol{w}_L \in \mathbb{R}^m$. All $L$ layers have the same width of $m$. We minimize the empirical risk $\mathcal{L}(\boldsymbol{\Theta}) = \ell(f(\boldsymbol{x}; \boldsymbol{\Theta}) - y)$. We visualize GD trajectories under the canonical parameterization, where each plot shows five different randomly initialized weights using Xavier initialization multiplied with a rescaling factor of $\alpha$. For this analysis, we fix the training data point and hyperparameters as $\boldsymbol{x} = \boldsymbol{e}_1 = (1, 0, \ldots, 0)$, $y = 1$, $\eta = 0.01$, $d = 10$, and focus on the log-cosh loss for $\ell$, with either $\phi(t) = t$ (linear) or $\phi(t) = \tanh(t)$. We note that the trajectory alignment phenomenon is consistently observed in other settings, including square root loss, different activations (e.g., ELU), and various hyperparameters, in particular for sufficiently wide networks (additional experimental results are provided in Appendix A).

**The effect of initialization scale.** In Figures 2a and 2b, we examine the effect of the initialization scale $\alpha$ on GD trajectories in a two-layer fully-connected linear network with a width of $m = 256$. In Figure 2a, when the weights are initialized with a smaller scale ($\alpha = 5$), the initial value of $q$ is greater than 1, and it converges towards the minimum with only a small change in $q_t$ until convergence. In this case, the limiting sharpness is relatively smaller than $2/\eta$, and the EoS phenomenon does not occur. This case is referred to as the *gradient flow regime* (Ahn et al., 2023). On the other hand, in Figure 2b, when the weights are initialized with a larger scale ($\alpha = 10$), the initial value of $q$ is less than 1, and we observe convergence towards the point (close to) $(p, q) = (0, 1)$. This case is referred to as the *EoS regime*. We note that choosing larger-than-standard scale $\alpha$ is not a necessity for observing EoS; we note that even with $\alpha = 1$, we observe the EoS regime when $\eta$ is larger.

**Trajectory alignment on the bifurcation diagram.** In order to investigate the trajectory alignment phenomenon on the bifurcation diagram, we plot the bifurcation diagram $q = r(p) = \frac{\ell'(p)}{p}$ and observe that GD trajectories tend to align with this curve, which depends solely on $\ell$. Figure 2b clearly demonstrates this alignment phenomenon. Additionally, we analyze the evolution of $\frac{q_t}{r(p_t)}$ and $p_t$ in Figure 2c. We observe that the evolution of $\frac{q_t}{r(p_t)}$ follows two phases. In **Phase I**, $\frac{q_t}{r(p_t)}$ approaches to 1 quickly. In **Phase II**, the ratio remains close to 1. Notably, the convergence speed of $\frac{q_t}{r(p_t)}$ towards 1 is much faster than the convergence speed of $p_t$ towards 0. In Sections 4 and 5, we will provide a rigorous analysis of this behavior, focusing on the separation between Phase I and Phase II.

**The effect of width and depth.** In Figure 3, we present the GD trajectories of $\mathtt{tanh}$-activated networks with different widths and depths ($\alpha = 5$). All three cases belong to the EoS regime, where GD converges to a point close to $(p, q) = (0, 1)$, resulting in a limiting sharpness near $2/\eta$. However, when comparing Figures 3a and 3b, we observe that the trajectory alignment phenomenon is not

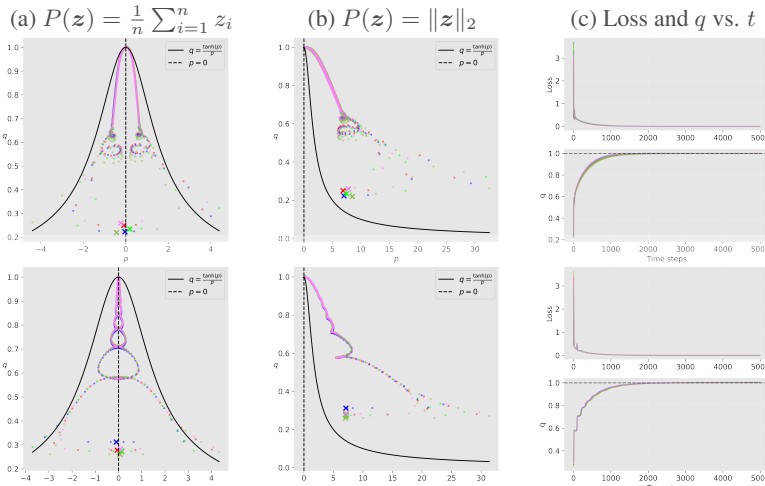

**Figure 4: Training on a subset of CIFAR-10.** The GD trajectories trained on a 50 example subset of CIFAR-10 under the **generalized canonical reparameterization** (5) with different choices of $P$ and network architecture. Each color corresponds to a single run of GD. **Top row:** fully-connected neural network with $\tanh$ activation. **Bottom row:** CNN with $\tanh$ activation. Full implementation details and further experimental results are given in Appendix A.4.

observed for the narrower network with $m = 64$, whereas the GD trajectories for the wider network with $m = 256$ are clearly aligned on the bifurcation diagram. This suggests that network width plays a role in the trajectory alignment phenomenon, which is reasonable since wide networks are well approximated by their linearized models, hence Eq. (3) is more accurate. Furthermore, we note that the trajectory alignment phenomenon is also observed for a deeper network with $L = 10$, as depicted in Figure 3c.

**Multiple training data points.** In our trajectory alignment analysis, we have primarily focused on training with a single data point. However, it is important to explore the extension of this phenomenon to scenarios with multiple training data points.

To investigate this, we train a neural network on a dataset $\{(\boldsymbol{x}_i, y_i)\}_{i=1}^n$, where $\boldsymbol{x}_i \in \mathbb{R}^d$ and $y_i \in \mathbb{R}$, by minimizing the empirical risk $\mathcal{L}(\boldsymbol{\Theta}) := \frac{1}{n} \sum_{i=1}^n \ell(f(\boldsymbol{x}_i; \boldsymbol{\Theta}) - y_i)$. Defining $\boldsymbol{X} \in \mathbb{R}^{n \times d}$ as the data matrix and $\boldsymbol{y} \in \mathbb{R}^n$ as the label vector, we introduce a *generalized canonical reparameterization*:

$$(p, q) := \left( P\left( f(\boldsymbol{X}; \boldsymbol{\Theta}) - \boldsymbol{y} \right), \frac{2n}{\eta \left\| \sum_{i=1}^n (\nabla_{\boldsymbol{\Theta}} f(\boldsymbol{x}_i; \boldsymbol{\Theta}))^{\otimes 2} \right\|_2} \right), \tag{5}$$

where $P : \mathbb{R}^n \to \mathbb{R}$ can be a function such as a mean value or a specific vector norm.

In Figure 4, we consider training on a 50 example subset of CIFAR-10 with only 2 classes and vary the network architecture. We use three-layer fully-connected network with $\tanh$ activation and convolutional network (CNN) with $\tanh$ activation. Under the generalized canonical reparameterization (5) for various choices of $P$, including the mean and the $\ell_2$ norm, we observe the trajectory alignment phenomenon throughout all settings, indicating a common alignment property of the GD trajectories. However, unlike the single data point case, the alignment does not happen on the curve $q = \frac{\ell'(p)}{p}$. The precise characterization of the coinciding curve is an interesting direction for future research.

## 4 Trajectory Alignment of GD: A Theoretical Study

In this section, we study a two-layer fully-connected linear network defined as $f(\boldsymbol{x}; \boldsymbol{\Theta}) := \boldsymbol{v}^\top \boldsymbol{U} \boldsymbol{x}$, where $\boldsymbol{U} \in \mathbb{R}^{d \times m}$, $\boldsymbol{v} \in \mathbb{R}^m$, and $\boldsymbol{\Theta}$ denote the collection of all parameters $(\boldsymbol{U}, \boldsymbol{v})$. We consider training this network with a single data point $\{(\boldsymbol{x}, 0)\}$, where $\boldsymbol{x} \in \mathbb{R}^d$ and $\|\boldsymbol{x}\|_2 = 1$. We run GD with step size $\eta$ on the empirical risk

$$\mathcal{L}(\boldsymbol{\Theta}) := \ell(f(\boldsymbol{x}; \boldsymbol{\Theta}) - 0) = \ell(\boldsymbol{v}^\top \boldsymbol{U} \boldsymbol{x}),$$

where $\ell$ is a loss function satisfying Assumption 4.1. We note that our assumptions on $\ell$ is motivated from the single-neuron linear network analysis ($d = m = 1$) by Ahn et al. (2023).

**Assumption 4.1.** *The loss $\ell$ is convex, even, 1-Lipschitz, and twice differentiable with $\ell''(0) = 1$.*

The canonical reparameterization (Definition 2.1) of $\boldsymbol{\Theta} = (\boldsymbol{U}, \boldsymbol{v})$ is given by

$$(p, q) := \left( \boldsymbol{v}^\top \boldsymbol{U} \boldsymbol{x}, \frac{2}{\eta(\|\boldsymbol{U}\boldsymbol{x}\|_2^2 + \|\boldsymbol{v}\|_2^2)} \right).$$

Under the canonical reparameterization, the 1-step update rule of GD can be written as

$$p_{t+1} = \left[ 1 - \frac{2r(p_t)}{q_t} + \eta^2 p_t^2 r(p_t)^2 \right] p_t, \quad q_{t+1} = \left[ 1 - \eta^2 p_t^2 r(p_t)(2q_t - r(p_t)) \right]^{-1} q_t, \quad (6)$$

where we define the function $r$ by $r(p) := \frac{\ell'(p)}{p}$ for $p \neq 0$ and $r(0) := 1$. Note that the sequence $(q_t)_{t=0}^\infty$ is monotonically increasing if $q_0 \geq \frac{1}{2}$, which is the case our analysis will focus on.

We have an additional assumption on $\ell$ as below, motivated from Lemma 2.1.

**Assumption 4.2.** *The function $r(p) = \frac{\ell'(p)}{p}$ corresponding to the loss $\ell$ satisfies Assumption 2.4.*

We now present our theoretical results on this setting, and defer the proofs to Appendix B.

## 4.1 Gradient flow regime

We first consider the gradient flow regime, where $q$ is initialized with $q_0 > 1$.

**Theorem 4.1** (gradient flow regime). *Let $\eta \in (0, \frac{2}{33})$ be a fixed step size and $\ell$ be a loss function satisfying Assumptions 4.1 and 4.2. Suppose that the initialization $(p_0, q_0)$ satisfies $|p_0| \leq 1$ and $q_0 \in \left( \frac{2}{2-\eta}, \min\left\{ \frac{1}{16\eta}, \frac{r(1)}{2\eta} \right\} \right)$. Consider the GD trajectory characterized in Eq. (6). Then, the GD iterations $(p_t, q_t)$ converge to the point $(0, q^*)$ such that*

$$q_0 \leq q^* \leq \exp(C\eta^2) q_0 \leq 2q_0,$$

*where $C = 8q_0 \left[ \min\left\{ \frac{2(q_0-1)}{q_0}, \frac{r(1)}{2q_0} \right\} \right]^{-1} > 0$.*

Theorem 4.1 implies that in gradient flow regime, GD with initialization $\boldsymbol{\Theta}_0 = (\boldsymbol{U}_0, \boldsymbol{v}_0)$ and step size $\eta$ converges to $\boldsymbol{\Theta}^*$ which has the sharpness bounded by:

$$(1 - C\eta^2)(\|\boldsymbol{U}_0\boldsymbol{x}\|_2^2 + \|\boldsymbol{v}_0\|_2^2) \leq \lambda_{\max}(\boldsymbol{\Theta}^*) \leq (\|\boldsymbol{U}_0\boldsymbol{x}\|_2^2 + \|\boldsymbol{v}_0\|_2^2).$$

Hence, for small step size $\eta$, if the initialization satisfies $\|\boldsymbol{U}_0\boldsymbol{x}\|_2^2 + \|\boldsymbol{v}_0\|_2^2 < \frac{2}{\eta} - 1$, then the limiting sharpness is slightly below $\|\boldsymbol{U}_0\boldsymbol{x}\|_2^2 + \|\boldsymbol{v}_0\|_2^2$. Note that we assumed the bound $|p_0| \leq 1$ for simplicity, but our proof also works with the assumption $|p_0| \leq K$ for any positive constant $K$ modulo some changes in numerical constants. Moreover, our assumption on the upper bound of $q_0$ is $1/\eta$ up to a constant factor, which covers most realistic choices of initialization.

## 4.2 EoS regime

We now provide rigorous results in the EoS regime, where the GD trajectory aligns on the bifurcation diagram $q = r(p)$. To establish these results, we introduce additional assumptions on the loss $\ell$.

**Assumption 4.3.** *The function $r(z) = \frac{\ell'(z)}{z}$ is $C^4$ on $\mathbb{R}$ and satisfies*

(i) $z \mapsto \frac{r'(z)}{r(z)^2}$ *is decreasing on $\mathbb{R}$,*

(ii) $z \mapsto \frac{zr'(z)}{r(z)}$ *is decreasing on $z > 0$ and increasing on $z < 0$,*

(iii) $z \mapsto \frac{zr(z)}{r'(z)}$ *is decreasing on $z > 0$ and increasing on $z < 0$.*

We note that both the log-cosh loss and the square root loss satisfy Assumptions 4.1, 4.2, and 4.3.

**Theorem 4.2** (EoS regime, Phase I). *Let $\eta$ be a small enough step size and $\ell$ be a loss function satisfying Assumptions 4.1, 4.2, and 4.3. Let $z_0 := \sup_z \{\frac{zr'(z)}{r(z)} \geq -\frac{1}{2}\}$ and $c_0 := \max\{r(z_0), \frac{1}{2}\}$. Let $\delta \in (0, 1 - c_0)$ be any given constant. Suppose that the initialization $(p_0, q_0)$ satisfies $|p_0| \leq 1$ and $q_0 \in (c_0, 1 - \delta)$. Consider the reparameterized GD trajectory characterized in Eq. (6). We assume that for all $t \geq 0$ such that $q_t < 1$, we have $p_t \neq 0$. Then, there exists a time step $t_a = \mathcal{O}_{\delta,\ell}(\log(\eta^{-1}))$, such that for any $t \geq t_a$,*

$$\frac{q_t}{r(p_t)} = 1 + h(p_t)\eta^2 + \mathcal{O}_{\delta,\ell}(\eta^4),$$

*where $h(p) := -\frac{1}{2}\left(\frac{pr(p)^3}{r'(p)} + p^2 r(p)^2\right)$ for $p \neq 0$ and $h(p) := -\frac{1}{2r''(0)}$ for $p = 0$.*

One can check that for log-cosh and square-root losses, the ranges of $h$ are $(0, 3/4]$ and $(0, 1/2]$, respectively. Theorem 4.2 implies that in the early phase of training ($t \leq t_a = \mathcal{O}(\log(\eta^{-1}))$), GD iterates $(p_t, q_t)$ approach closely to the bifurcation diagram $r(p) = q$, which we called Phase I in Section 3. In Phase II, GD trajectory aligns on this curve in the remaining of the training ($t \geq t_a$). Theorem 4.3 provides an analysis on Phase II stated as below.

**Theorem 4.3** (EoS regime, Phase II). *Under the same settings as in Theorem 4.2, there exists a time step $t_b = \Omega((1 - q_0)\eta^{-2})$ such that $q_{t_b} \leq 1$ and $q_t > 1$ for any $t > t_b$. Moreover, the GD iterates $(p_t, q_t)$ converge to the point $(0, q^*)$ such that*

$$q^* = 1 - \frac{\eta^2}{2r''(0)} + \mathcal{O}_{\delta,\ell}(\eta^4).$$

Theorem 4.3 implies that in EoS regime, GD with step size $\eta$ converges to $\mathbf{\Theta}^*$ with sharpness

$$\lambda_{\max}(\mathbf{\Theta}^*) = \frac{2}{\eta} - \frac{\eta}{|r''(0)|} + \mathcal{O}_{\delta,\ell}(\eta^3).$$

Note that Ahn et al. (2023) study the special case $d = m = 1$ and prove that the limiting sharpness is between $2/\eta - \mathcal{O}(\eta)$ and $2/\eta$. Theorem 4.3 provides tighter analysis on the limiting sharpness in more general settings, reducing the gap between the upper bound and lower bound to only $\mathcal{O}(\eta^3)$. Also, our result is the first to prove that the limiting sharpness in the EoS regime is bounded away from $2/\eta$ by a nontrivial margin.

We also study the evolution of sharpness along the GD trajectory and prove that progressive sharpening (i.e., sharpness increases) occurs during Phase II.

**Theorem 4.4** (progressive sharpening). *Under the same setting as in Theorem 4.2, let $t_a$ denote the obtained iteration. Define the function $\tilde{\lambda} : \mathbb{R}_{>0} \to \mathbb{R}$ given by*

$$\tilde{\lambda}(q) := \begin{cases} \left(1 + \frac{\hat{r}(q)r'(\hat{r}(q))}{q}\right)\frac{2}{\eta} & \text{if } q \leq 1, \text{ and} \\ \frac{2}{\eta} & \text{otherwise.} \end{cases}$$

*Then, the sequence $\left(\tilde{\lambda}(q_t)\right)_{t=0}^{\infty}$ is monotonically increasing. Moreover, for any $t \geq t_a$, the sharpness at GD iterate $\mathbf{\Theta}_t$ closely follows the sequence $\left(\tilde{\lambda}(q_t)\right)_{t=0}^{\infty}$ by satisfying*

$$\left|\lambda_{\max}(\mathbf{\Theta}_t) - \tilde{\lambda}(q_t)\right| \leq 1 + \mathcal{O}_\ell(\eta).$$

The gap between $\lambda_{\max}(\mathbf{\Theta}_t)$ and $\tilde{\lambda}(q_t)$ is bounded by a numerical constant, which becomes negligible compared to $2/\eta$ for small $\eta$. In Figure 1a, we perform numerical experiments on a single-neuron case and observe that $\tilde{\lambda}(q_t)$ closely approximates the sharpness.

Note that Cohen et al. (2021) observe an increase in sharpness during training in the gradient flow regime, while our work reveals that sharpness increases when exhibiting oscillations in the EoS regime. This distinction may be linked to the selection of the loss function, as our study focuses on Lipschitz convex losses, while Cohen et al. (2021) examine the squared loss.

## 5 EoS in Squared Loss: Single-neuron Nonlinear Network

Our canonical reparameterization has a limitation in explaining the EoS phenomenon under squared loss $\ell(p) = \frac{1}{2}p^2$, as the function $r(p) = \frac{\ell'(p)}{p} = 1$ does not satisfy Assumption 2.4. However, empirical studies by Cohen et al. (2021) have observed the EoS phenomenon in GD training with squared loss. In this section, we analyze a simple toy model to gain insight into the EoS phenomenon and trajectory alignment of GD under squared loss.

We study the GD dynamics on a two-dimensional function $\mathcal{L}(x, y) := \frac{1}{2}(\phi(x)y)^2$, where $x$, $y$ are scalars and $\phi$ is a nonlinear activation satisfying Assumption 5.1 below.

**Assumption 5.1** (sigmoidal activation). *The activation function $\phi : \mathbb{R} \to \mathbb{R}$ is odd, increasing, 1-Lipschitz and twice continuously differentiable. Moreover, $\phi(0) = 0$, $\phi'(0) = 1$, $\lim_{x \to \infty} \phi(x) = 1$, and $\lim_{x \to -\infty} \phi(x) = -1$.*

One good example of $\phi$ satisfying Assumption 5.1 is $\tanh$. For this section, we use an alternative reparameterization defined as below.

**Definition 5.2.** *For given step size $\eta$, the $(p, q)$ reparameterization of $(x, y) \in \mathbb{R}^2$ is defined as*

$$(p, q) := \left( x, \frac{2}{\eta y^2} \right).$$

Under the reparameterization, the 1-step update rule can be written as

$$p_{t+1} = \left( 1 - \frac{2r(p_t)}{q_t} \right) p_t, \quad q_{t+1} = (1 - \eta\phi(p_t)^2)^{-2}q_t, \tag{7}$$

where the function $r$ is given by $r(z) := \frac{\phi(z)\phi'(z)}{z}$ for $z \neq 0$ and $r(0) := 1$.

We can observe a notable resemblance between Eq. (7) and Eq. (6). Indeed, our theoretical findings for a single-neuron nonlinear network closely mirror those of the two-layer linear network discussed in Section 4. Due to lack of space, we summarize our theorems in this setting as the following:

**Theorem 5.1** (informal). *Under suitable assumptions on $\phi$, step size, and initialization, GD trained on the squared loss $\mathcal{L}(x, y) := \frac{1}{2}(\phi(x)y)^2$ exhibits the same gradient flow, EoS (Phase I, II), and progressive sharpening phenomena as shown in Section 4.*

In Theorem C.3, we prove that in the EoS regime, the limiting sharpness is $\frac{2}{\eta} - \frac{2}{|r''(0)|} + \mathcal{O}(\eta)$. For formal statements of the theorems and the proofs, we refer the reader to Appendix C.

## 6 Conclusion

In this paper, we provide empirical evidence and rigorous analysis to demonstrate the interesting phenomenon of GD trajectory alignment in the EoS regime. Importantly, we show that different GD trajectories, under the canonical reparameterization, align on a bifurcation diagram independent of initialization. This discovery is notable due to the intricate and non-convex nature of neural network optimization, where the algorithm trajectory is heavily influenced by initialization choices. Our theoretical analysis not only characterizes the behavior of limiting sharpness but also establishes progressive sharpening of GD. One immediate future direction is to understand the trajectory alignment behavior when trained on multiple data points. Lastly, it will be interesting to extend our analysis to encompass squared loss for general neural network, going beyond the toy single-neuron example.

## Acknowledgments

This paper was supported by Institute of Information & communications Technology Planning & Evaluation (IITP) grant (No. 2019-0-00075, Artificial Intelligence Graduate School Program (KAIST)) funded by the Korea government (MSIT), two National Research Foundation of Korea (NRF) grants (No. NRF-2019R1A5A1028324, RS-2023-00211352) funded by the Korea government (MSIT), and a grant funded by Samsung Electronics Co., Ltd.

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

# Contents

# A    Additional Experiments

In this section, we present additional empirical evidence demonstrating the phenomenon of trajectory alignment, which supports the findings discussed in Section 3 of our main paper.

## A.1    Experimental setup

**Objective function.**    We run gradient descent (GD) to minimize the objective function defined as

$$\mathcal{L}(\boldsymbol{\Theta}) = \ell(f(\boldsymbol{x}; \boldsymbol{\Theta}) - y),$$

where $\boldsymbol{\Theta}$ represents the parameters, $\ell : \mathbb{R} \to \mathbb{R}$ is a loss function, $f : \mathbb{R}^d \to \mathbb{R}$ is a neural network, and $\{(\boldsymbol{x}, y)\}$ denotes a single data point with $\boldsymbol{x} \in \mathbb{R}^d$ and $y \in \mathbb{R}$. We also consider training on multiple data points $\{(\boldsymbol{x}_i, y_i)\}_{i=1}^n$ with $\boldsymbol{x}_i \in \mathbb{R}^d$ and $y_i \in \mathbb{R}$ for each $1 \leq i \leq n$, where we minimize the objective function

$$\mathcal{L}(\boldsymbol{\Theta}) \coloneqq \frac{1}{n} \sum_{i=1}^n \ell(f(\boldsymbol{x}_i; \boldsymbol{\Theta}) - y_i).$$

In our experiments, we primarily focus on the log-cosh loss function $\ell_{\text{log-cosh}}(p) = \log(\cosh(p))$, but we also investigate the square root loss $\ell_{\text{sqrt}}(p) = \sqrt{1 + p^2}$.

**Model architecture.**    We train a fully-connected $L$-layer neural network, denoted as $f(\,\cdot\,; \boldsymbol{\Theta}) : \mathbb{R}^d \to \mathbb{R}$. The network is defined as follows:

$$f(\boldsymbol{x}; \boldsymbol{\Theta}) = \boldsymbol{w}_L^\top \phi(\boldsymbol{W}_{L-1} \phi(\cdots \phi(\boldsymbol{W}_2 \phi(\boldsymbol{W}_1 \boldsymbol{x})) \cdots)),$$

where $\phi : \mathbb{R} \to \mathbb{R}$ is an activation function applied entry-wise, $\boldsymbol{W}_1 \in \mathbb{R}^{m \times d}$, $\boldsymbol{W}_l \in \mathbb{R}^{m \times m}$ for $2 \leq l \leq L - 1$, and $\boldsymbol{w}_L \in \mathbb{R}^m$. All $L$ layers have the same width of $m$, and the biases of all layers are fixed to 0.[4] We consider three activations: hyperbolic tangent $\phi(t) = \tanh(t)$, exponential linear unit $\phi(t) = \text{ELU}(t)$, and linear $\phi(t) = t$.

**Weight initialization.**    We perform gradient descent (GD) using five different randomly initialized sets of weights. The weights are initialized using Xavier initialization, and each layer is multiplied by a rescaling factor (gain) of $\alpha$. In the plots presented throughout this section, we mark the initialization points with an 'x' to distinguish them from other points on the trajectories.

**Canonical reparameterization.**    We plot GD trajectories after applying the canonical reparameterization introduced in Definition 2.1:

$$(p, q) \coloneqq \left( f(\boldsymbol{x}; \boldsymbol{\Theta}) - y, \ \frac{2}{\eta \|\nabla_{\boldsymbol{\Theta}} f(\boldsymbol{x}; \boldsymbol{\Theta})\|_2^2} \right),$$

where $\eta$ denotes the step size. For training on multiple data points, we employ the generalized canonical reparameterization as defined in Eq. (5):

$$(p, q) \coloneqq \left( P\left(f(\boldsymbol{X}; \boldsymbol{\Theta}) - \boldsymbol{y}\right), \ \frac{2n}{\eta \left\| \sum_{i=1}^n (\nabla_{\boldsymbol{\Theta}} f(\boldsymbol{x}_i; \boldsymbol{\Theta}))^{\otimes 2} \right\|_2} \right),$$

where $P : \mathbb{R}^n \to \mathbb{R}$ can represent the mean value or vector norms. Specifically, we mainly focus on the mean value $P(\boldsymbol{z}) = \frac{1}{n} \sum_{i=1}^n z_i$, but we also examine vector norms such as $P(\boldsymbol{z}) = \|\boldsymbol{z}\|_1$, $P(\boldsymbol{z}) = \|\boldsymbol{z}\|_2$, and $P(\boldsymbol{z}) = \|\boldsymbol{z}\|_\infty$, where $\boldsymbol{z} = (z_1, z_2, \ldots, z_n) \in \mathbb{R}^n$.

For large networks, explicitly calculating the $\ell_2$ matrix norm $\left\| \sum_{i=1}^n (\nabla_{\boldsymbol{\Theta}} f(\boldsymbol{x}_i; \boldsymbol{\Theta}))^{\otimes 2} \right\|_2$ is infeasible. Therefore, we adopt a fast and efficient matrix-free method based on power iteration, as proposed by Yao et al. (2020). This method allows us to numerically compute the $\ell_2$ norm of large-scale symmetric matrices.

---

[4]While we fix the biases of all layers to 0 to maintain consistency with our theory, the trajectory alignment phenomenon is consistently observed even for neural networks with bias. In Appendix A.4, we consider training networks with bias.

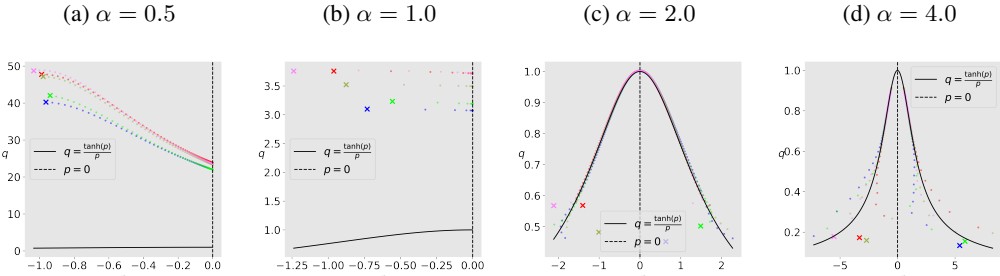

Figure 5: GD trajectories (1000 iterations) of 3-layer fully connected neural networks with $\tanh$ activation and a width of $m = 256$. The plots depict the trajectories for **different initialization scales**: $\alpha = 0.5, 1.0, 2.0, 4.0$. Smaller initialization scales ($\alpha = 0.5, 1.0$) lead to trajectories in the gradient flow regime, while larger initialization scales ($\alpha = 2.0, 4.0$) result in trajectories in the EoS regime.

## A.2 Training on a single data point

**Training data.** We conduct experiments on a synthetic single data point $(\boldsymbol{x}, y)$, where $\boldsymbol{x} = \boldsymbol{e}_1 = (1, 0, \ldots, 0) \in \mathbb{R}^d$ and $y \in \mathbb{R}$. Throughout the experiments in this subsection, we keep the data dimension fixed at $d = 10$, the data label at $y = 1$, and the step size at $\eta = 0.01$.

**The effect of initialization scale.** We investigate the impact of the initialization scale $\alpha$ while keeping other hyperparameters fixed. We vary the initialization scale across $\{0.5, 1.0, 2.0, 4.0\}$. Specifically, in Figure 5, we train a 3-layer fully connected neural network with $\tanh$ activation. We observe that smaller initialization scales ($\alpha = 0.5, 1.0$) lead to trajectories in the gradient flow regime, while larger initialization scales ($\alpha = 2.0, 4.0$) result in trajectories in the EoS regime. This behavior is primarily due to the initial value of $q$ being smaller than $1$ for larger initialization scales, which causes the trajectory to fall into the EoS regime.

**The effect of network width.** We investigate the impact of network width on the trajectory alignment phenomenon. While keeping other hyperparameters fixed, we control the width $m$ with values of $\{64, 128, 256, 512\}$. In Figure 6, we train 3-layer fully connected neural networks with $\tanh$ activation and an initialization scale of $\alpha = 4$. Additionally, in Figure 7, we examine the same setting but with different depth, training 10-layer fully connected neural networks.

It is commonly observed that the alignment trend becomes more pronounced as the network width increases. In Figures 6 and 7, all trajectories are in the EoS regime. However, narrower networks ($m = 64, 128$) do not exhibit the trajectory alignment phenomenon, while wider networks ($m = 256, 512$) clearly demonstrate this behavior. These results indicate that network width plays a significant role in the trajectory alignment property of GD.

**The effect of loss and activation functions.** The trajectory alignment phenomenon is consistently observed in various settings, including those with square root loss and different activation functions. In Figure 8, we investigate a 3-layer fully connected neural network with a width of $m = 256$ and an initialization scale of $\alpha = 2$. We explore different activation functions, including $\tanh$, ELU, and linear, and consider both $\log$-$\cosh$ loss and square-root loss. Across all these settings, we observe the trajectory alignment phenomenon, where the GD trajectories align on a curve $q = \dfrac{\ell'(p)}{p}$.

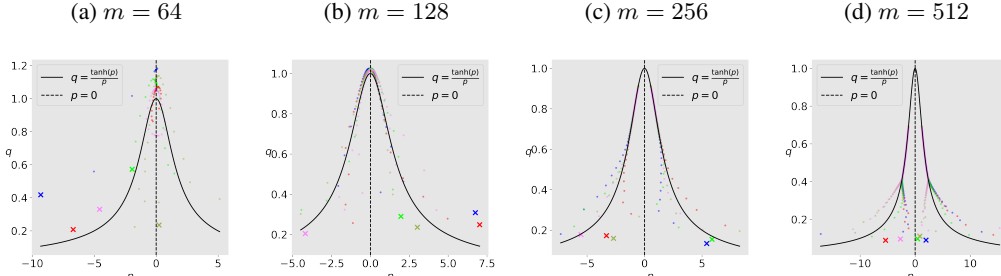

(a) $m = 64$  (b) $m = 128$  (c) $m = 256$  (d) $m = 512$

Figure 6: GD trajectories (1000 iterations) of 3-layer fully connected neural networks with $\tanh$ activation and an initialization scale of $\alpha = 4$. The plots depict the trajectories for **different network widths**: $m = 64, 128, 256, 512$. Narrower networks ($m = 64, 128$) do not exhibit trajectory alignment, while wider networks ($m = 256, 512$) clearly demonstrate the trajectory alignment phenomenon.

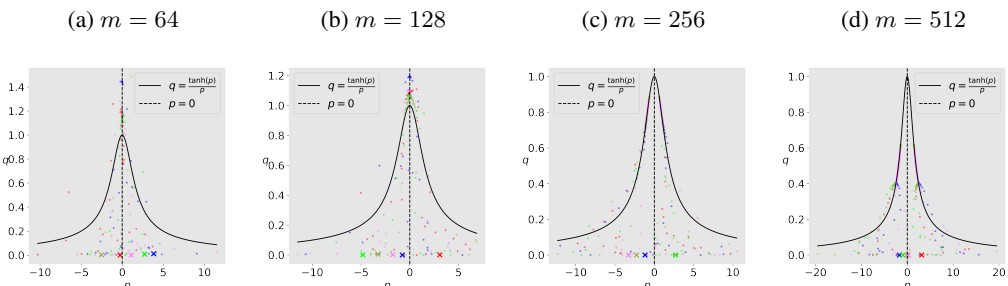

(a) $m = 64$  (b) $m = 128$  (c) $m = 256$  (d) $m = 512$

Figure 7: GD trajectories (1000 iterations) of 10-layer fully connected neural networks with $\tanh$ activation and an initialization scale of $\alpha = 4$. The plots depict the trajectories for **different network widths**: $m = 64, 128, 256, 512$. Similar to Figure 6, narrower networks ($m = 64, 128$) do not exhibit trajectory alignment, while wider networks ($m = 256, 512$) clearly demonstrate the trajectory alignment phenomenon.

### A.3 Training on multiple synthetic data Points

**Training data.** We consider training on a synthetic dataset consisting of $n$ data points, denoted as $(\boldsymbol{x}_i, y_i)_{i=1}^{n}$. The input vectors $\boldsymbol{x}_i$ are sampled from a standard Gaussian distribution $\mathcal{N}(\boldsymbol{0}, \boldsymbol{I})$, where $\boldsymbol{x}_i \in \mathbb{R}^d$, and the corresponding target values $y_i$ are sampled from a Gaussian distribution $\mathcal{N}(0, 1)$, where $y_i \in \mathbb{R}$. Throughout our experiments in this subsection, we use a fixed data dimension of $d = 10$ and a step size of $\eta = 0.01$.

**The effect of function $P$.** To investigate the impact of different choices of the function $P$, we train a 3-layer fully connected neural network with $\tanh$ activation. The network has a width of $m = 256$ and is initialized with a scale of $\alpha = 4$. The training is performed on a dataset consisting of $n = 10$ data points. Figure 9 displays the trajectories of GD trajectories under the generalized canonical reparameterization defined in Eq. (5) for various choices of the function $P$. These choices include the mean, $\ell_1$ norm, $\ell_2$ norm, and $\ell_\infty$ norm.

We observe that GD trajectories exhibit alignment behavior across different choices of the function $P$. Notably, when $P$ is selected as the mean, the trajectories align on the curve $q = \frac{\ell'(p)}{p}$. However, when $P$ is based on vector norms, the alignment occurs on different curves. The precise characterization of these curves remains as an interesting open question for further exploration.

**The effect of the number of data points.** We examine how the size of the training dataset, denoted by $n$, influences the trajectory alignment behavior of GD. While keeping other hyperparameters constant, we vary $n$ with values $\{2, 4, 8, 16, 32, 64, 128, 512, 1024\}$. In Figure 10, we train a 3-layer fully connected neural network with $\tanh$ activation, a width of $m = 256$, and an initialization scale

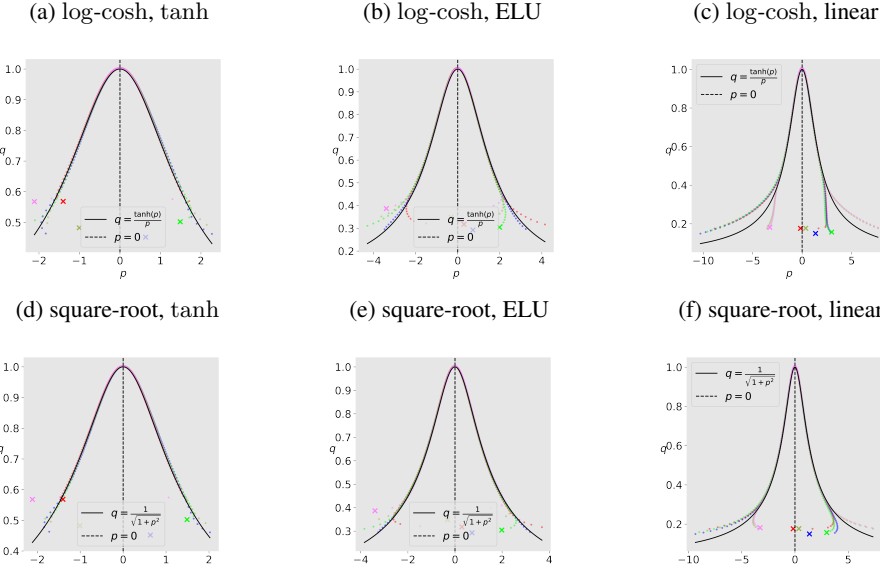

Figure 8: GD trajectories (1000 iterations) of 3-layer fully connected neural networks with **different loss and activation** functions. The networks have a width of $m = 256$ and an initialization scale of $\alpha = 2$. The plots illustrate the trajectories for various combinations of activation functions (tanh, ELU, and linear) and loss functions (log-cosh and square-root loss). In all these settings, the trajectory alignment phenomenon is observed, where GD trajectories align on a curve $q = \frac{\ell'(p)}{p}$. **Top row**: trajectories with log-cosh loss. From left to right: tanh, ELU, and linear activations. **Bottom row**: trajectories with square-root loss. From left to right: tanh, ELU, and linear activations.

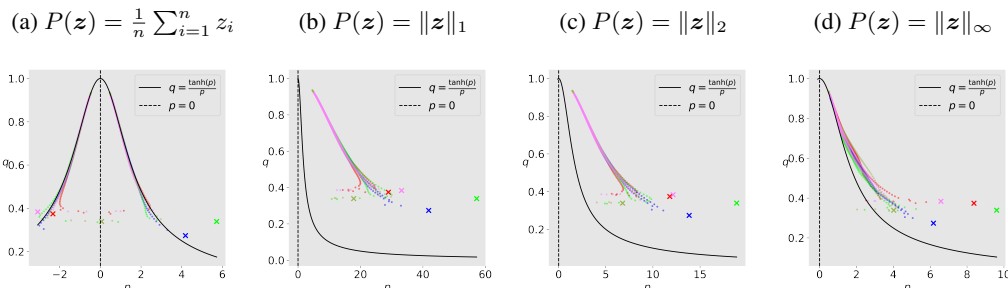

Figure 9: GD trajectories (500 iterations) under the generalized canonical reparameterization (Eq. (5)) for **different choices of the function** $P$. The plots depict the training of a 3-layer fully connected neural network with tanh activation, a width of $m = 256$, and an initialization scale of $\alpha = 4$. Each plot corresponds to a different parameterization, including the mean, $\ell_1$ norm, $\ell_2$ norm, and $\ell_\infty$ norm. GD trajectories commonly exhibit alignment behavior, with the mean parameterization aligning on the curve $q = \frac{\ell'(p)}{p}$.

of $\alpha = 4$. Additionally, in Figure 11, we investigate the same setting but with a different activation function, training ELU-activated fully connected neural networks. The GD trajectories are plotted under the generalized canonical reparameterization using the mean function $P(z) = \frac{1}{n} \sum_{i=1}^{n} z_i$.

We observe a consistent trajectory alignment phenomenon across different choices of the number of data points. Interestingly, for small values of $n$, the trajectories clearly align on the curve $q = \frac{\ell'(p)}{p}$. However, as the number of data points $n$ increases, it seems that the trajectories no longer align on this curve but different "narrower" curves. Understanding the underlying reasons for this phenomenon poses an intriguing open question.

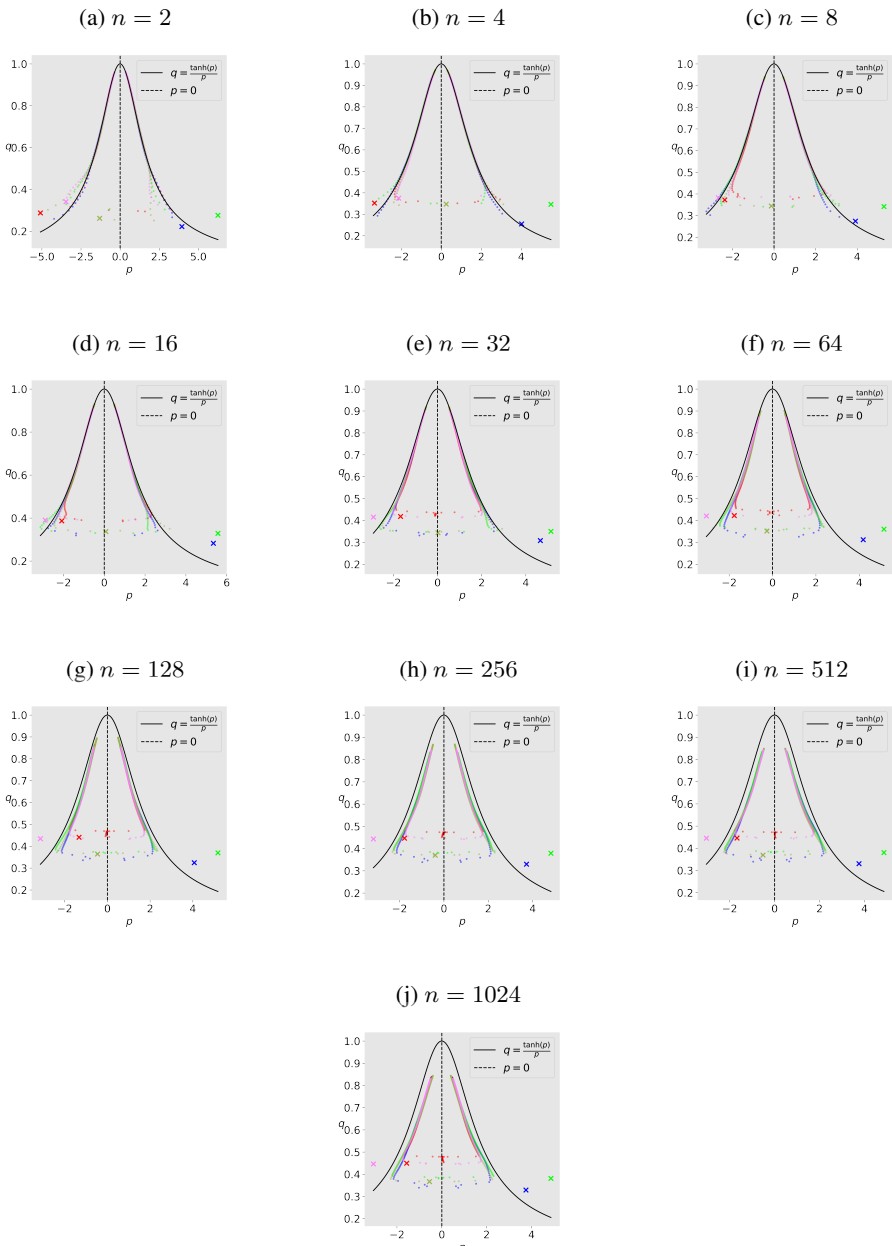

Figure 10: GD trajectories (500 iterations) under the generalized canonical reparameterization (Eq. (5)) for **different choices of the number of data points** $n$. The plots depict the training of a 3-layer fully connected neural network with $\tanh$ activation, a width of $m = 256$, and an initialization scale of $\alpha = 4$. The function $P$ is chosen to be the mean $P(\boldsymbol{z}) = \frac{1}{n} \sum_{i=1}^{n} z_i$. The trajectories exhibit alignment behavior, where the curves followed by the trajectories change depending on the value of $n$. For small values of $n$, the trajectories align on the curve $q = \frac{\ell'(p)}{p}$, while for large values of $n$, they align on a distinct curve.

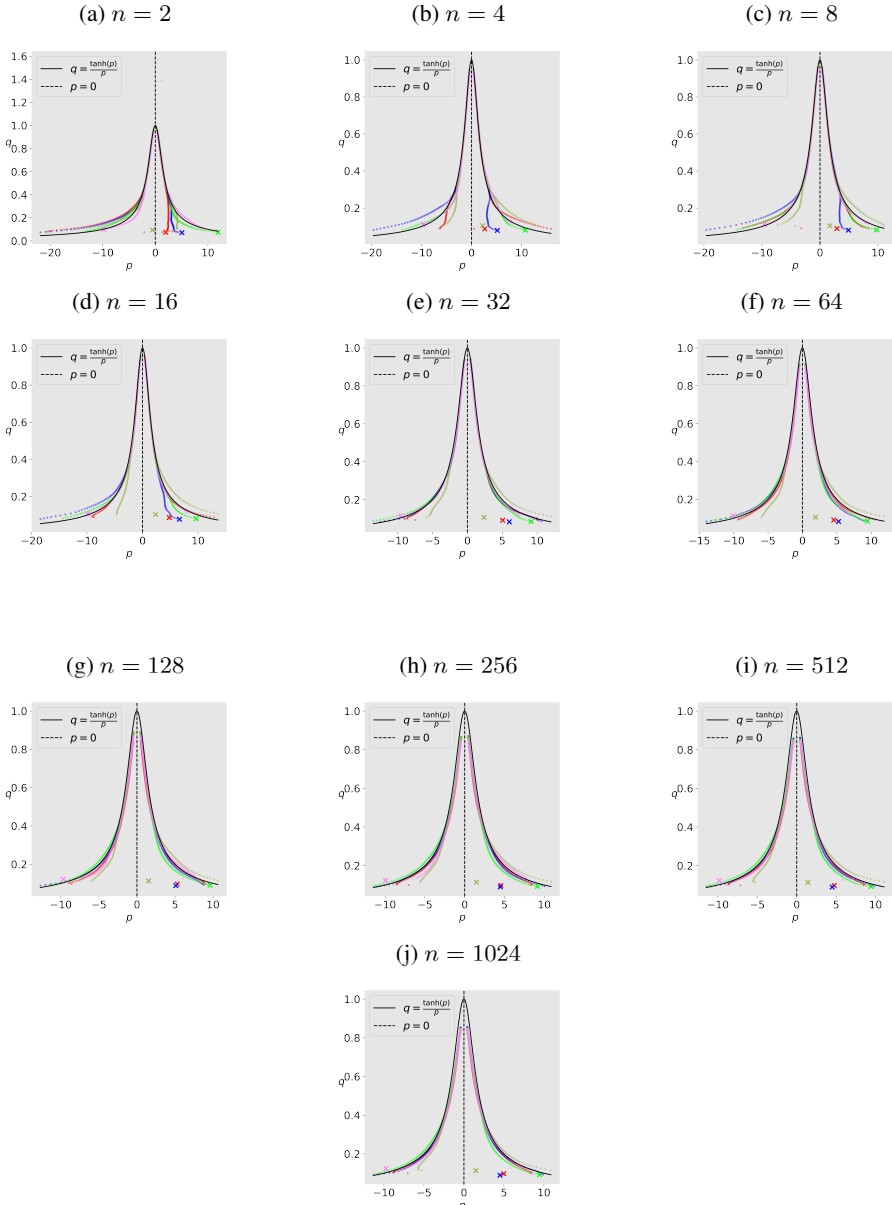

Figure 11: GD trajectories (500 iterations) under the generalized canonical reparameterization (Eq. (5)) for **different choices of the number of data points** $n$. The plots depict the training of a 3-layer fully connected neural network with ELU activation, a width of $m = 256$, and an initialization scale of $\alpha = 4$. The function $P$ is chosen to be the mean $P(\boldsymbol{z}) = \frac{1}{n} \sum_{i=1}^{n} z_i$. The trajectories exhibit alignment behavior, where the curves followed by the trajectories change depending on the value of $n$. For small values of $n$, the trajectories align on the curve $q = \frac{\ell'(p)}{p}$, while for large values of $n$, they align on a distinct curve.

### A.4 Training on real-world dataset

**Training data.** In this subsection, we investigate a binary classification problem using a subset of the CIFAR-10 image classification dataset. Our dataset consists of 50 samples, with 25 samples from class 0 (airplane) and 25 samples from class 1 (automobile). We assign a label of $+1$ to samples from class 0 and a label of $-1$ to samples from class 1. This dataset was used in the experimental setup by Zhu et al. (2023).

**Architectures.** In Figure 4, we examine the training of two types of network architectures: (top row) fully-connected $\tanh$ network and (bottom row) convolutional $\tanh$ network. The PyTorch code for the fully-connected $\tanh$ network is provided as follows:

```
nn.Sequential(
  nn.Flatten(start_dim=1, end_dim=-1),
  nn.Linear(3072, 500, bias=True),
  nn.Tanh(),
  nn.Linear(500, 500, bias=True),
  nn.Tanh(),
  nn.Linear(500, 1, bias=True)
)
```

Similarly, the PyTorch code for the convolutional $\tanh$ network is as follows:

```
nn.Sequential(
  nn.Flatten(start_dim=1, end_dim=-1),
  nn.Unflatten(dim=1, unflattened_size=(3, 32, 32)),
  nn.Conv2d(3, 500, kernel_size=(3, 3), stride=(1, 1), padding=(1, 1), bias=True),
  nn.Tanh(),
  nn.Conv2d(500, 500, kernel_size=(3, 3), stride=(1, 1), padding=(1, 1), bias=True),
  nn.Tanh(),
  nn.Flatten(start_dim=1, end_dim=-1),
  nn.Linear(512000, 1, bias=True)
)
```

Note that we consider networks *with* bias. We use a step size of $\eta = 0.01$ for the fully-connected network and $\eta = 0.001$ for the CNN. The default PyTorch initialization (Paszke et al., 2019) is applied to all these networks.

In this subsection, we further explore the (reparameterized) GD trajectories of fully-connected networks with different activation functions, network widths, and the choice of function $P$ in (5).

**The effect of function $P$.** We investigate the impact of different choices of the function $P$ on the GD trajectories. We train a 3-layer fully connected neural network with ELU activation, a width of $m = 256$, and an initialization scale of $\alpha = 1$. Figure 12 illustrates the GD trajectories under the generalized canonical reparameterization defined in Eq. (5) for various choices of the function $P$, including the mean, $\ell_1$ norm, $\ell_2$ norm, and $\ell_\infty$ norm.

We observe that the GD trajectories exhibit alignment behavior, which is more pronounced when $P$ is chosen to be the mean or $\ell_1$ norm, but less evident for the $\ell_\infty$ norm. Unlike in Figure 9, the trajectories do not align on the curve $q = \frac{\ell'(p)}{p}$ when $P$ is selected as the mean $P(\boldsymbol{z}) = \frac{1}{n}\sum_{i=1}^{n} z_i$.

**The effect of network width.** We investigate how the width of the network influences the trajectory alignment phenomenon. We vary the width $m$ using values from $\{64, 128, 256, 512\}$ while keeping other hyperparameters constant. In Figure 13, we train 3-layer fully connected neural networks with $\tanh$ activation and an initialization scale of $\alpha = 1$. Similarly, in Figure 14, we conduct experiments using the same configuration but with ELU activation, training ELU-activated fully connected neural networks.

Consistent with the observations from Figures 6 and 7 in the single data point setting, we commonly find that as the network width increases, the alignment trend becomes more pronounced. In both Figure 13 and Figure 14, all trajectories fall within the EoS regime. However, narrower networks

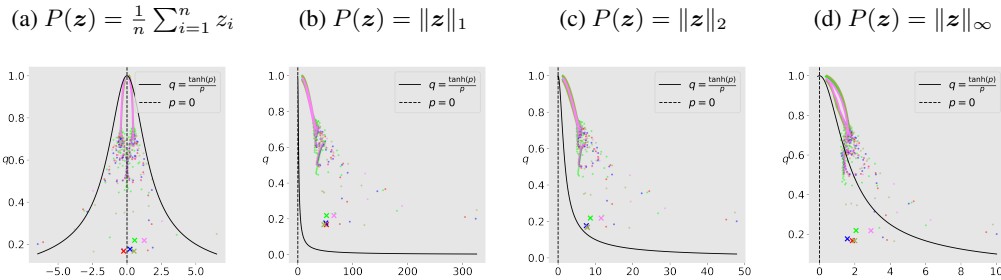

Figure 12: GD trajectories (1000 iterations) under the generalized canonical reparameterization (Eq. (5)) for different choices of the function $P$ trained on a small subset of CIFAR-10 image dataset. The plots depict the training of a 3-layer fully connected neural network with ELU activation, a width of $m = 256$, and an initialization scale of $\alpha = 1$. The function $P$ is varied to be the mean, $\ell_1$ norm, $\ell_2$ norm, and $\ell_\infty$ norm. The alignment behavior of the trajectories is more prominent when $P$ is chosen as the mean or $\ell_1$ norm, but less evident for the $\ell_\infty$ norm.

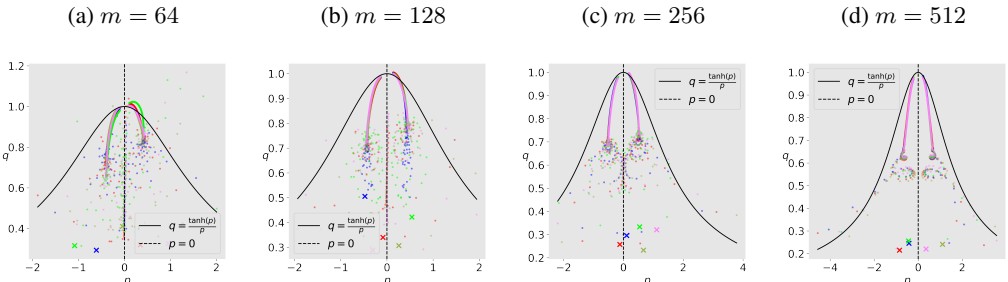

Figure 13: GD trajectories (1000 iterations) under the generalized canonical reparameterization (Eq. (5)) for **different network widths** trained on a small subset of CIFAR-10 image dataset. The plots depict the training of 3-layer fully connected neural networks with $\tanh$ activation and an initialization scale of $\alpha = 1$. The trajectories are shown for network widths of $m = 64$, $m = 128$, $m = 256$, and $m = 512$.

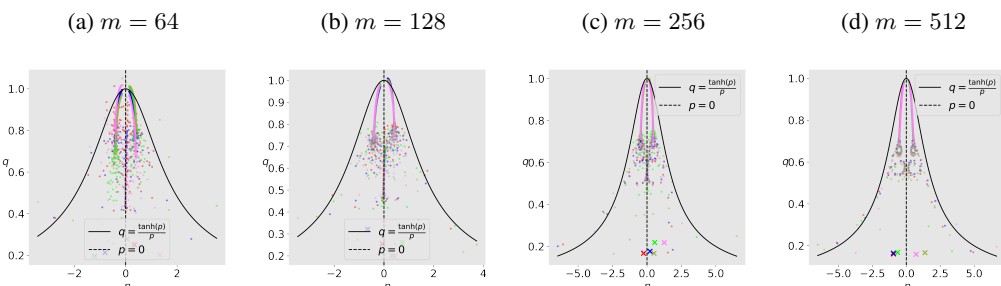

Figure 14: GD trajectories (1000 iterations) under the generalized canonical reparameterization (Eq. (5)) for **different network widths** trained on a small subset of CIFAR-10 image dataset. The plots depict the training of 3-layer fully connected neural networks with ELU activation and an initialization scale of $\alpha = 1$. The trajectories are shown for network widths of $m = 64$, $m = 128$, $m = 256$, and $m = 512$.

($m = 64$) show less evidence of the trajectory alignment phenomenon, while wider networks ($m = 256, 512$) clearly demonstrate this behavior. These findings emphasize the significant impact of network width on the trajectory alignment property of GD.

**The effect of data label.** In our previous experiments, we assigned labels of $+1$ and $-1$ to the dataset. However, in this particular experiment, we investigate the training process on a dataset with

zero labels. This means that all samples in the dataset are labeled as zero ($y_i = 0$ for all $1 \leq i \leq n$). Figure 15 visualizes the training of 3-layer fully connected neural networks with $\tanh$ activation and an initialization scale of $\alpha = 1$. The network widths $m$ are varied from $\{256, 512, 1024\}$. Interestingly, the GD trajectories align with the curve $q = \frac{\ell'(p)}{p}$, in contrast to our observations in Figures 13 and 14. These results suggest that the data label distribution also influences the alignment curve of GD trajectories. As a future research direction, it would be intriguing to investigate why setting the labels as zero leads to alignment towards the curve $q = \frac{\ell'(p)}{p}$, which aligns with our theoretical findings in the single data point setting.

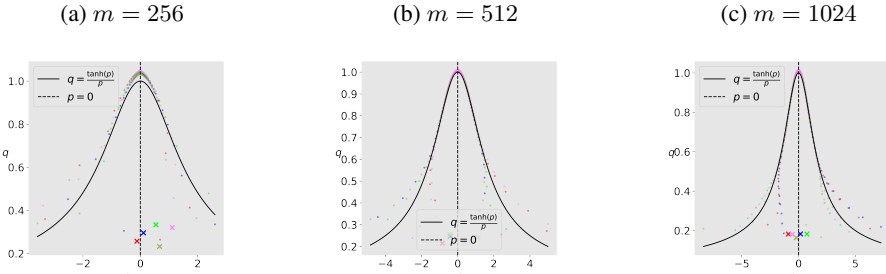

Figure 15: GD trajectories (1000 iterations) of the training for 3-layer fully connected neural networks with $\tanh$ activation and an initialization scale of $\alpha = 1$. The networks are trained on a small subset of the CIFAR-10 dataset, where **all labels are set to zero**. The network widths $m$ are varied, including values of $256$, $512$, and $1024$. GD trajectories exhibit alignment behavior, aligning on the curve $q = \frac{\ell'(p)}{p}$.

# B    Proofs for the Two-Layer Fully-Connected Linear Network

## B.1    Proof of Theorem 4.1

We give the proof of Theorem 4.1, restated below for the sake of readability.

**Theorem 4.1** (gradient flow regime). *Let $\eta \in (0, \frac{2}{33})$ be a fixed step size and $\ell$ be a loss function satisfying Assumptions 4.1 and 4.2. Suppose that the initialization $(p_0, q_0)$ satisfies $|p_0| \leq 1$ and $q_0 \in \left( \frac{2}{2-\eta}, \min\left\{ \frac{1}{16\eta}, \frac{r(1)}{2\eta} \right\} \right)$. Consider the GD trajectory characterized in Eq. (6). Then, the GD iterations $(p_t, q_t)$ converge to the point $(0, q^*)$ such that*

$$q_0 \leq q^* \leq \exp(C\eta^2)q_0 \leq 2q_0,$$

*where $C = 8q_0 \left[ \min\left\{ \frac{2(q_0-1)}{q_0}, \frac{r(1)}{2q_0} \right\} \right]^{-1} > 0$.*

We note that in the given interval $\left( \frac{2}{2-\eta}, \min\left\{ \frac{1}{16\eta}, \frac{r(1)}{2\eta} \right\} \right)$, it is possible that the interval is empty, depending on the value of $r(1)$. However, this does not impact the correctness of the theorem.

*Proof.* By Proposition B.1, $p_t$ converges to 0 as $t \to \infty$, and for all $t \geq 0$, we have

$$q_0 \leq q_t \leq \exp(C\eta^2)q_0.$$

Since the sequence $(q_t)_{t=0}^{\infty}$ is monotonic increasing and bounded, it converges. Suppose that $q_t \to q^*$ as $t \to \infty$. Then, we can obtain the inequality

$$q_0 \leq q^* \leq \exp(C\eta^2)q_0,$$

as desired. $\square$

**Proposition B.1.** *Suppose that $\eta \in (0, \frac{2}{33})$, $|p_0| \leq 1$, and $q_0 \in \left( \frac{2}{2-\eta}, \min\left\{ \frac{1}{16\eta}, \frac{r(1)}{2\eta} \right\} \right)$. Then for any $t \geq 0$, we have*

$$|p_t| \leq \left[ 1 - \min\left\{ \frac{2(q_0-1)}{q_0}, \frac{r(1)}{2q_0} \right\} \right]^t \leq 1,$$

*and*

$$q_0 \leq q_t \leq \exp\left( 8\eta^2 q_0 \left[ \min\left\{ \frac{2(q_0-1)}{q_0}, \frac{r(1)}{2q_0} \right\} \right]^{-1} \right) q_0 \leq 2q_0.$$

*Proof.* We give the proof by induction; namely, if

$$|p_t| \leq \left[ 1 - \min\left\{ \frac{2(q_0-1)}{q_0}, \frac{r(1)}{2q_0} \right\} \right]^t, \quad q_0 \leq q_t \leq \exp\left( 8\eta^2 q_0 \left[ \min\left\{ \frac{2(q_0-1)}{q_0}, \frac{r(1)}{2q_0} \right\} \right]^{-1} \right) q_0 \leq 2q_0$$

are satisfied for time steps $0 \leq t \leq k$ for some $k$, then the inequalities are also satisfied for the next time step $k+1$.

For the base case, the inequalities are satisfied for $t = 0$ by assumptions. For the induction step, we assume that the inequalities hold for any $0 \leq t \leq k$. We will prove that the inequalities are also satisfied for $t = k+1$.

By induction assumptions, $q_0 \leq q_k \leq 2q_0$ and $|p_k| \leq 1$, so that we have

$$1 - \frac{2}{q_0} \leq \frac{p_{k+1}}{p_k} = 1 - \frac{2r(p_k)}{q_k} + \eta^2 p_k^2 r(p_k)^2 \leq 1 - \frac{r(1)}{q_0} + \eta^2,$$

where we used $r(1) \leq r(p_k) \leq r(0) = 1$. Since $q_0 \leq \frac{r(1)}{2\eta} \leq \frac{r(1)}{2\eta^2}$, we have

$$\left| \frac{p_{k+1}}{p_k} \right| \leq \max\left\{ \frac{2}{q_0} - 1, 1 - \frac{r(1)}{q_0} + \eta^2 \right\}$$

$$= 1 - \min\left\{ \frac{2(q_0-1)}{q_0}, \frac{r(1)}{q_0} - \eta^2 \right\}$$

$$\leq 1 - \min\left\{\frac{2(q_0 - 1)}{q_0}, \frac{r(1)}{2q_0}\right\}.$$

This implies that

$$|p_{k+1}| \leq \left(1 - \min\left\{\frac{2(q_0 - 1)}{q_0}, \frac{r(1)}{2q_0}\right\}\right)|p_k| \leq \left[1 - \min\left\{\frac{2(q_0 - 1)}{q_0}, \frac{r(1)}{2q_0}\right\}\right]^{k+1},$$

which is the desired bound for $|p_{k+1}|$.

For any $t \leq k$, we also have

$$1 - 4\eta^2 p_t^2 q_0 \leq \frac{q_t}{q_{t+1}} = 1 - \eta^2 p_t^2 r(p_t)(2q_t - r(p_t)) \leq 1,$$

where we used the induction assumptions to deduce $4q_0 \geq 2q_t - r(p_t) \geq 2q_0 - 1 \geq \frac{4}{2-\eta} - 1 > 0$. This gives $q_{k+1} \geq q_k \geq q_0$. Furthermore, note that $4\eta^2 p_t^2 q_0 \leq 4\eta^2 \cdot \frac{1}{16\eta} \leq \frac{1}{4}$, so the ratio $\frac{q_t}{q_{t+1}} \in [\frac{3}{4}, 1]$. From this, we have

$$\left|\log\left(\frac{q_0}{q_{k+1}}\right)\right| \leq \sum_{t=0}^{k}\left|\log\left(\frac{q_t}{q_{t+1}}\right)\right| \leq 2\sum_{t=0}^{k}\left|\frac{q_t}{q_{t+1}} - 1\right|$$

$$\leq 8\eta^2 q_0 \sum_{t=0}^{k} p_t^2$$

$$\leq 8\eta^2 q_0 \sum_{t=0}^{k}\left[1 - \min\left\{\frac{2(q_0 - 1)}{q_0}, \frac{r(1)}{2q_0}\right\}\right]^{2t}$$

$$\leq 8\eta^2 q_0 \left[\min\left\{\frac{2(q_0 - 1)}{q_0}, \frac{r(1)}{2q_0}\right\}\right]^{-1},$$

where the second inequality holds since $|\log(1 + z)| \leq 2|z|$ if $|z| \leq \frac{1}{2}$. Moreover, this implies that

$$q_{k+1} \leq \exp\left(8\eta^2 q_0 \left[\min\left\{\frac{2(q_0 - 1)}{q_0}, \frac{r(1)}{2q_0}\right\}\right]^{-1}\right) q_0.$$

Since $q_0 \in \left(\frac{2}{2-\eta}, \frac{r(1)}{2\eta}\right)$, we have

$$\min\left\{\frac{2(q_0 - 1)}{q_0}, \frac{r(1)}{2q_0}\right\} \geq \eta.$$

Therefore, since $q_0 \leq \frac{1}{16\eta}$, we can conclude that

$$q_0 \leq q_{k+1} \leq \exp\left(8\eta^2 q_0 \left[\min\left\{\frac{2(q_0 - 1)}{q_0}, \frac{r(1)}{2q_0}\right\}\right]^{-1}\right) q_0 \leq \exp(8\eta q_0) q_0 \leq 2q_0,$$

the desired bounds for $q_{k+1}$. $\qquad\square$

### B.2 Proof of Theorem 4.2

In this subsection, we prove Theorem 4.2. From here onwards, we use the following notation:

$$s_t := \frac{q_t}{r(p_t)}.$$

All the lemmas in this subsection are stated in the context of Theorem 4.2.

**Lemma B.2.** *Suppose that the initialization $(p_0, q_0)$ satisfies $|p_0| \leq 1$ and $q_0 \in (c_0, 1 - \delta)$. Then for any $t \geq 0$ such that $q_t \leq 1$, it holds that*

$$|p_t| \leq 4, \text{ and } q_t \leq q_{t+1} \leq (1 + \mathcal{O}(\eta^2))q_t.$$

*Proof.* We prove by induction. We assume that for some $t \geq 0$, it holds that $|p_t| \leq 4$ and $\frac{1}{2} \leq q_t \leq 1$. We will prove that $|p_{t+1}| \leq 4$ and $\frac{1}{2} \leq q_t \leq q_{t+1} \leq (1 + \mathcal{O}(\eta^2))q_t$. For the base case, $|p_0| \leq 1 \leq 4$ and $\frac{1}{2} \leq c_0 < q_t \leq 1$ holds by the assumptions on the initialization. Now suppose that for some $t \geq 0$, it holds that $|p_t| \leq 4$ and $\frac{1}{2} \leq q_t \leq 1$. Then for small step size $\eta$, we have

$$\left| \frac{2\ell'(p_t)}{q_t} \right| \geq 2|\ell'(p_t)| \geq \frac{1}{2}\ell'(p_t)^2|p_t| \geq \eta^2\ell'(p_t)^2|p_t|.$$

Consequently, by Eq. (6),

$$|p_{t+1}| = \left| (1 + \eta^2\ell'(p_t)^2)p_t - \frac{2\ell'(p_t)}{q_t} \right| \leq \max\left\{ |p_t|, \frac{2}{q_t} \right\} \leq 4.$$

where we used 1-Lipshitzness of $\ell$. Moreover,

$$1 - 8\eta^2 \leq 1 - 2|p_t|\eta^2 \leq \frac{q_t}{q_{t+1}} = 1 - \eta^2 p_t^2 r(p_t)(2q_t - r(p_t)) \leq 1,$$

where we used $q_t \in [\frac{1}{2}, 1]$ and $|p_t r(p_t)| = |\ell'(p_t)| \leq 1$ from 1-Lipschitzness of $\ell$. Hence, $q_t \leq q_{t+1} \leq (1 + \mathcal{O}(\eta^2))q_t$, as desired. $\square$

Lemma B.2 implies that $p_t$ is bounded by a constant throughout the iterations, and $q_t$ monotonically increases slowly, where the increment for each step is $\mathcal{O}(\eta^2)$. Hence, there exists a time step $T = \Omega(\delta\eta^{-2}) = \Omega_\delta(\eta^{-2})$ such that for any $t \leq T$, it holds that $q_t \leq 1 - \frac{\delta}{2}$. Through out this subsection, we focus on these $T$ early time steps. Note that for all $0 \leq t \leq T$, it holds that $q_t \in (c_0, 1 - \frac{\delta}{2})$.

**Intuition on Theorem 4.2.** Before we dive in to the rigorous proofs, we provide an intuition on Theorem 4.2. Lemma B.2 establishes that $p_t$ is bounded and $q_t$ monotonically increases slowly, with an increment of $\mathcal{O}(\eta^2)$ per step. Lemma 2.1 shows that the map $f_{q_t}(p) = \left(1 - \frac{2r(p_t)}{q_t}\right)p_t$ has a stable 2-period orbit $\{\pm\hat{r}(q_t)\}$ when $q_t \in (0, 1)$. Consequently, when $q_t$ is treated as a fixed value, $p_t$ converges to the orbit $\{\pm\hat{r}(q_t)\}$, leading to $s_t$ converging to 1. In the (early) short-term dynamics, $q_t$ is nearly fixed for small step size $\eta$, and hence $s_t$ converges to 1. In long-term dynamics perspective, $q_t$ gradually increases and at the same time, $s_t$ stays near the value 1. In Theorem 4.2, we prove that it takes only $t_a = \mathcal{O}_{\delta,\ell}(\log(\eta^{-1}))$ time steps for $s_t$ to converge close to 1 (Phase I, $t \leq t_a$), and after that, $s_t$ stay close to 1 for the remaining iterations (Phase II, $t > t_a$).

We informally summarize the lemmas used in the proof of Theorem 4.2. Lemma B.3 states that in the early phase of training, there exists a time step $t_0$ where $s_{t_0}$ becomes smaller or equal to $\frac{2}{2-r(1)}$, which is smaller than $2(1 + \eta^2)^{-1}$. Lemma B.5 demonstrates that if $s_t$ is smaller than $2(1 + \eta^2)^{-1}$ and $|p_t| \geq \hat{r}(1 - \frac{\delta}{4})$, then $|s_t - 1|$ decreases exponentially. For the case where $|p_t| < \hat{r}(1 - \frac{\delta}{4})$, Lemma B.6 proves that $|p_t|$ increases at an exponential rate. Moreover, Lemma B.4 shows that if $s_t < 1$ at some time step, then $s_{t+1}$ is upper bounded by $1 + \mathcal{O}(\eta^2)$. Combining these findings, Proposition B.7 establishes that in the early phase of training, there exists a time step $t_a^*$ such that $s_{t_a^*} = 1 + \mathcal{O}_{\delta,\ell}(\eta^2)$. Lastly, Lemma B.8 demonstrates that if $s_t = 1 + \mathcal{O}_{\delta,\ell}(\eta^2)$, then $|s_t - 1 - h(p_t)\eta^2|$ decreases exponentially.

Now we prove Theorem 4.2, starting with the lemma below.

**Lemma B.3.** *There exists a time step $t_0 = \mathcal{O}_{\delta,\ell}(1)$ such that $s_{t_0} \leq \frac{2}{2-r(1)}$.*

*Proof.* We start by proving the following statement: for any $0 \leq t \leq T$, if $\frac{2}{2-r(1)} < s_t < 2r(1)^{-1}$, then $s_{t+1} < 2r(1)^{-1}$ and $|p_{t+1}| \leq (1 - \frac{r(1)}{2})|p_t|$. Suppose that $\frac{2}{2-r(1)} < s_t < 2r(1)^{-1}$. Then from Eq. (6), it holds that

$$\left| \frac{p_{t+1}}{p_t} \right| = \left| 1 - \frac{2}{s_t} + \eta^2 p_t^2 r(p_t)^2 \right| \leq \left| 1 - \frac{2}{s_t} \right| + \eta^2 \leq 1 - r(1) + \eta^2 \leq 1 - \frac{r(1)}{2},$$

for small step size $\eta$. Hence, $|p_{t+1}| \leq (1 - \frac{r(1)}{2})|p_t|$. Now we prove $s_{t+1} < 2r(1)^{-1}$. Assume the contrary that $s_{t+1} \geq 2r(1)^{-1}$. Then, $r(p_{t+1}) = \frac{q_{t+1}}{s_{t+1}} < q_{t+1} < 1 - \frac{\delta}{2}$ so that $|p_{t+1}| \geq \hat{r}(1 - \frac{\delta}{2})$. By Mean Value Theorem, there exists $p_t^* \in (|p_{t+1}|, |p_t|)$ such that (recall that $\frac{r'(p)}{r(p)^2} < 0$ for $p > 0$)

$$
\begin{aligned}
\frac{1}{r(p_{t+1})} = \frac{1}{r(|p_t| - (|p_t| - |p_{t+1}|))} &= \frac{1}{r(p_t)} + \frac{r'(p_t^*)}{r(p_t^*)^2}(|p_t| - |p_{t+1}|) \\
&\leq \frac{1}{r(p_t)} + \frac{r'(|p_{t+1}|)}{r(|p_{t+1}|)^2}\left(\frac{r(1)|p_t|}{2}\right) \\
&\leq \frac{1}{r(p_t)} - \frac{|r'(\hat{r}(1 - \frac{\delta}{2}))|}{(1 - \frac{\delta}{2})^2}\left(\frac{r(1)\hat{r}(1 - \frac{\delta}{2})}{2}\right) \\
&= \frac{1}{r(p_t)} - \Omega_{\delta,\ell}(1),
\end{aligned}
$$

where we used Assumption 4.3 (i) and $\hat{r}(1 - \frac{\delta}{2}) \leq |p_{t+1}| \leq (1 - \frac{r(1)}{2})|p_t| \leq |p_t|$. Consequently,

$$
s_{t+1} = \frac{q_{t+1}}{r(p_{t+1})} = (1 + \mathcal{O}(\eta^2))q_t\left(\frac{1}{r(p_t)} - \Omega_{\delta,\ell}(1)\right) \leq \frac{q_t}{r(p_t)} = s_t < 2r(1)^{-1},
$$

for small step size $\eta$. This gives a contradiction to our assumption that $s_{t+1} \geq 2r(1)^{-1}$. Hence, we can conclude that $s_{t+1} < 2r(1)^{-1}$, as desired.

We proved that for any $0 \leq t \leq T$, if $\frac{2}{2-r(1)} < s_t < 2r(1)^{-1}$, it holds that $s_{t+1} < 2r(1)^{-1}$ and $|p_{t+1}| \leq (1 - \frac{r(1)}{2})|p_t|$. At initialization, $|p_0| \leq 1$ and $q_0 < 1$, so that $s_0 < r(1)^{-1}$. If $s_0 \leq \frac{2}{2-r(1)}$, then $t_0 = 0$ is the desired time step. Suppose that $s_0 > \frac{2}{2-r(1)}$. Then, we have $s_1 < 2r(1)^{-1}$ and $|p_1| \leq (1 - \frac{r(1)}{2})|p_0| \leq 1 - \frac{r(1)}{2}$. Then we have either $s_1 \leq \frac{2}{2-r(1)}$, or $\frac{2}{2-r(1)} < s_1 < 2r(1)^{-1}$. In the previous case, $t_0 = 1$ is the desired time step. In the latter case, we can repeat the same argument and obtain $s_2 < 2r(1)^{-1}$ and $|p_2| \leq (1 - \frac{r(1)}{2})^2$. By inductively repeating the same argument, we can obtain a time step $t_0 \leq \log(\hat{r}(1 - \frac{\delta}{2}))/\log(1 - \frac{r(1)}{2}) = \mathcal{O}_{\delta,\ell}(1)$ such that either $s_{t_0} \leq \frac{2}{2-r(1)}$, or $|p_{t_0}| \leq \hat{r}(1 - \frac{\delta}{2})$. In the latter case, $r(p_{t_0}) \geq 1 - \frac{\delta}{2} > q_{t_0}$, and hence $s_{t_0} < 1 < \frac{2}{2-r(1)}$. Therefore, $t_0 = \mathcal{O}_{\delta,\ell}(1)$ is the desired time step satisfying $s_{t_0} \leq \frac{2}{2-r(1)}$. $\qquad\square$

According to Lemma B.3, there exists a time step $t_0 = \mathcal{O}_{\delta,\ell}(1)$ such that $s_{t_0} \leq \frac{2}{2-r(1)} < 2(1 + \eta^2)^{-1}$ for small step size $\eta$. Now we prove the lemma below.

**Lemma B.4.** *Suppose that $s_t \leq 1$. Then, it holds that $s_{t+1} \leq 1 + \mathcal{O}(\eta^2)$.*

*Proof.* For any $p \in (0, \hat{r}(\frac{q_t}{2}))$, we have $r(p) \geq \frac{q_t}{2}$ so that $|f_{q_t}(p)| = (-1 + \frac{2r(p)}{q_t})p$. Hence,

$$
\frac{\partial}{\partial p}|f_{q_t}(p)| = \frac{2r(p)}{q_t}\left(1 + \frac{pr'(p)}{r(p)}\right) - 1,
$$

for any $p \in (0, \hat{r}(\frac{q_t}{2}))$. By Assumption 4.3 (ii) and convexity of $\ell$, both $r(p)$ and $1 + \frac{pr'(p)}{r(p)} = \frac{\ell''(p)}{r(p)}$ are positive, decreasing function on $(0, \hat{r}(\frac{q_t}{2}))$. Consequently, $\frac{\partial}{\partial p}|f_{q_t}(p)|$ is a decreasing function on $(0, \hat{r}(\frac{q_t}{2}))$.

Now note that $\frac{q_t}{2} < q_t < 1$, which means $\hat{r}(1) = 0 < \hat{r}(q_t) < \hat{r}(\frac{q_t}{2})$ by the definition of $\hat{r}$. Note that $\frac{\partial}{\partial p}|f_{q_t}(p)|$ at $p = \hat{r}(q_t)$ evaluates to

$$
\frac{\partial}{\partial p}|f_{q_t}(\hat{r}(q_t))| = 1 + \frac{2\hat{r}(q_t)r'(\hat{r}(q_t))}{r(\hat{r}(q_t))} \geq 1 + \frac{2\hat{r}(c_0)r'(\hat{r}(c_0))}{r(\hat{r}(c_0))} \geq 0,
$$

where the first inequality used Assumption 4.3 (ii) and $\hat{r}(q_t) < \hat{r}(c_0)$, which comes from $q_t > c_0 := \max\{r(z_0), \frac{1}{2}\}$. The second inequality holds because $q_t > c_0 \geq r(z_0)$ where $z_0 := \sup_z\{\frac{zr'(z)}{r(z)} \geq -\frac{1}{2}\}$, from the statement of Theorem 4.2.

Therefore, since $\frac{\partial}{\partial p}|f_{q_t}(p)|$ is decreasing on $(0, \hat{r}(\frac{q_t}{2}))$ and is nonnegative at $\hat{r}(q_t)$, for any $p \in (0, \hat{r}(q_t))$, it holds that $\frac{\partial}{\partial p}|f_{q_t}(p)| \geq 0$. In other words, $|f_{q_t}(p)|$ is an increasing function on $(0, \hat{r}(q_t))$. Since $0 \leq s_t \leq 1$, we have $|p_t| \leq \hat{r}(q_t)$ and it holds that

$$|p_{t+1}| = \left(-1 + \frac{2}{s_t} - \eta^2 p_t^2 r(p_t)^2\right)|p_t| \leq \left(-1 + \frac{2}{s_t}\right)|p_t| = |f_{q_t}(p_t)| \leq |f_{q_t}(\hat{r}(q_t))| = \hat{r}(q_t).$$

Therefore, with this inequality and Lemma B.2, we can conclude that

$$s_{t+1} = \frac{q_{t+1}}{r(p_{t+1})} = \frac{q_t}{r(p_{t+1})}(1 + \mathcal{O}(\eta^2)) \leq \frac{q_t}{r(\hat{r}(q_t))}(1 + \mathcal{O}(\eta^2)) = 1 + \mathcal{O}(\eta^2).$$

$\square$

Using Lemma B.4, we prove the following lemma.

**Lemma B.5.** *For any $0 \leq t \leq T$, if $s_t < 2(1 + \eta^2)^{-1}$, $|s_t - 1| > \frac{\eta^2}{2}$, and $r(p_t) \leq 1 - \frac{\delta}{4}$, then*

$$|s_{t+1} - 1| \leq (1 - d)|s_t - 1| + \mathcal{O}(\eta^2),$$

*where $d \in (0, \frac{1}{2}]$ is a constant which depends on $\delta$ and $\ell$.*

*Proof.* By Eq. (6) and 1-Lipschitzness of $\ell$,

$$\frac{p_{t+1}}{p_t} = 1 - \frac{2}{s_t} + \eta^2 p_t^2 r(p_t)^2 < 1 - (1 + \eta^2) + \eta^2 = 0,$$

so that $p_t$ and $p_{t+1}$ have opposite signs. By Mean Value Theorem, there exists $\theta_t$ between $-1$ and $(1 - \frac{2}{s_t} + \eta^2 p_t^2 r(p_t)^2)$ satisfying

$$
\begin{aligned}
\frac{1}{r(p_{t+1})} &= \frac{1}{r\left(-p_t + \left(\frac{2(s_t-1)}{s_t} + \eta^2 p_t^2 r(p_t^2)\right) p_t\right)} \\
&= \frac{1}{r(-p_t)} - \frac{r'(\theta_t p_t)}{r(\theta_t p_t)^2}\left(\frac{2(s_t-1)}{s_t} + \eta^2 p_t^2 r(p_t)^2\right) p_t \\
&= \frac{1}{r(p_t)} - \frac{|r'(\theta_t p_t)|}{r(\theta_t p_t)^2}\left(\frac{2(s_t-1)}{s_t} + \eta^2 p_t^2 r(p_t)^2\right)|p_t|, \quad (8)
\end{aligned}
$$

where the last equality used the fact that $p_t$ and $\theta_t p_t$ have opposite signs and $r'(z)$ and $z$ have opposite signs. Note that $|\theta_t p_t|$ is between $|p_t|$ and $|p_{t+1}|$. Consequently, the value $\frac{|r'(\theta_t p_t)|}{r(\theta_t p_t)^2}$ is between $\frac{|r'(p_t)|}{r(p_t)^2}$ and $\frac{|r'(p_{t+1})|}{r(p_{t+1})^2}$ by Assumption 4.3 (i). We will prove the current lemma based on Eq. (8). We divide into following three cases: (1) $s_t \geq 1$ and $s_{t+1} \geq 1$, (2) $s_t \geq 1$ and $s_t < 1$, and (3) $s_t < 1$.

**Case 1.** Suppose that $s_t \geq 1$ and $s_{t+1} \geq 1$. Here, we have $|p_t| \geq \hat{r}(q_t) \geq \hat{r}(1 - \frac{\delta}{2})$ and similarly $|p_{t+1}| \geq \hat{r}(1 - \frac{\delta}{2})$. By Assumption 4.3 (i), $\frac{|r'(\theta_t p_t)|}{r(\theta_t p_t)^2} \geq \frac{|r'(\hat{r}(1-\frac{\delta}{2}))|}{(1-\frac{\delta}{2})^2}$. Hence, Eq. (8) gives

$$\frac{1}{r(p_{t+1})} \leq \frac{1}{r(p_t)} - \frac{|r'(\hat{r}(1-\frac{\delta}{2}))|}{(1-\frac{\delta}{2})^2}\left(\frac{2(s_t-1)}{s_t}\right)\hat{r}\left(1 - \frac{\delta}{2}\right).$$

Consequently, by Lemma B.2,

$$
\begin{aligned}
s_{t+1} = \frac{q_t(1 + \mathcal{O}(\eta^2))}{r(p_{t+1})} &= \frac{q_t}{r(p_{t+1})} + \mathcal{O}(\eta^2) \leq s_t - \frac{|r'(\hat{r}(1-\frac{\delta}{2}))|}{(1-\frac{\delta}{2})^2}\left(\frac{2(s_t-1)}{s_t}\right)\hat{r}\left(1 - \frac{\delta}{2}\right)q_t + \mathcal{O}(\eta^2) \\
&\leq s_t - \frac{|r'(\hat{r}(1-\frac{\delta}{2}))|}{(1-\frac{\delta}{2})^2}(s_t - 1)\hat{r}\left(1 - \frac{\delta}{2}\right)\frac{1}{2} + \mathcal{O}(\eta^2) \\
&\leq s_t - \frac{\hat{r}(1-\frac{\delta}{2})|r'(\hat{r}(1-\frac{\delta}{2}))|}{2(1-\frac{\delta}{2})^2}(s_t - 1) + \mathcal{O}(\eta^2),
\end{aligned}
$$

where we used $q_t > c_0 \geq \frac{1}{2}$ and $s_t < 2(1 + \eta^2)^{-1} < 2$. Therefore, we can obtain the following inequality:

$$0 \leq s_{t+1} - 1 \leq \left(1 - \frac{\hat{r}(1-\frac{\delta}{2})|r'(\hat{r}(1-\frac{\delta}{2}))|}{2(1-\frac{\delta}{2})^2}\right)(s_t - 1) + \mathcal{O}(\eta^2).$$

**Case 2.** Suppose that $s_t \geq 1$ and $s_{t+1} < 1$. Here, we have $r(p_{t+1}) > q_{t+1} \geq q_t \geq r(p_t)$, so that $|p_{t+1}| < |p_t|$. Consequently, $\frac{|r'(\theta_t p_t)|}{r(\theta_t p_t)^2} \leq \frac{|r'(p_t)|}{r(p_t)^2}$ by Assumption 4.3 (i). Hence, we can deduce from Eq. (8) that

$$
\begin{aligned}
\frac{1}{r(p_{t+1})} &\geq \frac{1}{r(p_t)} - \frac{|r'(p_t)|}{r(p_t)^2}\left(\frac{2(s_t - 1)}{s_t} + \eta^2 p_t^2 r(p_t)^2\right)|p_t| \\
&= \frac{1}{r(p_t)} - \frac{2|p_t r'(p_t)|}{r(p_t)q_t}(s_t - 1) - \eta^2 |p_t^3 r'(p_t)| \\
&\geq \frac{1}{r(p_t)} - \frac{2|p_t r'(p_t)|}{r(p_t)q_t}(s_t - 1) - \eta^2 p_t^2 r(p_t) \\
&= \frac{1}{r(p_t)} + \frac{2p_t r'(p_t)}{r(p_t)q_t}(s_t - 1) - \mathcal{O}(\eta^2),
\end{aligned}
$$

where we used $|p_t r'(p_t)| \leq r(p_t)$ since $1 + \frac{p_t r'(p_t)}{r(p_t)} = \frac{\ell''(p_t)}{r(p_t)} > 0$ and $|p_t| \leq 4$ by Lemma B.2. Consequently, by Lemma B.2 ($q_t \leq q_{t+1}$) and Assumption 4.3 (ii),

$$
s_{t+1} \geq \frac{q_t}{r(p_{t+1})} \geq s_t + \frac{2p_t r'(p_t)}{r(p_t)}(s_t - 1) - \mathcal{O}(\eta^2) \geq s_t + \frac{8r'(4)}{r(4)}(s_t - 1) - \mathcal{O}(\eta^2).
$$

Note that $1 > 1 + \frac{4r'(4)}{r(4)} = \frac{\ell''(4)}{r(4)} > 0$ holds by convexity of $\ell$. Therefore, we can obtain the following inequality:

$$
0 \leq 1 - s_{t+1} \leq -\left(1 + \frac{8r'(4)}{r(4)}\right)(s_t - 1) + \mathcal{O}(\eta^2),
$$

where $-1 < 1 + \frac{8r'(4)}{r(4)} < 1$.

**Case 3.** Suppose that $s_t < 1$. By Lemma B.4, it holds that $s_{t+1} \leq 1 + \mathcal{O}(\eta^2)$. Moreover, we assumed $r(p_t) \leq 1 - \frac{\delta}{4}$, so that $|p_t| \geq \hat{r}(1 - \frac{\delta}{4})$. We also have

$$
|p_{t+1}| = \left(-1 + \frac{2}{s_t} - \eta^2 p_t^2 r(p_t^2)\right)|p_t| \geq \left(-1 + \frac{2}{1 - \frac{\eta^2}{2}} - \eta^2\right)|p_t| > |p_t| \geq \hat{r}\left(1 - \frac{\delta}{4}\right),
$$

where we used the assumption $|s_t - 1| > \frac{\eta^2}{2}$, and $|pr(p)| = |\ell'(p)| \leq 1$ due to 1-Lipschitzness of $\ell$. Consequently, by Assumption 4.3 (i), it holds that $\frac{|r'(\theta_t p_t)|}{r(\theta_t p_t)^2} \geq \frac{|r'(\hat{r}(1 - \frac{\delta}{4}))|}{(1 - \frac{\delta}{4})^2}$. Hence, by Eq. (8), we have

$$
\begin{aligned}
\frac{1}{r(p_{t+1})} &\geq \frac{1}{r(p_t)} + \frac{|r'(\hat{r}(1 - \frac{\delta}{4}))|}{(1 - \frac{\delta}{4})^2}\left(\frac{2(1 - s_t)}{s_t}\right)\hat{r}\left(1 - \frac{\delta}{4}\right) \\
&\geq \frac{1}{r(p_t)} + \frac{|r'(\hat{r}(1 - \frac{\delta}{4}))|}{(1 - \frac{\delta}{4})^2}2(1 - s_t)\hat{r}\left(1 - \frac{\delta}{4}\right) \\
&= \frac{1}{r(p_t)} + \frac{2\hat{r}(1 - \frac{\delta}{4})|r'(\hat{r}(1 - \frac{\delta}{4}))|}{(1 - \frac{\delta}{4})^2}(1 - s_t),
\end{aligned}
$$

and hence, by Lemma B.2 ($q_t \leq q_{t+1}$) and $q_t > c_0 \geq \frac{1}{2}$, we get

$$
s_{t+1} \geq \frac{q_t}{r(p_{t+1})} \geq s_t + \frac{\hat{r}(1 - \frac{\delta}{4})|r'(\hat{r}(1 - \frac{\delta}{4}))|}{(1 - \frac{\delta}{4})^2}(1 - s_t).
$$

Therefore, we can obtain the following inequality:

$$
-\mathcal{O}(\eta^2) \leq 1 - s_{t+1} \leq \left(1 - \frac{\hat{r}(1 - \frac{\delta}{4})|r'(\hat{r}(1 - \frac{\delta}{4}))|}{(1 - \frac{\delta}{4})^2}\right)(1 - s_t),
$$

where we used Lemma B.4 to obtain the first inequality.

Combining the three cases, we can finally conclude that if we choose

$$d := \min\left\{\frac{1}{2}, \frac{\hat{r}(1-\frac{\delta}{2})|r'(\hat{r}(1-\frac{\delta}{2}))|}{2(1-\frac{\delta}{2})^2}, 2\left(1+\frac{4r'(4)}{r(4)}\right), \frac{\hat{r}(1-\frac{\delta}{4})|r'(\hat{r}(1-\frac{\delta}{4}))|}{(1-\frac{\delta}{4})^2}\right\} \in \left(0, \frac{1}{2}\right],$$

then $|s_{t+1} - 1| \leq (1-d)|s_t - 1| + \mathcal{O}(\eta^2)$. $\qquad\square$

Lemma B.5 implies that if $s_t < 2(1+\eta^2)^{-1}$ and $|p_t| \geq \hat{r}(1-\frac{\delta}{4})$, then $|s_t - 1|$ exponentially decreases. We prove Lemma B.6 to handle the regime $|p_t| < \hat{r}(1-\frac{\delta}{4})$, which is stated below.

**Lemma B.6.** *For any $0 \leq t \leq T$, if $r(p_t) \geq 1 - \frac{\delta}{4}$, it holds that*

$$\left|\frac{p_{t+1}}{p_t}\right| \geq \frac{4}{4-\delta}.$$

*Proof.* If $r(p_t) \geq 1 - \frac{\delta}{4}$, then $s_t = \frac{q_t}{r(p_t)} < \frac{1-\frac{\delta}{2}}{1-\frac{\delta}{4}} = \frac{4-2\delta}{4-\delta}$, where we used $q_t < 1 - \frac{\delta}{2}$ for any $0 \leq t \leq T$. Consequently,

$$\left|\frac{p_{t+1}}{p_t}\right| = \frac{2}{s_t} - 1 - \eta^2 p_t^2 r(p_t^2) \geq \frac{2(4-\delta)}{4-2\delta} - 1 - \eta^2 = \frac{2}{2-\delta} - \eta^2 \geq \frac{4}{4-\delta},$$

for small step size $\eta$. $\qquad\square$

Now we prove Proposition B.7, which proves that $s_t$ reaches close to 1 with error bound of $\mathcal{O}(\eta^2)$.

**Proposition B.7.** *There exists a time step $t_a^* = \mathcal{O}_{\delta,\ell}(\log(\eta^{-1}))$ satisfying*

$$s_{t_a^*} = 1 + \mathcal{O}_{\delta,\ell}(\eta^2). \tag{9}$$

*Proof.* By Lemma B.3, there exists a time step $t_0 = \mathcal{O}_{\delta,\ell}(1)$ such that $s_{t_0} \leq \frac{2}{2-r(1)}$. Here, we divide into two possible cases: (1) $s_{t_0} < 1$, and (2) $1 \leq s_{t_0} \leq \frac{2}{2-r(1)}$.

**Case 1.** Suppose that $s_{t_0} < 1$. By Lemma B.6, if $r(p_{t_0}) \geq 1 - \frac{\delta}{4}$ (or equivalently, $|p_{t_0}| \leq \hat{r}(1-\frac{\delta}{4})$), then there exists a time step $t_1 \leq t_0 + \log(\frac{\hat{r}(1-\frac{\delta}{4})}{|p_{t_0}|})/\log(\frac{4}{4-\delta}) = \mathcal{O}_{\delta,\ell}(1)$ such that $|p_{t_1}| \geq \hat{r}(1-\frac{\delta}{4})$. We denote the first time step satisfying $|p_{t_1}| \geq \hat{r}(1-\frac{\delta}{4})$ and $t_1 \geq t_0$ by $t_1 = \mathcal{O}_{\delta,\ell}(1)$. By Lemma B.4, it holds that $s_{t_1} \leq 1 + \mathcal{O}(\eta^2)$ since $s_{t_1-1} < 1$. Consequently, if $s_{t_1} \geq 1 - \frac{\eta^2}{2}$, then $|s_{t_1} - 1| \leq \mathcal{O}(\eta^2)$ so that $t_a^* = t_1$ is the desired time step. Hence, it suffices to consider the case when $s_{t_1} < 1 - \frac{\eta^2}{2}$. Here, we can apply Lemma B.5 which implies that

$$|s_{t_1+1} - 1| \leq (1-d)|s_{t_1} - 1| + \mathcal{O}(\eta^2),$$

where $d$ is a constant which depends on $\delta$ and $\ell$. Then, there are two possible cases: either $|s_{t_1} - 1| \leq \mathcal{O}(\eta^2 d^{-1})$, or $|s_{t_1+1} - 1| \leq (1-\frac{d}{2})|s_{t_1} - 1|$. It suffices to consider the latter case, suppose that $|s_{t_1+1} - 1| \leq (1-\frac{d}{2})|s_{t_1} - 1|$. Since we are considering the case $s_{t_1} < 1 - \frac{\eta^2}{2}$, again by Lemma B.4, we have $s_{t_1+1} \leq 1 + \mathcal{O}(\eta^2)$. Since $|\frac{p_{t_1+1}}{p_{t_1}}| = \frac{2}{s_{t_1}} - 1 - \mathcal{O}(\eta^2)$, $|p_{t_1+1}| \geq |p_{t_1}| \geq \hat{r}(1-\frac{\delta}{4})$ must be satisfied unless $s_{t_1} = 1 + \mathcal{O}(\eta^2)$ already holds. If $s_{t_1+1} \geq 1 - \frac{\eta^2}{2}$, then $|s_{t_1+1} - 1| \leq \mathcal{O}(\eta^2)$ so that $t_a^* = t_1 + 1$ is the desired time step; if not, we can again apply Lemma B.5 and repeat the analogous argument. Hence, there exists a time step $t_2 \leq t_1 + \log(\frac{\eta^2}{1-s_{t_1}})/\log(1-\frac{d}{2}) = \mathcal{O}_{\delta,\ell}(\log(\eta^{-1}))$, such that $|s_{t_2} - 1| \leq \mathcal{O}(\eta^2 d^{-1}) = \mathcal{O}_{\delta,\ell}(\eta^2)$.

**Case 2.** Suppose that $1 \leq s_{t_0} \leq \frac{2}{2-r(1)}$. Then, $r(p_{t_0}) \leq q_{t_0} \leq 1 - \frac{\delta}{2}$, so we can apply Lemma B.5. There are two possible cases: either $|s_{t_0+1} - 1| \leq \mathcal{O}(\eta^2 d^{-1}) = \mathcal{O}_{\delta,\ell}(\eta^2)$, or $|s_{t_0+1} - 1| \leq (1-\frac{d}{2})|s_{t_0} - 1|$. It suffices to consider the latter case. If $s_{t_0+1} \geq 1$, we can again apply Lemma B.5 and repeat the analogous argument. Hence, we can obtain a time step $t_0' \leq t_0 + \log(\frac{\eta^2}{1-s_{t_0}})/\log(1-\frac{d}{2}) = \mathcal{O}_{\delta,\ell}(\log(\eta^{-1}))$ such that either $s_{t_0'} < 1$ or $|s_{t_0'} - 1| = \mathcal{O}_{\delta,\ell}(\eta^2)$ is satisfied. If $s_{t_0'} < 1$, we proved in Case 1 that there exists a time step $t_2' = t_0' + \mathcal{O}_{\delta,\ell}(\log(\eta^{-1}))$ such that $|s_{t_2'} - 1| \leq \mathcal{O}_{\delta,\ell}(\eta^2)$, and this is the desired bound. $\qquad\square$

Now we carefully handle the error term $\mathcal{O}(\eta^2)$ obtained in Proposition B.7 and a provide tighter bound on $s_t$ by proving Lemma B.8 stated below.

**Lemma B.8.** *If* $|s_t - 1| = \mathcal{O}_{\delta,\ell}(\eta^2)$, *then it holds that*

$$|s_{t+1} - 1 - h(p_{t+1})\eta^2| \leq \left(1 + \frac{2p_t r'(p_t)}{r(p_t)}\right)|s_t - 1 - h(p_t)\eta^2| + \mathcal{O}_{\delta,\ell}(\eta^4 p_t^2),$$

*where* $h(p) := -\frac{1}{2}\left(\frac{pr(p)^3}{r'(p)} + p^2 r(p)^2\right)$ *for* $p \neq 0$ *and* $h(p) := -\frac{1}{2r''(0)}$ *for* $p = 0$.

*Proof.* Suppose that $s_t = 1 + \mathcal{O}_{\delta,\ell}(\eta^2)$. Then, $|p_{t+1}| = \left|1 - \frac{2}{s_t} + \eta^2 p_t^2 r(p_t)^2\right| \cdot |p_t| = (1 + \mathcal{O}_{\delta,\ell}(\eta^2))|p_t|$. By Eq. (8) proved in Lemma B.5, there exists $\epsilon_t = \mathcal{O}_{\delta,\ell}(\eta^2)$ which satisfies the following:

$$
\begin{aligned}
\frac{1}{r(p_{t+1})} &= \frac{1}{r(p_t)} + \frac{r'((1+\epsilon_t)p_t)}{r((1+\epsilon_t)p_t)^2}\left(\frac{2(s_t - 1)}{s_t} + \eta^2 p_t^2 r(p_t)^2\right)p_t \\
&= \frac{1}{r(p_t)} + \left(\frac{r'(p_t)}{r(p_t)^2} + \mathcal{O}_{\delta,\ell}(\eta^2 p_t)\right)\left(\frac{2(s_t - 1)}{s_t} + \eta^2 p_t^2 r(p_t)^2\right)p_t \\
&= \frac{1}{r(p_t)} + \frac{r'(p_t)}{r(p_t)^2}\left(\frac{2(s_t - 1)}{s_t} + \eta^2 p_t^2 r(p_t)^2\right)p_t + \mathcal{O}_{\delta,\ell}(\eta^4 p_t^2),
\end{aligned}
$$

where we used the Taylor expansion on $\frac{r'(p)}{r(p)^2}$ with the fact that $\frac{d}{dp}\left(\frac{r'(p)}{r(p)^2}\right)$ is bounded on $[-4, 4]$ and that $|p_t| \leq 4$ to obtain the second equality. Note that $q_{t+1} = (1 - \eta^2 p_t^2 r(p_t)(2q_t - r(p_t)))^{-1} q_t$ by Eq. (6). Consequently,

$$
\begin{aligned}
s_{t+1} &= (1 - \eta^2 p_t^2 r(p_t)(2q_t - r(p_t)))^{-1}\left(s_t + \frac{2p_t r'(p_t)}{r(p_t)}(s_t - 1) + \eta^2 p_t^3 r'(p_t)q_t\right) + \mathcal{O}_{\delta,\ell}(\eta^4 p_t^2) \\
&= (1 + \eta^2 p_t^2 r(p_t)(2q_t - r(p_t)))s_t + \frac{2p_t r'(p_t)}{r(p_t)}(s_t - 1) + \eta^2 p_t^3 r'(p_t)q_t + \mathcal{O}_{\delta,\ell}(\eta^4 p_t^2) \\
&= 1 + \left(1 + \frac{2p_t r'(p_t)}{r(p_t)}\right)(s_t - 1) + \eta^2 p_t^2 r(p_t)(2q_t - r(p_t))s_t + \eta^2 p_t^3 r'(p_t)q_t + \mathcal{O}_{\delta,\ell}(\eta^4 p_t^2).
\end{aligned}
$$

Here, since $s_t = 1 + \mathcal{O}_{\delta,\ell}(\eta^2)$, we can rewrite

$$
\begin{aligned}
&\eta^2 p_t^2 r(p_t)(2q_t - r(p_t))s_t + \eta^2 p_t^3 r'(p_t)q_t \\
&= \eta^2 p_t^2 r(p_t)^2(2s_t - 1)s_t + \eta^2 p_t^3 r'(p_t)r(p_t)s_t \\
&= \eta^2 p_t^2 r(p_t)^2 + \eta^2 p_t^3 r'(p_t)r(p_t) + \mathcal{O}_{\delta,\ell}(\eta^4 p_t^2),
\end{aligned}
$$

which results in

$$s_{t+1} = 1 + \left(1 + \frac{2p_t r'(p_t)}{r(p_t)}\right)(s_t - 1) + \eta^2 p_t^2 r(p_t)^2 + \eta^2 p_t^3 r(p_t)r'(p_t) + \mathcal{O}_{\delta,\ell}(\eta^4 p_t^2).$$

Note that $h$ is even, and twice continuously differentiable function by Lemma B.9. Consequently, $h'(0) = 0$ and $h'(p) = \mathcal{O}_\ell(p)$, since $h''$ is bounded on closed interval. Consequently, $h(p_{t+1}) = h((1 + \mathcal{O}_{\delta,\ell}(\eta^2))p_t) = h(p_t) + \mathcal{O}_{\delta,\ell}(\eta^2 p_t^2)$. Hence, we can obtain the following:

$$
\begin{aligned}
s_{t+1} - 1 - h(p_{t+1})\eta^2 &= s_{t+1} - 1 - h(p_t)\eta^2 + \mathcal{O}_{\delta,\ell}(\eta^4 p_t^2) \\
&= s_{t+1} - 1 + \frac{1}{2}\left(\frac{p_t r(p_t)^3}{r'(p_t)} + p_t^2 r(p_t)^2\right)\eta^2 + \mathcal{O}_{\delta,\ell}(\eta^4 p_t^2) \\
&= \left(1 + \frac{2p_t r'(p_t)}{r(p_t)}\right)\left(s_t - 1 + \frac{1}{2}\left(\frac{p_t r(p_t)^3}{r'(p_t)} + p_t^2 r(p_t)^2\right)\eta^2\right) + \mathcal{O}_{\delta,\ell}(\eta^4 p_t^2) \\
&= \left(1 + \frac{2p_t r'(p_t)}{r(p_t)}\right)(s_t - 1 - h(p_t)\eta^2) + \mathcal{O}_{\delta,\ell}(\eta^4 p_t^2).
\end{aligned}
$$

Note that $r(p_t) = (1 + \mathcal{O}_{\delta,\ell}(\eta^2))q_t \geq (1 + \mathcal{O}_{\delta,\ell}(\eta^2))q_0 \geq c_0 \geq r(z_0)$ for small step size $\eta$, where $z_0 = \sup\{\frac{zr'(z)}{r(z)} \geq -\frac{1}{2}\}$. Consequently, it holds that $1 + \frac{2p_t r'(p_t)}{r(p_t)} \geq 0$. Therefore, we have the desired inequality:

$$|s_{t+1} - 1 - h(p_{t+1})\eta^2| \leq \left(1 + \frac{2p_t r'(p_t)}{r(p_t)}\right)|s_t - 1 - h(p_t)\eta^2| + \mathcal{O}_{\delta,\ell}(\eta^4 p_t^2).$$

$\square$

We now provide the proof of Theorem 4.2, restated below for the sake of readability.

**Theorem 4.2** (EoS regime, Phase I). *Let $\eta$ be a small enough step size and $\ell$ be a loss function satisfying Assumptions 4.1, 4.2, and 4.3. Let $z_0 := \sup_z\{\frac{zr'(z)}{r(z)} \geq -\frac{1}{2}\}$ and $c_0 := \max\{r(z_0), \frac{1}{2}\}$. Let $\delta \in (0, 1 - c_0)$ be any given constant. Suppose that the initialization $(p_0, q_0)$ satisfies $|p_0| \leq 1$ and $q_0 \in (c_0, 1 - \delta)$. Consider the reparameterized GD trajectory characterized in Eq. (6). We assume that for all $t \geq 0$ such that $q_t < 1$, we have $p_t \neq 0$. Then, there exists a time step $t_a = \mathcal{O}_{\delta,\ell}(\log(\eta^{-1}))$, such that for any $t \geq t_a$,*

$$\frac{q_t}{r(p_t)} = 1 + h(p_t)\eta^2 + \mathcal{O}_{\delta,\ell}(\eta^4),$$

*where $h(p) := -\frac{1}{2}\left(\frac{pr(p)^3}{r'(p)} + p^2 r(p)^2\right)$ for $p \neq 0$ and $h(p) := -\frac{1}{2r''(0)}$ for $p = 0$.*

*Proof of Theorem 4.2.* By Proposition B.7, there exists a time step $t_a^* = \mathcal{O}_{\delta,\ell}(\log(\eta^{-1}))$ which satisfies:

$$|s_{t_a^*} - 1| = \left|\frac{q_{t_a^*}}{r(p_{t_a^*})} - 1\right| = \mathcal{O}_{\delta,\ell}(\eta^2).$$

By Lemma B.8, there exists a constant $D > 0$ which depends on $\delta, \ell$ such that if $|s_t - 1| = \mathcal{O}_{\delta,\ell}(\eta^2)$, then

$$|s_{t+1} - 1 - h(p_{t+1})\eta^2| \leq \left(1 + \frac{2p_t r'(p_t)}{r(p_t)}\right)|s_t - 1 - h(p_t)\eta^2| + D\eta^4 p_t^2. \tag{10}$$

Hence, if $|s_t - 1| = \mathcal{O}_{\delta,\ell}(\eta^2)$ and $|s_t - 1 - h(p_t)\eta^2| \geq \left(-\frac{p_t r(p_t)}{r'(p_t)}\right)D\eta^4$, then

$$|s_{t+1} - 1 - h(p_{t+1})\eta^2| \leq \left(1 + \frac{p_t r'(p_t)}{r(p_t)}\right)|s_t - 1 - h(p_t)\eta^2|. \tag{11}$$

For any $t \leq T$, we have $q_t < 1 - \frac{\delta}{2}$ so that if $|s_t - 1| = \mathcal{O}_{\delta,\ell}(\eta^2)$, then $r(p_t) \leq (1 + \mathcal{O}_{\delta,\ell}(\eta^2))q_t < 1 - \frac{\delta}{4}$ for small step size $\eta$. From Eq. (11) with $t = t_a^*$, we have either

$$|s_{t_a^*} - 1 - h(p_{t_a^*})\eta^2| < \left(-\frac{p_{t_a^*} r(p_{t_a^*})}{r'(p_{t_a^*})}\right)D\eta^4,$$

or

$$|s_{t_a^*+1} - 1 - h(p_{t_a^*+1})\eta^2| \leq \left(1 + \frac{\hat{r}(1 - \frac{\delta}{4})r'(\hat{r}(1 - \frac{\delta}{4}))}{(1 - \frac{\delta}{4})}\right)|s_{t_a^*} - 1 - h(p_{t_a^*})\eta^2|,$$

where we used Assumption 4.3 (ii) and $|p_t| > \hat{r}(1 - \frac{\delta}{4})$. In the later case, $|s_{t_a^*+1} - 1| = \mathcal{O}_{\delta,\ell}(\eta^2)$ continues to hold and we can again use Eq. (11) with $t = t_a^* + 1$. By repeating the analogous arguments, we can obtain the time step

$$t_a \leq t_a^* + \frac{\log\left(-\frac{D\eta^4}{r''(0)|s_{t_a^*} - 1 - h(p_{t_a^*})\eta^2|}\right)}{\log\left(1 + \frac{\hat{r}(1 - \frac{\delta}{4})r'(\hat{r}(1 - \frac{\delta}{4}))}{(1 - \frac{\delta}{4})}\right)} = \mathcal{O}_{\delta,\ell}(\log(\eta^{-1})),$$

which satisfies: either

$$|s_{t_a} - 1 - h(p_{t_a})\eta^2| < \left(-\frac{p_{t_a} r(p_{t_a})}{r'(p_{t_a})}\right)D\eta^4,$$

or

$$|s_{t_a} - 1 - h(p_{t_a})\eta^2| \leq \left(-\frac{1}{r''(0)}\right) D\eta^4 \leq \left(-\frac{p_{t_a} r(p_{t_a})}{r'(p_{t_a})}\right) D\eta^4 \leq \left(-\frac{4r(4)}{r'(4)}\right) D\eta^4,$$

where we used $|p_t| \leq 4$ from Lemma B.2 and $-\frac{zr(z)}{r'(z)} \geq -\frac{1}{r''(0)}$ for any $z$ by Assumption 4.3 (iii).

By Eq. (10), if $|s_t - 1 - h(p_t)\eta^2| \leq \left(-\frac{4r(4)}{r'(4)}\right) D\eta^4$ is satisfied for any time step $t$, then

$$|s_{t+1} - 1 - h(p_{t+1})\eta^2| \leq \left(1 + \frac{2p_t r'(p_t)}{r(p_t)}\right) \left(-\frac{4r(4)}{r'(4)}\right) D\eta^4 + D\eta^4 p_t^2 \leq \left(-\frac{4r(4)}{r'(4)}\right) D\eta^4,$$

by $|p_t| \leq 4$ from Lemma B.2 and Assumption 4.3 (iii).

Hence, by induction, we have the desired bound as following: for any $t \geq t_a$,

$$|s_t - 1 - h(p_t)\eta^2| \leq \left(-\frac{4r(4)}{r'(4)}\right) D\eta^4 = \mathcal{O}_{\delta,\ell}(\eta^4),$$

by $|p_t| \leq 4$ from Lemma B.2 and Assumption 4.3 (iii). $\qquad \square$

## B.3 Proof of Theorem 4.3

In this subsection, we prove Theorem 4.3. We start by proving Lemma B.9 which provides a useful property of $h$ defined in Theorem 4.2.

**Lemma B.9.** *Consider the function $h$ defined in Theorem 4.2, given by*

$$h(p) := \begin{cases} -\frac{1}{2}\left(\frac{pr(p)^3}{r'(p)} + p^2 r(p)^2\right) & \text{if } p \neq 0, \text{ and} \\ -\frac{1}{2r''(0)} & \text{if } p = 0. \end{cases}$$

*Then, $h$ is a positive, even, and bounded twice continuously differentiable function.*

*Proof.* It is clear that $h$ is even. We first prove that $h$ is positive. For any $p \neq 0$, it holds that

$$h(p) = -\frac{pr(p)^3}{2r'(p)}\left(1 + \frac{pr'(p)}{r(p)}\right) > 0,$$

since $\frac{pr(p)}{r'(p)} < 0$ and $1 + \frac{pr'(p)}{r(p)} = \frac{\ell''(p)}{r(p)} > 0$ by Assumption 4.2 and convexity of $\ell$. The function $h$ is continuous since $\lim_{p \to 0} h(p) = h(0)$. Continuous function on a compact domain is bounded, so $h$ is bounded on the closed interval $[-1, 1]$. We can rewrite $h$ as

$$h(p) = \frac{1}{2}p^2 r(p)^2 \left(-\frac{r(p)}{pr'(p)} - 1\right).$$

Note that $p^2 r(p)^2 = \ell'(p)^2 \leq 1$, and $\left(-\frac{r(p)}{pr'(p)} - 1\right)$ is positive, decreasing function on $p > 0$ by Assumption 4.3 (ii). Hence, $h$ is bounded on $[1, \infty)$. Since $h$ is even, $h$ is bounded on $(-\infty, 1]$. Therefore, $h$ is a bounded function on $\mathbb{R}$.

We finally prove that $h$ is twice continuously differentiable. Since $r$ is even and $C^4$ on $\mathbb{R}$, we can check that

$$h'(p) := \begin{cases} -\frac{1}{2}\left[\frac{r(p)^3(r'(p) - pr''(p))}{r'(p)^2} + pr(p)(5r(p) + 2pr'(p))\right] & \text{if } p \neq 0, \text{ and} \\ 0 & \text{if } p = 0. \end{cases}$$

Moreover, for any $p \neq 0$,

$$h''(p) = -\frac{1}{2}\left(\frac{2r(p)^2 r''(p)(pr''(p) - r'(p))}{r'(p)^3} - \frac{pr(p)^3 r^{(3)}(p)}{r'(p)^2} - \frac{3pr(p)^2 r''(p)}{r'(p)}\right)$$
$$- 4r(p)^2 - 7pr(p)r'(p) - p^2(r(p)r''(p) + r'(p)^2),$$

and

$$h''(0) = \frac{r^{(4)}(0)}{6r''(0)^2} - \frac{5}{2}.$$

Since $\lim_{p \to 0} h''(p) = h''(0)$, we can conclude that $h$ is a twice continuously differentiable function. $\qquad \square$

We now give the proof of Theorem 4.3, restated below for the sake of readability.

**Theorem 4.3** (EoS regime, Phase II). *Under the same settings as in Theorem 4.2, there exists a time step $t_b = \Omega((1-q_0)\eta^{-2})$ such that $q_{t_b} \leq 1$ and $q_t > 1$ for any $t > t_b$. Moreover, the GD iterates $(p_t, q_t)$ converge to the point $(0, q^*)$ such that*

$$q^* = 1 - \frac{\eta^2}{2r''(0)} + \mathcal{O}_{\delta,\ell}(\eta^4).$$

*Proof.* We first prove that there exists a time step $t_b \geq 0$ such that $q_{t_b} > 1$. Assume the contrary that $q_t \leq 1$ for all $t \geq 0$. Let $t_a$ be the time step obtained in Theorem 4.2. Then for any $t \geq t_a$, we have

$$r(p_t) = (1 - h(p_t)\eta^2 + \mathcal{O}_{\delta,\ell}(\eta^4))q_t \leq 1 - \frac{h(p_t)\eta^2}{2},$$

for small step size $\eta$. The function $g(p) := r(p) - 1 + \frac{h(p)\eta^2}{2}$ is even, continuous, and has the function value $g(0) = \frac{\eta^2}{4|r''(0)|} > 0$. Consequently, there exists a positive constant $\epsilon > 0$ such that $g(p) > 0$ for all $p \in (-\epsilon, \epsilon)$. Then, we have $|p_t| \geq \epsilon$ for all $t \geq t_a$, since $g(p_t) \leq 0$. Moreover, $s_t \geq \frac{3}{4}$ for any $t \geq t_a$ by Theorem 4.2 for small step size $\eta$. This implies that for any $t \geq t_a$,

$$\frac{q_t}{q_{t+1}} = 1 - \eta^2 p_t^2 r(p_t)^2 (2s_t - 1) \leq 1 - \frac{1}{2}\eta^2 \ell'(p_t)^2 \leq 1 - \frac{1}{2}\eta^2 \ell'(\epsilon)^2,$$

so $q_t$ grows exponentially, which results in the existence of a time step $t_b' \geq t_a$ such that $q_{t_b'} > 1$, a contradiction.

Therefore, there exists a time step $t_b$ such that $q_{t_b} \leq 1$ and $q_t > 1$ for any $t > t_b$, i.e., $q_t$ jumps across the value 1. This holds since the sequence $(q_t)$ is monotonically increasing. For any $t \leq t_b$, we have $q_{t+1} \leq q_t + \mathcal{O}(\eta^2)$ by Lemma B.2, and this implies that $t_b \geq \Omega((1-q_0)\eta^{-2})$, as desired.

Lastly, we prove the convergence of GD iterates $(p_t, q_t)$. Let $t > t_b$ be given. Then, $q_t \geq q_{t_b+1} > 1$ and it holds that

$$\left|\frac{p_{t+1}}{p_t}\right| = \frac{2r(p_t)}{q_t} - 1 - \eta^2 p_t^2 r(p_t)^2 \leq \frac{2}{q_{t_b+1}} - 1 < 1.$$

Hence, $|p_t|$ is exponentially decreasing for $t > t_b$. Therefore, $p_t$ converges to 0 as $t \to \infty$. Since the sequence $(q_t)_{t=0}^\infty$ is monotonically increasing and bounded (due to Theorem 4.2), it converges. Suppose that $(p_t, q_t)$ converges to the point $(0, q^*)$. By Theorem 4.2, we can conclude that

$$\left|q^* - 1 + \frac{\eta^2}{2r''(0)}\right| = \mathcal{O}_{\delta,\ell}(\eta^4),$$

which is the desired bound. $\square$

## B.4 Proof of Theorem 4.4

In this subsection, we prove Theorem 4.4. We first prove a useful lemma which bounds the Hessian of the function $(\boldsymbol{U}, \boldsymbol{v}) \mapsto \boldsymbol{v}^\top \boldsymbol{U} \boldsymbol{x}$, stated below.

**Lemma B.10.** *For any $\boldsymbol{\Theta} = (\boldsymbol{U}, \boldsymbol{v})$ with $\boldsymbol{U} \in \mathbb{R}^{m \times d}$, $\boldsymbol{v} \in \mathbb{R}^m$, and $\boldsymbol{x} \in \mathbb{R}^d$ with $\|\boldsymbol{x}\|_2 = 1$, the following equality holds:*

$$\left\|\nabla^2_{(\boldsymbol{U},\boldsymbol{v})}(\boldsymbol{v}^\top \boldsymbol{U} \boldsymbol{x})\right\|_2 \leq 1.$$

*Moreover, if $\lambda$ is an eigenvalue of $\nabla^2_{(\boldsymbol{U},\boldsymbol{v})}(\boldsymbol{v}^\top \boldsymbol{U} \boldsymbol{x})$, then $-\lambda$ is also an eigenvalue of $\nabla^2_{(\boldsymbol{U},\boldsymbol{v})}(\boldsymbol{v}^\top \boldsymbol{U} \boldsymbol{x})$.*

*Proof.* We first define the notations. We use the operator $\otimes$ to represent tensor product, or Kronecker product between matrices. For example, for any given two matrices $A = (a_{ij}) \in \mathbb{R}^{m \times n}$ and $B$, we define $A \otimes B$ by

$$A \otimes B = \begin{pmatrix} a_{11}B & \ldots & a_{1n}B \\ \vdots & \ddots & \vdots \\ a_{m1}B & \ldots & a_{mn}B \end{pmatrix}.$$

We use $\mathbf{0}_{m \times n}$ to denote a $m$ by $n$ matrix with all entries filled with zero, and $\mathbf{I}_n$ denotes $n$ by $n$ identity matrix. Now we provide the proof of the original problem.

Let $\mathbf{U} = (u_{ij}) \in \mathbb{R}^{m \times d}$, $\mathbf{v} = (v_i) \in \mathbb{R}^m$, and $\mathbf{x} = (x_j) \in \mathbb{R}^d$ be given. Then,

$$\mathbf{v}^\top \mathbf{U} \mathbf{x} = \sum_{i,j} v_i U_{ij} x_j.$$

We vectorize the parameter $\mathbf{\Theta} = (\mathbf{U}, \mathbf{v})$ by $(v_1, \ldots, v_m, U_{11}, \ldots, U_{m1}, \ldots, U_{1d}, \ldots, U_{md}) \in \mathbb{R}^{m+md}$. Then, we can represent the Hessian as

$$\nabla^2_{(\mathbf{U}, \mathbf{v})}(\mathbf{v}^\top \mathbf{U} \mathbf{x}) = \left( \begin{array}{c|c} 0 & \mathbf{x}^\top \\ \hline \mathbf{x} & \mathbf{0}_{d \times d} \end{array} \right) \otimes \mathbf{I}_m. \tag{12}$$

For any given $c \in \mathbb{R}$ and $\mathbf{y} \in \mathbb{R}^d$ with $c^2 + \|\mathbf{y}\|_2^2 = 1$, we have

$$\left\| \left( \begin{array}{c|c} 0 & \mathbf{x}^\top \\ \hline \mathbf{x} & \mathbf{0}_{d \times d} \end{array} \right) \left( \begin{array}{c} c \\ \mathbf{y} \end{array} \right) \right\|_2 = \left\| \left( \begin{array}{c} \mathbf{x}^\top \mathbf{y} \\ c\, \mathbf{x} \end{array} \right) \right\|_2 \leq \|\mathbf{x}\|_2 = 1.$$

Hence, by definition of matrix operator norm, we have

$$\left\| \left( \begin{array}{c|c} 0 & \mathbf{x}^\top \\ \hline \mathbf{x} & \mathbf{0}_{d \times d} \end{array} \right) \right\|_2 \leq 1.$$

Therefore, we can conclude that

$$\left\| \nabla^2_{(\mathbf{U}, \mathbf{v})}(\mathbf{v}^\top \mathbf{U} \mathbf{x}) \right\|_2 = \left\| \left( \begin{array}{c|c} 0 & \mathbf{x}^\top \\ \hline \mathbf{x} & \mathbf{0}_{d \times d} \end{array} \right) \otimes \mathbf{I}_m \right\|_2 = \left\| \left( \begin{array}{c|c} 0 & \mathbf{x}^\top \\ \hline \mathbf{x} & \mathbf{0}_{d \times d} \end{array} \right) \right\|_2 \leq 1.$$

Now suppose that $\lambda$ is an eigenvalue of $\nabla^2_{(\mathbf{U}, \mathbf{v})}(\mathbf{v}^\top \mathbf{U} \mathbf{x})$. We note that for any given matrices $\mathbf{A}$ and $\mathbf{B}$, if $\lambda_a$ is an eigenvalue of $\mathbf{A}$ with the corresponding eigenvector $\mathbf{u}_a$ and $\lambda_b$ is an eigenvalue of $\mathbf{B}$ with the corresponding eigenvector $\mathbf{u}_b$, then $\lambda_a \lambda_b$ is an eigenvalue of $\mathbf{A} \otimes \mathbf{B}$ with the corresponding eigenvector $\mathbf{u}_a \otimes \mathbf{u}_b$. Moreover, any eigenvalue of $\mathbf{A} \otimes \mathbf{B}$ arises as such a product of eigenvalues of $\mathbf{A}$ and $\mathbf{B}$. Hence, using Eq. (12), we have

$$\lambda \text{ is an eigenvalue of the matrix } \left( \begin{array}{c|c} 0 & \mathbf{x}^\top \\ \hline \mathbf{x} & \mathbf{0}_{d \times d} \end{array} \right).$$

We denote the corresponding eigenvector by $(c, \mathbf{y}^\top)^\top$ where $c \in \mathbb{R}$ and $\mathbf{y} \in \mathbb{R}^d$, i.e., it holds that

$$\left( \begin{array}{c|c} 0 & \mathbf{x}^\top \\ \hline \mathbf{x} & \mathbf{0}_{d \times d} \end{array} \right) \left( \begin{array}{c} c \\ \mathbf{y} \end{array} \right) = \left( \begin{array}{c} \mathbf{x}^\top \mathbf{y} \\ c\, \mathbf{x} \end{array} \right) = \lambda \left( \begin{array}{c} c \\ \mathbf{y} \end{array} \right).$$

Consequently, we have

$$\left( \begin{array}{c|c} 0 & \mathbf{x}^\top \\ \hline \mathbf{x} & \mathbf{0}_{d \times d} \end{array} \right) \left( \begin{array}{c} -c \\ \mathbf{y} \end{array} \right) = \left( \begin{array}{c} \mathbf{x}^\top \mathbf{y} \\ -c\, \mathbf{x} \end{array} \right) = -\lambda \left( \begin{array}{c} -c \\ \mathbf{y} \end{array} \right),$$

and this implies that

$$-\lambda \text{ is an eigenvalue of the matrix } \left( \begin{array}{c|c} 0 & \mathbf{x}^\top \\ \hline \mathbf{x} & \mathbf{0}_{d \times d} \end{array} \right).$$

Therefore, by Eq. (12), $-\lambda$ is an eigenvalue of $\nabla^2_{(\mathbf{U}, \mathbf{v})}(\mathbf{v}^\top \mathbf{U} \mathbf{x})$. $\qquad \square$

Using Lemma B.10, we prove an important bound on the sharpness value provided by the Proposition B.11 stated below.

**Proposition B.11.** *For any $\mathbf{\Theta} = (\mathbf{U}, \mathbf{v})$ with $\mathbf{U} \in \mathbb{R}^{m \times d}$, $\mathbf{v} \in \mathbb{R}^m$, and $\mathbf{x} \in \mathbb{R}^d$ with $\|\mathbf{x}\|_2 = 1$, the following bound holds:*

$$\left| \lambda_{\max}(\mathbf{\Theta}) - \ell''(\mathbf{v}^\top \mathbf{U} \mathbf{x}) \left( \|\mathbf{U} \mathbf{x}\|_2^2 + \|\mathbf{v}\|_2^2 \right) \right| \leq 1$$

*Proof.* The loss Hessian at $\boldsymbol{\Theta} = (\boldsymbol{U}, \boldsymbol{v})$ can be characterized as:

$$\nabla_{\boldsymbol{\Theta}}^2 \mathcal{L}(\boldsymbol{\Theta}) = \ell''(\boldsymbol{v}^\top \boldsymbol{U} \boldsymbol{x}) \left(\nabla_{\boldsymbol{\Theta}}(\boldsymbol{v}^\top \boldsymbol{U} \boldsymbol{x})\right)^{\otimes 2} + \ell'(\boldsymbol{v}^\top \boldsymbol{U} \boldsymbol{x}) \nabla_{\boldsymbol{\Theta}}^2 (\boldsymbol{v}^\top \boldsymbol{U} \boldsymbol{x}). \tag{13}$$

We first prove that $\lambda_{\max}(\nabla_{\boldsymbol{\Theta}}^2 \mathcal{L}(\boldsymbol{\Theta})) = \left\|\nabla_{\boldsymbol{\Theta}}^2 \mathcal{L}(\boldsymbol{\Theta})\right\|_2$. Note that the largest absolute value of the eigenvalue of a symmetric matrix equals to its spectral norm. Hence, $\left\|\nabla_{\boldsymbol{\Theta}}^2 \mathcal{L}(\boldsymbol{\Theta})\right\|_2 = \max\{\lambda_{\max}(\nabla_{\boldsymbol{\Theta}}^2 \mathcal{L}(\boldsymbol{\Theta})), -\lambda_{\min}(\nabla_{\boldsymbol{\Theta}}^2 \mathcal{L}(\boldsymbol{\Theta}))\}$, so it suffices to prove that $\lambda_{\max}(\nabla_{\boldsymbol{\Theta}}^2 \mathcal{L}(\boldsymbol{\Theta})) \geq -\lambda_{\min}(\nabla_{\boldsymbol{\Theta}}^2 \mathcal{L}(\boldsymbol{\Theta}))$. Let $\boldsymbol{w}$ denote the eigenvector of $\nabla_{\boldsymbol{\Theta}}^2 \mathcal{L}(\boldsymbol{\Theta})$ corresponding to the smallest eigenvalue $\lambda_{\min}(\nabla_{\boldsymbol{\Theta}}^2 \mathcal{L}(\boldsymbol{\Theta}))$ with $\|\boldsymbol{w}\|_2 = 1$. Then, using Eq. (13), we have

$$\begin{aligned}
\lambda_{\min}(\nabla_{\boldsymbol{\Theta}}^2 \mathcal{L}(\boldsymbol{\Theta})) = \boldsymbol{w}^\top \nabla_{\boldsymbol{\Theta}}^2 \mathcal{L}(\boldsymbol{\Theta}) \boldsymbol{w} &= \ell''(\boldsymbol{v}^\top \boldsymbol{U} \boldsymbol{x}) \boldsymbol{w}^\top \left(\nabla_{\boldsymbol{\Theta}}(\boldsymbol{v}^\top \boldsymbol{U} \boldsymbol{x})\right)^{\otimes 2} \boldsymbol{w} + \ell'(\boldsymbol{v}^\top \boldsymbol{U} \boldsymbol{x}) \boldsymbol{w}^\top \nabla_{\boldsymbol{\Theta}}^2 (\boldsymbol{v}^\top \boldsymbol{U} \boldsymbol{x}) \boldsymbol{w} \\
&\geq \ell'(\boldsymbol{v}^\top \boldsymbol{U} \boldsymbol{x}) \boldsymbol{w}^\top \nabla_{\boldsymbol{\Theta}}^2 (\boldsymbol{v}^\top \boldsymbol{U} \boldsymbol{x}) \boldsymbol{w} \\
&\geq -|\ell'(\boldsymbol{v}^\top \boldsymbol{U} \boldsymbol{x})| \|\nabla_{\boldsymbol{\Theta}}^2 (\boldsymbol{v}^\top \boldsymbol{U} \boldsymbol{x})\|_2,
\end{aligned}$$

where we used Lemma B.10 to obtain the last inequality. Note that the matrix $\ell''(\boldsymbol{v}^\top \boldsymbol{U} \boldsymbol{x}) \left(\nabla_{\boldsymbol{\Theta}}(\boldsymbol{v}^\top \boldsymbol{U} \boldsymbol{x})\right)^{\otimes 2}$ is PSD, so that $\lambda_{\max}(\nabla_{\boldsymbol{\Theta}}^2 \mathcal{L}(\boldsymbol{\Theta})) \geq \lambda_{\max}(\ell'(\boldsymbol{v}^\top \boldsymbol{U} \boldsymbol{x}) \nabla_{\boldsymbol{\Theta}}^2 (\boldsymbol{v}^\top \boldsymbol{U} \boldsymbol{x})) = |\ell'(\boldsymbol{v}^\top \boldsymbol{U} \boldsymbol{x})| \|\nabla_{\boldsymbol{\Theta}}^2 (\boldsymbol{v}^\top \boldsymbol{U} \boldsymbol{x})\|_2 \geq -\lambda_{\min}(\nabla_{\boldsymbol{\Theta}}^2 \mathcal{L}(\boldsymbol{\Theta}))$. Therefore, $\lambda_{\max}(\nabla_{\boldsymbol{\Theta}}^2 \mathcal{L}(\boldsymbol{\Theta})) = \left\|\nabla_{\boldsymbol{\Theta}}^2 \mathcal{L}(\boldsymbol{\Theta})\right\|_2$.

Now, we have the following triangle inequality:

$$\begin{aligned}
\left|\lambda_{\max}(\boldsymbol{\Theta}) - \ell''(\boldsymbol{v}^\top \boldsymbol{U} \boldsymbol{x}) \left(\|\boldsymbol{U} \boldsymbol{x}\|_2^2 + \|\boldsymbol{v}\|_2^2\right)\right| &= \left|\left\|\nabla_{\boldsymbol{\Theta}}^2 \mathcal{L}(\boldsymbol{\Theta})\right\|_2 - \left\|\ell''(\boldsymbol{v}^\top \boldsymbol{U} \boldsymbol{x}) \left(\nabla_{\boldsymbol{\Theta}}(\boldsymbol{v}^\top \boldsymbol{U} \boldsymbol{x})\right)^{\otimes 2}\right\|_2\right| \\
&\leq \left\|\ell'(\boldsymbol{v}^\top \boldsymbol{U} \boldsymbol{x}) \nabla_{\boldsymbol{\Theta}}^2 (\boldsymbol{v}^\top \boldsymbol{U} \boldsymbol{x})\right\|_2 \\
&= \left|\ell'(\boldsymbol{v}^\top \boldsymbol{U} \boldsymbol{x})\right| \left\|\nabla_{(\boldsymbol{U}, \boldsymbol{v})}^2 (\boldsymbol{v}^\top \boldsymbol{U} \boldsymbol{x})\right\|_2 \\
&\leq 1,
\end{aligned}$$

where the last inequality holds by Lemma B.10 and 1-Lipschitzness of $\ell$. $\qquad\square$

We now give the proof of Theorem 4.4, restated below for the sake of readability.

**Theorem 4.4** (progressive sharpening). *Under the same setting as in Theorem 4.2, let $t_a$ denote the obtained iteration. Define the function $\tilde{\lambda} : \mathbb{R}_{>0} \to \mathbb{R}$ given by*

$$\tilde{\lambda}(q) := \begin{cases} \left(1 + \frac{\hat{r}(q) r'(\hat{r}(q))}{q}\right) \frac{2}{\eta} & \text{if } q \leq 1, \text{ and} \\ \frac{2}{\eta} & \text{otherwise.} \end{cases}$$

*Then, the sequence $\left(\tilde{\lambda}(q_t)\right)_{t=0}^{\infty}$ is monotonically increasing. Moreover, for any $t \geq t_a$, the sharpness at GD iterate $\boldsymbol{\Theta}_t$ closely follows the sequence $\left(\tilde{\lambda}(q_t)\right)_{t=0}^{\infty}$ by satisfying*

$$\left|\lambda_{\max}(\boldsymbol{\Theta}_t) - \tilde{\lambda}(q_t)\right| \leq 1 + \mathcal{O}_\ell(\eta).$$

*Proof.* By Proposition B.11, we can bound the sharpness $\lambda_{\max}(\boldsymbol{\Theta}_t)$ at time step $t$ by

$$\left|\lambda_{\max}(\boldsymbol{\Theta}_t) - \frac{2\ell''(p_t)}{\eta q_t}\right| \leq 1.$$

Since $\ell''(z) = r(z) + z r'(z)$, we can rewrite as following:

$$\left|\lambda_{\max}(\boldsymbol{\Theta}_t) - \left(s_t^{-1} + \frac{p_t r'(p_t)}{q_t}\right) \frac{2}{\eta}\right| \leq 1. \tag{14}$$

By Theorem 4.2 and since $h$ is a bounded function by Lemma B.9, we have $s_t = 1 + \mathcal{O}_\ell(\eta^2)$ for any $t \geq t_a$. Consequently, $|s_t^{-1} - 1| = \mathcal{O}_\ell(\eta^2)$ and $|r(p_t) - q_t| = \mathcal{O}_\ell(\eta^2)$. Moreover, for any $0 < q < 1$,

$$\begin{aligned}
\frac{d}{dq}\left(\frac{\hat{r}(q) r'(\hat{r}(q))}{q}\right) &= \frac{\hat{r}'(q)(r'(\hat{r}(q)) + \hat{r}(q) r''(\hat{r}(q)))}{q} - \frac{\hat{r}(q) r'(\hat{r}(q))}{q^2} \\
&= \frac{1}{q}\left(1 + \frac{\hat{r}(q) r''(\hat{r}(q))}{r'(\hat{r}(q))}\right) - \frac{\hat{r}(q) r'(\hat{r}(q))}{q^2},
\end{aligned}$$

so that

$$\lim_{q \to 1^-} \left( \frac{d}{dq} \left( \frac{\hat{r}(q) r'(\hat{r}(q))}{q} \right) \right) = \lim_{p \to 0^+} \left( 1 + \frac{p r''(p)}{r'(p)} \right) = 2.$$

Therefore, $\frac{d}{dq} \left( \frac{\hat{r}(q) r'(\hat{r}(q))}{q} \right)$ is bounded on $[\frac{1}{4}, 1)$ and Taylor's theorem gives

$$\left| \frac{p_t r'(p_t)}{r(p_t)} - \frac{\hat{r}(q_t) r'(\hat{r}(q_t))}{q_t} \right| = \mathcal{O}_\ell(|r(p_t) - q_t|) = \mathcal{O}_\ell(\eta^2),$$

for any time step $t$ with $q_t < 1$. Hence, if $q_t < 1$, we have the following bound:

$$\left| \tilde{\lambda}(q_t) - \left( s_t^{-1} + \frac{p_t r'(p_t)}{q_t} \right) \frac{2}{\eta} \right| \le \left| 1 - s_t^{-1} + \frac{\hat{r}(q_t) r'(\hat{r}(q_t))}{q_t} - \frac{p_t r'(p_t)}{r(p_t)} \right| \frac{2}{\eta} + \mathcal{O}_\ell(\eta) = \mathcal{O}_\ell(\eta),$$
(15)

where we used $\frac{p_t r'(p_t)}{q_t} = \frac{p_t r'(p_t)}{r(p_t)}(1 + \mathcal{O}_\ell(\eta^2)) = \frac{p_t r'(p_t)}{r(p_t)} + \mathcal{O}_\ell(\eta^2)$, since $1 + \frac{p_t r'(p_t)}{r(p_t)} = \frac{\ell''(p_t)}{r(p_t)} > 0$ implies $|p_t r'(p_t)| \le r(p_t) \le 1$. Now let $t$ be any given time step with $q_t \ge 1$. Then, $r(p_t) = 1 - \mathcal{O}_\ell(\eta^2)$, and since $r(z) = 1 + r''(0)z^2 + \mathcal{O}_\ell(z^4)$ for small $z$, we have $|p_t| = \mathcal{O}_\ell(\eta)$. Hence,

$$\left| \tilde{\lambda}(q_t) - \left( s_t^{-1} + \frac{p_t r'(p_t)}{q_t} \right) \frac{2}{\eta} \right| \le \left| 1 - s_t^{-1} - \frac{p_t r'(p_t)}{r(p_t)} \right| \frac{2}{\eta} + \mathcal{O}_\ell(\eta) = \mathcal{O}_\ell(\eta), \qquad (16)$$

for any $t$ with $q_t \ge 1$. By Eqs. (14), (15), and (16), we can conclude that for any $t \ge t_a$, we have

$$\left| \lambda_{\max}(\boldsymbol{\Theta}_t) - \tilde{\lambda}(q_t) \right| \le 1 + \mathcal{O}_\ell(\eta).$$

Finally, we can easily check that the sequence $(\tilde{\lambda}(q_t))_{t=0}^\infty$ is monotonically increasing, since $z \mapsto \frac{z r'(z)}{r(z)}$ is a decreasing function by Assumption 4.3 (ii) and the sequence $(q_t)$ is monotonically increasing. $\qquad \square$

# C Proofs for the Single-neuron Nonlinear Network

## C.1 Formal statements of Theorem 5.1

In this subsection, we provide the formal statements of Theorem 5.1. We study the GD dynamics on a two-dimensional function $\mathcal{L}(x, y) := \frac{1}{2}(\phi(x)y)^2$, where $x$, $y$ are scalars and $\phi$ is a nonlinear activation satisfying Assumption 5.1. We consider the reparameterization given by Definition 5.2, which is $(p, q) := \left( x, \frac{2}{\eta y^2} \right)$.

We emphasize that the results we present in this subsection closely mirror those of the Section 4. In particular,

- Assumption C.1 mirrors Assumption 4.2,
- Assumption C.2 mirrors Assumption 4.3,
- (gradient flow regime) Theorem C.1 mirrors Theorem 4.1.
- (EoS regime, Phase I) Theorem C.2 mirrors Theorem 4.2,
- (EoS regime, Phase II) Theorem C.3 mirrors Theorem 4.3, and
- (progressive sharpening) Theorem C.4 mirrors Theorem 4.4.

The proof strategies are also similar. This is mainly because the 1-step update rule Eq. (7) resembles Eq. (6) for small step size $\eta$. We now present our rigorous results contained in Theorem 5.1.

Inspired by Lemma 2.1, we have an additional assumption on $\phi$ as below.

**Assumption C.1.** *Let $r$ be a function defined by $r(z) := \frac{\phi(z)\phi'(z)}{z}$ for $z \neq 0$ and $r(0) := 1$. The function $r$ satisfies Assumption 2.4.*

In contrast to the function $r$ defined in Section 4, the expression $1 + \frac{pr'(p)}{r(p)}$ can be negative, which implies that the constant $c$ defined in Lemma 2.1 is positive. As a result, the dynamics of $p_t$ may exhibit a period-$4$ (or higher) oscillation or even chaotic behavior (as illustrated in Figure 1b).

We first state our results on the gradient flow regime.

**Theorem C.1** (gradient flow regime). *Let $\eta \in (0, \frac{r(1)}{2(r(1)+2)})$ be a fixed step size and $\phi$ be a sigmoidal function satisfying Assumptions 5.1 and C.1. Suppose that the initialization $(p_0, q_0)$ satisfies $|p_0| \leq 1$ and $q_0 \in \left( \frac{1}{1-2\eta}, \frac{r(1)}{4\eta} \right)$. Consider the reparameterized GD trajectory characterized in Eq. (7). Then, the GD iterations $(p_t, q_t)$ converge to the point $(0, q^*)$ such that*

$$q_0 \leq q^* \leq \exp\left( 2\eta \left[ \min\left\{ \frac{2(q_0 - 1)}{q_0}, \frac{r(1)}{q_0} \right\} \right]^{-1} \right) q_0 \leq 2q_0.$$

Theorem C.1 implies that in gradient flow regime, GD with initialization $(x_0, y_0)$ and step size $\eta$ converges to $(0, y^*)$ which has the sharpness bounded by:

$$\left( 1 - 2\eta \left[ \min\left\{ \frac{2(q_0 - 1)}{q_0}, \frac{r(1)}{q_0} \right\} \right]^{-1} \right) y_0^2 \leq \lambda_{\max}(\nabla^2 \mathcal{L}(0, y^*)) \leq y_0^2.$$

Now we provide our results on the EoS regime with an additional assumption below.

**Assumption C.2.** *Let $r$ be a function defined in Assumption C.1. Then $r$ is $C^4$ on $\mathbb{R}$ and satisfies:*

(i) $z \mapsto \frac{r'(z)}{r(z)^2}$ *is decreasing on $\mathbb{R}$,*

(ii) $z \mapsto \frac{zr'(z)}{r(z)}$ *is decreasing on $z > 0$ and increasing on $z < 0$,*

(iii) $z \to \frac{zr(z)}{r'(z)}$ *is decreasing on $z > 0$ and increasing on $z < 0$, and*

(iv) $\hat{r}(\frac{1}{2})r'(\hat{r}(\frac{1}{2})) > -\frac{1}{2}$.

Note that the function $r$ that arise from the activation $\phi = \tanh$ satisfies Assumptions 5.1, C.1, and C.2.

**Theorem C.2** (EoS regime, Phase I). *Let $\eta > 0$ be a small enough constant and $\phi$ be an activation function satisfying Assumptions 5.1, C.1, and C.2. Let $z_0 := \sup_z \{ \frac{zr'(z)}{r(z)} \geq -\frac{1}{2} \}$, $z_1 := \sup_z \{ \frac{zr'(z)}{r(z)} \geq -1 \}$, and $c_0 := \max\{r(z_0), r(z_1) + \frac{1}{2}\} \in (\frac{1}{2}, 1)$. Let $\delta \in (0, 1 - c_0)$ be any given constant. Suppose that the initialization $(p_0, q_0)$ satisfies $|p_0| \leq 1$ and $q_0 \in (c_0, 1 - \delta)$. Consider the reparameterized GD trajectory characterized in Eq. (7). We assume that for all $t \geq 0$ such that $q_t < 1$, we have $p_t \neq 0$. Then, there exists a time step $t_a = \mathcal{O}_{\delta,\phi}(\log(\eta^{-1}))$ such that for any $t \geq t_a$,*

$$\frac{q_t}{r(p_t)} = 1 + h(p_t)\eta + \mathcal{O}_{\delta,\phi}(\eta^2)$$

*where $h : \mathbb{R} \to \mathbb{R}$ is a function defined as*

$$h(p) := \begin{cases} -\frac{\phi(p)^2 r(p)}{pr'(p)} & \text{if } p \neq 0, \text{ and} \\ -\frac{1}{r''(0)} & \text{if } p = 0. \end{cases}$$

The main difference between Theorem C.2 and Theorem 4.2 is the error term which is $\mathcal{O}(\eta^2)$ in the former and $\mathcal{O}(\eta^4)$ in the latter. This is because the 1-step update rule of $q_t$ in Theorem C.2 is given by $q_{t+1} = (1 + \mathcal{O}(\eta))q_t$, while in Theorem 4.2 we have $q_{t+1} = (1 + \mathcal{O}(\eta^2))q_t$.

**Theorem C.3** (EoS regime, Phase II). *Under the same settings as in Theorem C.2, there exists a time step $t_b = \Omega((1 - q_0)\eta^{-1})$, such that $q_{t_b} \leq 1$ and $q_t > 1$ for any $t > t_b$. Moreover, the GD iterates $(p_t, q_t)$ converge to the point $(0, q^*)$ such that*

$$q^* = 1 - \frac{\eta}{r''(0)} + \mathcal{O}_{\delta,\phi}(\eta^2).$$

Theorem C.3 implies that in the EoS regime, GD with step size $\eta$ converges to $(0, y^*)$ which has the sharpness approximated as:

$$\lambda_{\max}(\nabla^2 \mathcal{L}(0, y^*)) = \frac{2}{\eta} - \frac{2}{|r''(0)|} + \mathcal{O}_{\delta,\phi}(\eta).$$

Theorem C.4 proves that progressive sharpening (i.e., sharpness increases) occurs during Phase II.

**Theorem C.4** (progressive sharpening). *Under the same setting as in Theorem C.2, let $t_a$ denote the obtained time step. Define the function $\tilde{\lambda} : \mathbb{R}_{>0} \to \mathbb{R}$ given by*

$$\tilde{\lambda}(q) := \begin{cases} \left(1 + \frac{\hat{r}(q)r'(\hat{r}(q))}{q}\right)\frac{2}{\eta} & \text{if } q \leq 1, \text{ and} \\ \frac{2}{\eta} & \text{otherwise.} \end{cases}$$

*Then, the sequence $\left(\tilde{\lambda}(q_t)\right)_{t=0}^{\infty}$ is monotonically increasing. For any $t \geq t_a$, the sharpness at GD iterate $(x_t, y_t)$ closely follows the sequence $\left(\tilde{\lambda}(q_t)\right)_{t=0}^{\infty}$ satisfying the following:*

$$\lambda_{\max}(\nabla^2 \mathcal{L}(x_t, y_t)) = \tilde{\lambda}(q_t) + \mathcal{O}_{\phi}(1).$$

In Figure 1b, we conduct numerical experiments on single neuron model with $\tanh$-activation, demonstrating that $\tilde{\lambda}(q_t)$ provides a close approximation of the sharpness.

## C.2   Proof of Theorem C.1

We give the proof of Theorem C.1, restated below for the sake of readability.

**Theorem C.1** (gradient flow regime). *Let $\eta \in (0, \frac{r(1)}{2(r(1)+2)})$ be a fixed step size and $\phi$ be a sigmoidal function satisfying Assumptions 5.1 and C.1. Suppose that the initialization $(p_0, q_0)$ satisfies $|p_0| \leq 1$*

*and* $q_0 \in \left( \frac{1}{1-2\eta}, \frac{r(1)}{4\eta} \right)$. *Consider the reparameterized GD trajectory characterized in Eq.* (7)*. Then, the GD iterations* $(p_t, q_t)$ *converge to the point* $(0, q^*)$ *such that*

$$q_0 \leq q^* \leq \exp\left( 2\eta \left[ \min\left\{ \frac{2(q_0 - 1)}{q_0}, \frac{r(1)}{q_0} \right\} \right]^{-1} \right) q_0 \leq 2q_0.$$

Theorem C.1 directly follows from Proposition C.5 stated below.

**Proposition C.5.** *Suppose that* $\eta \in \left( 0, \frac{r(1)}{2(r(1)+2)} \right)$, $|p_0| \leq 1$ *and* $q_0 \in \left( \frac{1}{1-2\eta}, \frac{r(1)}{4\eta} \right)$. *Then for any* $t \geq 0$, *we have*

$$|p_t| \leq \left[ 1 - \min\left\{ \frac{2(q_0 - 1)}{q_0}, \frac{r(1)}{q_0} \right\} \right]^t \leq 1,$$

*and*

$$q_0 \leq q_t \leq \exp\left( 2\eta \left[ \min\left\{ \frac{2(q_0 - 1)}{q_0}, \frac{r(1)}{q_0} \right\} \right]^{-1} \right) q_0 \leq 2q_0.$$

*Proof.* We give the proof by induction; namely, if

$$|p_t| \leq \left[ 1 - \min\left\{ \frac{2(q_0 - 1)}{q_0}, \frac{r(1)}{q_0} \right\} \right]^t, \quad q_0 \leq q_t \leq \exp\left( 2\eta \left[ \min\left\{ \frac{2(q_0 - 1)}{q_0}, \frac{r(1)}{q_0} \right\} \right]^{-1} \right) q_0 \leq 2q_0$$

are satisfied for time steps $0 \leq t \leq k$ for some $k$, then the inequalities are also satisfied for the next time step $k + 1$.

For the base case, the inequalities are satisfied for $t = 0$ by assumptions. For the induction step, we assume that the inequalities hold for any $0 \leq t \leq k$. We will prove that the inequalities are also satisfied for $t = k + 1$.

By induction assumptions, we have $r(1) \leq r(p_k) \leq 1$ and $q_0 \leq q_k \leq 2q_0$. From Eq. (7), we get

$$\left| \frac{p_{k+1}}{p_k} \right| = \left| 1 - \frac{2r(p_k)}{q_k} \right| \leq \max\left\{ 1 - \frac{2r(1)}{2q_0}, -1 + \frac{2}{q_0} \right\} = 1 - \frac{\min\{2(q_0 - 1), r(1)\}}{q_0}.$$

Due to the induction assumption, we obtain the desired bound on $|p_{k+1}|$ as following:

$$|p_{k+1}| \leq \left[ 1 - \min\left\{ \frac{2(q_0 - 1)}{q_0}, \frac{r(1)}{q_0} \right\} \right]^{k+1}.$$

Moreover, for any $0 \leq t \leq k$, by Eq. (7) we have

$$1 - 2\eta p_t^2 \leq (1 - \eta p_t^2)^2 \leq \frac{q_t}{q_{t+1}} = (1 - \eta\phi(p_t)^2)^2 \leq 1,$$

where the second inequality comes from the fact that $\phi$ is 1-Lipschitz and $\phi(0) = 0$ (Assumption 5.1). Hence, we have $q_{k+1} \geq q_k \geq q_0$. Note that $\frac{q_t}{q_{t+1}} \in [\frac{1}{2}, 1]$ for small $\eta$. Consequently, we have

$$\left| \log\left( \frac{q_0}{q_{k+1}} \right) \right| \leq \sum_{t=0}^{k} \left| \log\left( \frac{q_t}{q_{t+1}} \right) \right| \leq 2 \sum_{t=0}^{k} \left| \frac{q_t}{q_{t+1}} - 1 \right|$$

$$\leq 2\eta \sum_{t=0}^{k} p_t^2$$

$$\leq 2\eta \sum_{t=0}^{k} \left[ 1 - \min\left\{ \frac{2(q_0 - 1)}{q_0}, \frac{r(1)}{q_0} \right\} \right]^{2t}$$

$$\leq 2\eta \left[ \min\left\{ \frac{2(q_0 - 1)}{q_0}, \frac{r(1)}{q_0} \right\} \right]^{-1},$$

where the second inequality holds since $|\log(1 + z)| \leq 2|z|$ if $|z| \leq \frac{1}{2}$. Therefore, we obtain the desired bound on $q_{k+1}$ as following:

$$q_0 \leq q_{k+1} \leq \exp\left(2\eta\left[\min\left\{\frac{2(q_0 - 1)}{q_0}, \frac{r(1)}{q_0}\right\}\right]^{-1}\right)q_0.$$

Since $q_0 \geq \frac{1}{1-2\eta}$ and $q_0 \leq \frac{r(1)}{4\eta}$, we have

$$\min\left\{\frac{2(q_0 - 1)}{q_0}, \frac{r(1)}{q_0}\right\} \geq 4\eta.$$

This implies that $q_{k+1} \leq \exp(\frac{1}{2})q_0 \leq 2q_0$, as desired. $\qquad\square$

## C.3  Proof of Theorem C.2

In this subsection, we prove Theorem C.2. We use the following notation:

$$s_t := \frac{q_t}{r(p_t)}.$$

All the lemmas in this subsection are stated in the context of Theorem C.2. The proof structure resembles that of Theorem 4.2. We informally summarize the lemmas used in the proof of Theorem C.2. Lemma C.6 proves that $p_t$ is bounded by a constant and $q_t$ increases monotonically with the increment bounded by $\mathcal{O}(\eta)$. Lemma C.7 states that in the early phase of training, there exists a time step $t_0$ where $s_{t_0}$ becomes smaller or equal to $\frac{2}{2-r(1)}$, which is smaller than 2. Lemma C.9 demonstrates that if $s_t$ is smaller than 2 and $|p_t| \geq \hat{r}(1 - \frac{\delta}{4})$, then $|s_t - 1|$ decreases exponentially. For the case where $|p_t| < \hat{r}(1 - \frac{\delta}{4})$, Lemma C.10 proves that $|p_t|$ increases at an exponential rate. Moreover, Lemma C.8 shows that if $s_t < 1$ at some time step, then $s_{t+1}$ is upper bounded by $1 + \mathcal{O}(\eta)$. Combining these findings, Proposition C.11 establishes that in the early phase of training, there exists a time step $t_a^*$ such that $s_{t_a^*} = 1 + \mathcal{O}_{\delta,\phi}(\eta)$. Lastly, Lemma C.12 demonstrates that if $s_t = 1 + \mathcal{O}_{\delta,\phi}(\eta)$, then $|s_t - 1 - h(p_t)\eta|$ decreases exponentially.

**Lemma C.6.** *Suppose that the initialization* $(p_0, q_0)$ *satisfies* $|p_0| \leq 1$ *and* $q_0 \in (c_0, 1 - \delta)$. *Then for any* $t \geq 0$ *such that* $q_t \leq 1$, *it holds that*

$$|p_t| \leq 4, \text{ and } q_t \leq q_{t+1} \leq (1 + \mathcal{O}(\eta))q_t.$$

*Proof.* We prove by induction. We assume that for some $t \geq 0$, it holds that $|p_t| \leq 4$ and $\frac{1}{2} \leq q_t \leq 1$. We will prove that $|p_{t+1}| \leq 4$ and $\frac{1}{2} \leq q_t \leq q_{t+1} \leq (1 + \mathcal{O}(\eta))q_t$. For the base case, $|p_0| \leq 1 \leq 4$ and $\frac{1}{2} \leq c_0 < q_t \leq 1$ holds by the assumptions on the initialization. Now suppose that for some $t \geq 0$, it holds that $|p_t| \leq 4$ and $\frac{1}{2} \leq q_t \leq 1$. By Eq. (7),

$$|p_{t+1}| = \left|p_t - \frac{2\phi(p_t)\phi'(p_t)}{q_t}\right| \leq \max\left\{|p_t|, \frac{2}{q_t}\right\} \leq 4.$$

where we used Assumption 5.1 to bound $|\phi(p_t)\phi'(p_t)| \leq 1$. Moreover,

$$1 - 2\eta \leq (1 - \eta)^2 \leq \frac{q_t}{q_{t+1}} = (1 - \eta\phi(p_t)^2)^2 \leq 1,$$

since $|\phi|$ is bounded by 1. Hence, $q_t \leq q_{t+1} \leq (1 + \mathcal{O}(\eta))q_t$, as desired. $\qquad\square$

Lemma C.6 implies that $p_t$ is bounded by a constant throughout the iterations, and $q_t$ monotonically increases slowly, where the increment for each step is $\mathcal{O}(\eta)$. Hence, there exists a time step $T = \Omega(\delta\eta^{-1}) = \Omega_\delta(\eta^{-1})$ such that for any $t \leq T$, it holds that $q_t \leq 1 - \frac{\delta}{2}$. Through out this subsection, we focus on these $T$ early time steps. Note that for all $0 \leq t \leq T$, it holds that $q_t \in (c_0, 1 - \frac{\delta}{2})$.

**Lemma C.7.** *There exists a time step* $t_0 = \mathcal{O}_{\delta,\phi}(1)$ *such that* $s_{t_0} \leq \frac{2}{2-r(1)}$.

*Proof.* We start by proving the following statement: for any $0 \le t \le T$, if $\frac{2}{2-r(1)} < s_t < 2r(1)^{-1}$, then $s_{t+1} < 2r(1)^{-1}$ and $|p_{t+1}| \le (1 - r(1))|p_t|$. Suppose that $\frac{2}{2-r(1)} < s_t < 2r(1)^{-1}$. Then from Eq. (7), it holds that

$$\left| \frac{p_{t+1}}{p_t} \right| = \left| 1 - \frac{2}{s_t} \right| \le 1 - r(1).$$

Hence, $|p_{t+1}| \le (1 - r(1))|p_t|$. Now we prove $s_{t+1} < 2r(1)^{-1}$. Assume the contrary that $s_{t+1} \ge 2r(1)^{-1}$. Then, $r(p_{t+1}) = \frac{q_{t+1}}{s_{t+1}} < q_{t+1} < 1 - \frac{\delta}{2}$ so that $|p_{t+1}| \ge \hat{r}(1 - \frac{\delta}{2})$. By Mean Value Theorem, there exists $p_t^* \in (|p_{t+1}|, |p_t|)$ such that

$$
\begin{aligned}
\frac{1}{r(p_{t+1})} &= \frac{1}{r(|p_t| - (|p_t| - |p_{t+1}|))} = \frac{1}{r(p_t)} + \frac{r'(p_t^*)}{r(p_t^*)^2}(|p_t| - |p_{t+1}|) \\
&\le \frac{1}{r(p_t)} + \frac{r'(|p_{t+1}|)}{r(p_{t+1})^2}(r(1)|p_t|) \\
&\le \frac{1}{r(p_t)} - \frac{|r'(\hat{r}(1 - \frac{\delta}{2}))|}{(1 - \frac{\delta}{2})^2}\left( r(1)\hat{r}\left(1 - \frac{\delta}{2}\right) \right) \\
&= \frac{1}{r(p_t)} - \Omega_{\delta,\phi}(1),
\end{aligned}
$$

where we used Assumption C.2 (i) and $\hat{r}(1 - \frac{\delta}{2}) \le |p_{t+1}| \le (1 - r(1))|p_t|$. Consequently,

$$s_{t+1} = \frac{q_{t+1}}{r(p_{t+1})} = (1 + \mathcal{O}(\eta))q_t\left( \frac{1}{r(p_t)} - \Omega_{\delta,\phi}(1) \right) \le \frac{q_t}{r(p_t)} = s_t < 2r(1)^{-1},$$

for small step size $\eta$. This gives a contradiction to our assumption that $s_{t+1} \ge 2r(1)^{-1}$. Hence, we can conclude that $s_{t+1} < 2r(1)^{-1}$, as desired.

We proved that for any $0 \le t \le T$, if $\frac{2}{2-r(1)} < s_t < 2r(1)^{-1}$, it holds that $s_{t+1} < 2r(1)^{-1}$ and $|p_{t+1}| \le (1 - r(1))|p_t|$. At initialization, $|p_0| \le 1$ and $q_0 < 1$, so that $s_0 < r(1)^{-1}$. If $s_0 \le \frac{2}{2-r(1)}$, then $t_0 = 0$ is the desired time step. Suppose that $s_0 > \frac{2}{2-r(1)}$. Then, we have $s_1 < 2r(1)^{-1}$ and $|p_1| \le (1 - r(1))|p_0| \le 1 - r(1)$. Then we have either $s_1 \le \frac{2}{2-r(1)}$, or $\frac{2}{2-r(1)} < s_1 < 2r(1)^{-1}$. In the previous case, $t_0 = 1$ is the desired time step. In the latter case, we can repeat the same argument and obtain $s_2 < 2r(1)^{-1}$ and $|p_2| \le (1 - r(1))^2$. By inductively repeating the same argument, we can obtain a time step $t_0 \le \log(\hat{r}(1 - \frac{\delta}{2}))/\log(1 - r(1)) = \mathcal{O}_{\delta,\phi}(1)$ such that either $s_{t_0} \le \frac{2}{2-r(1)}$, or $|p_{t_0}| \le \hat{r}(1 - \frac{\delta}{2})$. In the latter case, $r(p_{t_0}) \ge 1 - \frac{\delta}{2} > q_{t_0}$, and hence $s_{t_0} < 1 < \frac{2}{2-r(1)}$. Therefore, $t_0 = \mathcal{O}_{\delta,\phi}(1)$ is the desired time step satisfying $s_{t_0} \le \frac{2}{2-r(1)}$. $\qquad\square$

According to Lemma C.7, there exists a time step $t_0 = \mathcal{O}_{\delta,\phi}(1)$ such that $s_{t_0} \le \frac{2}{2-r(1)} < 2(1+\eta^2)^{-1}$ for small step size $\eta$. Now we prove the lemma below.

**Lemma C.8.** *Suppose that $s_t \le 1$. Then, it holds that $s_{t+1} \le 1 + \mathcal{O}(\eta)$.*

*Proof.* For any $p \in (0, \hat{r}(\frac{q_t}{2}))$, we have $r(p) \ge \frac{q_t}{2}$ so that $|f_{q_t}(p)| = (-1 + \frac{2r(p_t)}{q_t})p_t$. Hence,

$$\frac{\partial}{\partial p}|f_{q_t}(p)| = \frac{2r(p)}{q_t}\left( 1 + \frac{pr'(p)}{p} \right) - 1,$$

for any $p \in (0, \hat{r}(\frac{q_t}{2}))$. By Assumption C.2 (ii), (iv) and $q_t \ge c_0 \ge r(z_1) + \frac{1}{2} \ge 2r(z_1)$ where $z_1 = \sup_z\{ \frac{zr'(z)}{r(z)} \ge -1 \}$, both $r(p)$ and $(1 + \frac{pr'(p)}{r(p)})$ are positive, decreasing function on $(0, \hat{r}(\frac{q_t}{2}))$. Consequently, $\frac{\partial}{\partial p}|f_{q_t}(p)|$ is a decreasing function on $(0, \hat{r}(\frac{q_t}{2}))$.

Now note that $\frac{q_t}{2} < q_t < 1$, which means $\hat{r}(1) = 0 < \hat{r}(q_t) < \hat{r}(\frac{q_t}{2})$ by the definition of $\hat{r}$. Note that $\frac{\partial}{\partial p}|f_{q_t}(p)|$ at $p = \hat{r}(q_t)$ evaluates to

$$\frac{\partial}{\partial p}|f_{q_t}(\hat{r}(q_t))| = 1 + \frac{2\hat{r}(q_t)r'(\hat{r}(q_t))}{r(\hat{r}(q_t))} \ge 1 + \frac{2\hat{r}(c_0)r'(\hat{r}(c_0))}{r(\hat{r}(c_0))} \ge 0,$$

where the inequalities used Assumption C.2 (ii) and $q_t > c_0 \geq r(z_0)$ where $z_0 := \sup_z\{\frac{zr'(z)}{r(z)} \geq -\frac{1}{2}\}$, from the statement of Theorem C.2.

Therefore, since $\frac{\partial}{\partial p}|f_{q_t}(p)|$ is decreasing on $(0, \hat{r}(\frac{q_t}{2}))$ and is nonnegative at $\hat{r}(q_t)$, for any $p \in (0, \hat{r}(q_t))$, it holds that $\frac{\partial}{\partial p}|f_{q_t}(p)| \geq 0$. In other words, $|f_{q_t}(p)|$ is an increasing function on $(0, \hat{r}(q_t))$. Since $0 \leq s_t \leq 1$, we have $|p_t| \leq \hat{r}(q_t)$ and it holds that

$$|p_{t+1}| = \left(-1 + \frac{2}{s_t}\right)|p_t| = |f_{q_t}(p_t)| \leq |f_{q_t}(\hat{r}(q_t))| = \hat{r}(q_t).$$

Therefore, with this inequality and Lemma C.6, we can conclude that

$$s_{t+1} = \frac{q_{t+1}}{r(p_{t+1})} = \frac{(1 + \mathcal{O}(\eta))q_t}{r(p_{t+1})} \leq \frac{q_t}{r(\hat{r}(q_t))} + \mathcal{O}(\eta) = 1 + \mathcal{O}(\eta).$$

$\square$

Using Lemma C.8, we prove the following lemma.

**Lemma C.9.** *For any $0 \leq t \leq T$, if $s_t < 2$ and $r(p_t) \leq 1 - \frac{\delta}{4}$, then*

$$|s_{t+1} - 1| \leq (1 - d)|s_t - 1| + \mathcal{O}(\eta),$$

*where $d \in (0, \frac{1}{2}]$ is a constant which depends on $\delta$ and $\phi$.*

*Proof.* From Eq. (7) it holds that

$$\frac{p_{t+1}}{p_t} = 1 - \frac{2}{s_t} < 0,$$

so that $p_t$ and $p_{t+1}$ have opposite signs. By Mean Value Theorem, there exists $\theta_t$ between $-1$ and $(1 - \frac{2}{s_t})$ satisfying

$$
\begin{aligned}
\frac{1}{r(p_{t+1})} &= \frac{1}{r\left(-p_t + \left(\frac{2(s_t - 1)}{s_t}\right)p_t\right)} \\
&= \frac{1}{r(-p_t)} - \frac{r'(\theta_t p_t)}{r(\theta_t p_t)^2}\left(\frac{2(s_t - 1)}{s_t}\right)p_t \\
&= \frac{1}{r(p_t)} - \frac{|r'(\theta_t p_t)|}{r(\theta_t p_t)^2}\left(\frac{2(s_t - 1)}{s_t}\right)|p_t|.
\end{aligned}
\tag{17}
$$

where the last equality used the fact that $p_t$ and $\theta_t p_t$ have opposite signs and $r'(z)$ and $z$ have opposite signs. Note that $|\theta_t p_t|$ is between $|p_t|$ and $|p_{t+1}|$. Consequently, the value $\frac{|r'(\theta_t p_t)|}{r(\theta_t p_t)^2}$ is between $\frac{|r'(p_t)|}{r(p_t)^2}$ and $\frac{|r'(p_{t+1})|}{r(p_{t+1})^2}$ by Assumption C.2 (i). We will prove the current lemma based on Eq. (17). We divide into following three cases: (1) $s_t \geq 1$ and $s_{t+1} \geq 1$, (2) $s_t \geq 1$ and $s_t < 1$, and (3) $s_t < 1$.

**Case 1.** Suppose that $s_t \geq 1$ and $s_{t+1} \geq 1$. Here, we have $|p_t| \geq \hat{r}(q_t) \geq \hat{r}(1 - \frac{\delta}{2})$ and similarly $|p_{t+1}| \geq \hat{r}(1 - \frac{\delta}{2})$. By Assumption C.2 (i), $\frac{|r'(\theta_t p_t)|}{r(\theta_t p_t)^2} \geq \frac{|r'(\hat{r}(1 - \frac{\delta}{2}))|}{(1 - \frac{\delta}{2})^2}$. Hence, Eq. (17) gives

$$\frac{1}{r(p_{t+1})} \leq \frac{1}{r(p_t)} - \frac{|r'(\hat{r}(1 - \frac{\delta}{2}))|}{(1 - \frac{\delta}{2})^2}\left(\frac{2(s_t - 1)}{s_t}\right)\hat{r}\left(1 - \frac{\delta}{2}\right).$$

Consequently, by Lemma C.6,

$$
\begin{aligned}
s_{t+1} = \frac{q_t(1 + \mathcal{O}(\eta))}{r(p_{t+1})} &= \frac{q_t}{r(p_{t+1})} + \mathcal{O}(\eta) \leq s_t - \frac{|r'(\hat{r}(1 - \frac{\delta}{2}))|}{(1 - \frac{\delta}{2})^2}\left(\frac{2(s_t - 1)}{s_t}\right)\hat{r}\left(1 - \frac{\delta}{2}\right)q_t + \mathcal{O}(\eta) \\
&\leq s_t - \frac{|r'(\hat{r}(1 - \frac{\delta}{2}))|}{(1 - \frac{\delta}{2})^2}(s_t - 1)\hat{r}\left(1 - \frac{\delta}{2}\right)\frac{1}{2} + \mathcal{O}(\eta)
\end{aligned}
$$

$$\leq s_t - \frac{\hat{r}(1 - \frac{\delta}{2})|r'(\hat{r}(1 - \frac{\delta}{2}))|}{2(1 - \frac{\delta}{2})^2}(s_t - 1) + \mathcal{O}(\eta),$$

where we used $q_t > c_0 > \frac{1}{2}$ and $s_t < 2$. Therefore, we can obtain the following inequality:

$$0 \leq s_{t+1} - 1 \leq \left(1 - \frac{\hat{r}(1 - \frac{\delta}{2})|r'(\hat{r}(1 - \frac{\delta}{2}))|}{2(1 - \frac{\delta}{2})^2}\right)(s_t - 1) + \mathcal{O}(\eta).$$

**Case 2.** Suppose that $s_t \geq 1$ and $s_{t+1} < 1$. Here, we have $r(p_{t+1}) > q_{t+1} \geq q_t \geq r(p_t)$, so that $|p_{t+1}| < |p_t|$. Consequently, $\frac{|r'(\theta_t p_t)|}{r(\theta_t p_t)^2} \leq \frac{|r'(p_t)|}{r(p_t)^2}$ by Assumption C.2 (i). Hence, we can deduce from Eq. (17) that

$$\frac{1}{r(p_{t+1})} \geq \frac{1}{r(p_t)} - \frac{|r'(p_t)|}{r(p_t)^2}\left(\frac{2(s_t - 1)}{s_t}\right)|p_t|$$

$$= \frac{1}{r(p_t)} - \frac{2|p_t r'(p_t)|}{r(p_t)q_t}(s_t - 1)$$

$$= \frac{1}{r(p_t)} + \frac{2p_t r'(p_t)}{r(p_t)q_t}(s_t - 1).$$

Consequently, by Assumption C.2 (ii),

$$s_{t+1} \geq \frac{q_t}{r(p_{t+1})} \geq s_t + \frac{2p_t r'(p_t)}{r(p_t)}(s_t - 1) \geq s_t + \frac{2\hat{r}(\frac{c_0}{2})r'(\hat{r}(\frac{c_0}{2}))}{r(\hat{r}(\frac{c_0}{2}))}(s_t - 1),$$

where we used $r(p_t) \geq \frac{q_t}{2} > \frac{c_0}{2}$. Therefore, we can obtain the following inequality:

$$0 \leq 1 - s_{t+1} \leq -\left(1 + \frac{4\hat{r}(\frac{c_0}{2})r'(\hat{r}(\frac{c_0}{2}))}{c_0}\right)(s_t - 1),$$

where $-1 < \frac{\hat{r}(\frac{c_0}{2})r'(\hat{r}(\frac{c_0}{2}))}{r(\hat{r}(\frac{c_0}{2}))} = \frac{2\hat{r}(\frac{c_0}{2})r'(\hat{r}(\frac{c_0}{2}))}{c_0} < 0$, since $c_0 \geq r(z_1) + \frac{1}{2} \geq 2r(z_1)$ with $z_1 = \sup_z\{\frac{zr'(z)}{r(z)} \geq -1\}$, and $r(z_1) \leq \frac{1}{2}$ holds by Assumption C.2 (iv).

**Case 3.** Suppose that $s_t < 1$. By Lemma C.8, it holds that $s_{t+1} \leq 1 + \mathcal{O}(\eta)$. Moreover, we assumed $r(p_t) \leq 1 - \frac{\delta}{4}$, so that $|p_t| \geq \hat{r}(1 - \frac{\delta}{4})$. We also have

$$|p_{t+1}| = \left(-1 + \frac{2}{s_t}\right)|p_t| > |p_t| \geq \hat{r}\left(1 - \frac{\delta}{4}\right).$$

Consequently, by Assumption C.2 (i), it holds that $\frac{|r'(\theta_t p_t)|}{r(\theta_t p_t)^2} \geq \frac{|r'(\hat{r}(1 - \frac{\delta}{4}))|}{(1 - \frac{\delta}{4})^2}$. Hence, by Eq. (17), we have

$$\frac{1}{r(p_{t+1})} \geq \frac{1}{r(p_t)} + \frac{|r'(\hat{r}(1 - \frac{\delta}{4}))|}{(1 - \frac{\delta}{4})^2}\left(\frac{2(1 - s_t)}{s_t}\right)\hat{r}\left(1 - \frac{\delta}{4}\right)$$

$$\geq \frac{1}{r(p_t)} + \frac{|r'(\hat{r}(1 - \frac{\delta}{4}))|}{(1 - \frac{\delta}{4})^2}2(1 - s_t)\hat{r}\left(1 - \frac{\delta}{4}\right)$$

$$= \frac{1}{r(p_t)} + \frac{2\hat{r}(1 - \frac{\delta}{4})|r'(\hat{r}(1 - \frac{\delta}{4}))|}{(1 - \frac{\delta}{4})^2}(1 - s_t),$$

and hence,

$$s_{t+1} \geq \frac{q_t}{r(p_{t+1})} \geq s_t + \frac{\hat{r}(1 - \frac{\delta}{4})|r'(\hat{r}(1 - \frac{\delta}{4}))|}{(1 - \frac{\delta}{4})^2}(1 - s_t),$$

where we used $q_t > \frac{1}{2}$. Therefore, we can obtain the following inequality:

$$-\mathcal{O}(\eta) \leq 1 - s_{t+1} \leq \left(1 - \frac{\hat{r}(1 - \frac{\delta}{4})|r'(\hat{r}(1 - \frac{\delta}{4}))|}{(1 - \frac{\delta}{4})^2}\right)(1 - s_t),$$

where the first inequality is from Lemma C.8.

Combining the three cases, we can finally conclude that if we choose

$$d := \min\left\{\frac{1}{2}, \frac{\hat{r}(1-\frac{\delta}{2})|r'(\hat{r}(1-\frac{\delta}{2}))|}{2(1-\frac{\delta}{2})^2}, 2\left(1 + \frac{2\hat{r}(\frac{c_0}{2})r'(\hat{r}(\frac{c_0}{2}))}{c_0}\right), \frac{\hat{r}(1-\frac{\delta}{4})|r'(\hat{r}(1-\frac{\delta}{4}))|}{(1-\frac{\delta}{4})^2}\right\} \in \left(0, \frac{1}{2}\right],$$

then $|s_{t+1} - 1| \leq (1-d)|s_t - 1| + \mathcal{O}(\eta)$, as desired. $\qquad\square$

Lemma C.9 implies that if $s_t < 2$ and $|p_t| \geq \hat{r}(1-\frac{\delta}{4})$, then $|s_t - 1|$ exponentially decreases. We prove Lemma C.10 to handle the regime $|p_t| < \hat{r}(1-\frac{\delta}{4})$, which is stated below.

**Lemma C.10.** *For any $0 \leq t \leq T$, if $r(p_t) \geq 1 - \frac{\delta}{4}$, it holds that*

$$\left|\frac{p_{t+1}}{p_t}\right| \geq \frac{2}{2-\delta}.$$

*Proof.* If $r(p_t) \geq 1 - \frac{\delta}{4}$, then $s_t = \frac{q_t}{r(p_t)} < \frac{1-\frac{\delta}{2}}{1-\frac{\delta}{4}} = \frac{4-2\delta}{4-\delta}$, where we used $q_t < 1 - \frac{\delta}{2}$ for any $0 \leq t \leq T$. Consequently,

$$\left|\frac{p_{t+1}}{p_t}\right| = \frac{2}{s_t} - 1 \geq \frac{2(4-\delta)}{4-2\delta} - 1 = \frac{2}{2-\delta}.$$

$\qquad\square$

Now we prove Proposition C.11, which proves that $s_t$ reaches close to 1 with error bound of $\mathcal{O}(\eta)$.

**Proposition C.11.** *There exists a time step $t_a^* = \mathcal{O}_{\delta,\phi}(\log(\eta^{-1}))$ satisfying*

$$s_{t_a^*} = 1 + \mathcal{O}_{\delta,\phi}(\eta). \tag{18}$$

*Proof.* By Lemma C.7, there exists a time step $t_0 = \mathcal{O}_{\delta,\phi}(1)$ such that $s_{t_0} \leq \frac{2}{2-r(1)}$. Here, we divide into two possible cases: (1) $s_{t_0} < 1$, and (2) $1 \leq s_{t_0} \leq \frac{2}{2-r(1)}$.

**Case 1.** Suppose that $s_{t_0} < 1$. By Lemma C.10, if $r(p_{t_0}) \geq 1-\frac{\delta}{4}$ (or equivalently, $|p_{t_0}| \leq \hat{r}(1-\frac{\delta}{4})$), then there exists a time step $t_1 \leq t_0 + \log(\frac{\hat{r}(1-\frac{\delta}{4})}{|p_{t_0}|})/\log(\frac{2}{2-\delta}) = \mathcal{O}_{\delta,\phi}(1)$ such that $|p_{t_1}| \geq \hat{r}(1 - \frac{\delta}{4})$. We denote the first time step satisfying $|p_{t_1}| \geq \hat{r}(1-\frac{\delta}{4})$ and $t_1 \geq t_0$ by $t_1 = \mathcal{O}_{\delta,\phi}(1)$. By Lemma C.8, it holds that $s_{t_1} \leq 1 + \mathcal{O}(\eta)$ since $s_{t_1-1} < 1$. Consequently, if $s_{t_1} \geq 1$, then $|s_{t_1} - 1| \leq \mathcal{O}(\eta)$ so that $t_a^* = t_1$ is the desired time step. Hence, it suffices to consider the case when $s_{t_1} < 1$. Here, we can apply Lemma C.9 which implies that

$$|s_{t_1+1} - 1| \leq (1-d)|s_{t_1} - 1| + \mathcal{O}(\eta),$$

where $d$ is a constant which depends on $\delta$ and $\phi$. Then, there are two possible cases: either $|s_{t_1} - 1| \leq \mathcal{O}(\eta d^{-1})$, or $|s_{t_1+1} - 1| \leq (1-\frac{d}{2})|s_{t_1} - 1|$. It suffices to consider the latter case, suppose that $|s_{t_1+1}-1| \leq (1-\frac{d}{2})|s_{t_1}-1|$. Since we are considering the case $s_{t_1} < 1$, again by Lemma C.8, we have $s_{t_1+1} \leq 1+\mathcal{O}(\eta)$. Since $|\frac{p_{t_1+1}}{p_{t_1}}| = \frac{2}{s_{t_1}} - 1 > 1$, we have $|p_{t_1+1}| \geq |p_{t_1}| \geq \hat{r}(1-\frac{\delta}{4})$. This means that we can again apply Lemma C.9 and repeat the analogous argument. Hence, there exists a time step $t_2 \leq t_1 + \log(\frac{\eta}{1-s_{t_1}})/\log(1-\frac{d}{2}) = \mathcal{O}_{\delta,\phi}(\log(\eta^{-1}))$, such that $|s_{t_2} - 1| \leq \mathcal{O}(\eta d^{-1}) = \mathcal{O}_{\delta,\phi}(\eta)$.

**Case 2.** Suppose that $1 \leq s_{t_0} \leq \frac{2}{2-r(1)}$. Then, $r(p_{t_0}) \leq q_{t_0} \leq 1 - \frac{\delta}{2}$, so we can apply Lemma C.9. There are two possible cases: either $|s_{t_0+1} - 1| \leq \mathcal{O}(\eta d^{-1}) = \mathcal{O}_{\delta,\phi}(\eta)$, or $|s_{t_0+1} - 1| \leq (1-\frac{d}{2})|s_{t_0} - 1|$. It suffices to consider the latter case. If $s_{t_0+1} \geq 1$, we can again apply Lemma C.9 and repeat the analogous argument. Hence, we can obtain a time step $t_0' \leq t_0 + \log(\frac{\eta}{1-s_{t_0}})/\log(1-\frac{d}{2}) = \mathcal{O}_{\delta,\phi}(\log(\eta^{-1}))$ such that either $s_{t_0'} < 1$ or $|s_{t_0'} - 1| = \mathcal{O}_{\delta,\phi}(\eta)$ is satisfied. If $s_{t_0'} < 1$, we proved in Case 1 that there exists a time step $t_2' = t_0' + \mathcal{O}_{\delta,\phi}(\log(\eta^{-1}))$ such that $|s_{t_2'} - 1| \leq \mathcal{O}_{\delta,\phi}(\eta)$, and this is the desired bound. $\qquad\square$

Now we carefully handle the error term $\mathcal{O}(\eta)$ obtained in Proposition C.11 and a provide tighter bound on $s_t$ by proving Lemma C.12 stated below.

**Lemma C.12.** *If* $|s_t - 1| \le \mathcal{O}_{\delta,\phi}(\eta)$*, then it holds that*

$$|s_{t+1} - 1 - h(p_{t+1})\eta| \le \left(1 + \frac{2p_t r'(p_t)}{r(p_t)}\right)|s_t - 1 - h(p_t)\eta| + \mathcal{O}_{\delta,\phi}(\eta^2 p_t^2),$$

*where*

$$h(p) := \begin{cases} -\frac{\phi(p)^2 r(p)}{p r'(p)} & \text{if } p \ne 0, \text{ and} \\ -\frac{1}{r''(0)} & \text{if } p = 0. \end{cases}$$

*Proof.* Suppose that $|s_t - 1| \le \mathcal{O}_{\delta,\phi}(\eta)$. Then, $|p_{t+1}| = \left|1 - \frac{2}{s_t}\right||p_t| = (1 + \mathcal{O}_{\delta,\phi}(\eta))|p_t|$. By Eq. (17) proved in Lemma C.9, there exists $\epsilon_t = \mathcal{O}_{\delta,\phi}(\eta)$ such that

$$\begin{aligned}
\frac{1}{r(p_{t+1})} &= \frac{1}{r(p_t)} + \frac{r'((1+\epsilon_t)p_t)}{r((1+\epsilon_t)p_t)^2}\left(\frac{2(s_t - 1)}{s_t}\right)p_t \\
&= \frac{1}{r(p_t)} + \left(\frac{r'(p_t)}{r(p_t)^2} + \mathcal{O}_{\delta,\phi}(\eta p_t)\right)\left(\frac{2(s_t - 1)}{s_t}\right)p_t \\
&= \frac{1}{r(p_t)} + \frac{r'(p_t)}{r(p_t)^2}\left(\frac{2(s_t - 1)}{s_t}\right)p_t + \mathcal{O}_{\delta,\phi}(\eta^2 p_t^2),
\end{aligned}$$

where we used the Taylor expansion on $\frac{r'(p)}{r(p)^2}$ with the fact that $\frac{d}{dp}\left(\frac{r'(p)}{r(p)^2}\right)$ is bounded on $[-4,4]$ and that $|p_t| \le 4$ to obtain the second equality. Note that $q_{t+1} = (1 - \eta\phi(p_t)^2)^{-2}q_t = (1 + 2\eta\phi(p_t)^2)q_t + \mathcal{O}(\eta^2)$ by Eq (7). Consequently,

$$\begin{aligned}
s_{t+1} &= (1 + 2\eta\phi(p_t)^2)\left(s_t + \frac{r'(p_t)}{r(p_t)^2}\left(\frac{2(s_t - 1)}{s_t}\right)p_t q_t\right) + \mathcal{O}_{\delta,\phi}(\eta^2 p_t^2) \\
&= (1 + 2\eta\phi(p_t)^2)s_t + \frac{2p_t r'(p_t)}{r(p_t)}(s_t - 1) + \mathcal{O}_{\delta,\phi}(\eta^2 p_t^2) \\
&= 1 + \left(1 + \frac{2p_t r'(p_t)}{r(p_t)}\right)(s_t - 1) + 2\eta\phi(p_t)^2 + \mathcal{O}_{\delta,\phi}(\eta^2 p_t^2).
\end{aligned}$$

Note that $h$ is even, and twice continuously differentiable function by Lemma C.13. Consequently, $h'(0) = 0$ and $h'(p) = \mathcal{O}_\phi(p)$, since $h''$ is bounded on closed interval. Consequently, $h(p_{t+1}) = h((1 + \mathcal{O}_{\delta,\phi}(\eta))p_t) = h(p_t) + \mathcal{O}_{\delta,\phi}(\eta p_t^2)$. Hence, we can obtain the following:

$$\begin{aligned}
s_{t+1} - 1 - h(p_{t+1})\eta &= s_{t+1} - 1 - h(p_t)\eta + \mathcal{O}_{\delta,\phi}(\eta^2 p_t^2) \\
&= s_{t+1} - 1 + \frac{\phi(p_t)^2 r(p_t)}{p_t r'(p_t)}\eta + \mathcal{O}_{\delta,\phi}(\eta^2 p_t^2) \\
&= \left(1 + \frac{2p_t r'(p_t)}{r(p_t)}\right)\left(s_t - 1 + \frac{\phi(p_t)^2 r(p_t)}{p_t r'(p_t)}\eta\right) + \mathcal{O}_{\delta,\phi}(\eta^2 p_t^2) \\
&= \left(1 + \frac{2p_t r'(p_t)}{r(p_t)}\right)(s_t - 1 - h(p_t)\eta) + \mathcal{O}_{\delta,\phi}(\eta^2 p_t^2)
\end{aligned}$$

Note that $r(p_t) = (1 + \mathcal{O}_{\delta,\phi}(\eta))q_t \ge (1 + \mathcal{O}_{\delta,\phi}(\eta))q_0 \ge c_0 \ge r(z_0)$ for small step size $\eta$, where $z_0 = \sup\{\frac{zr'(z)}{r(z)} \ge -\frac{1}{2}\}$. Consequently, it holds that $1 + \frac{2p_t r'(p_t)}{r(p_t)} \ge 0$. Therefore, we can conclude that

$$|s_{t+1} - 1 - h(p_{t+1})\eta| \le \left(1 + \frac{2p_t r'(p_t)}{r(p_t)}\right)|s_t - 1 - h(p_t)\eta| + \mathcal{O}_{\delta,\phi}(\eta^2 p_t^2),$$

as desired. □

Now we give the proof of Theorem C.2, restated below for the sake of readability.

**Theorem C.2** (EoS regime, Phase I). *Let $\eta > 0$ be a small enough constant and $\phi$ be an activation function satisfying Assumptions 5.1, C.1, and C.2. Let $z_0 := \sup_z \{ \frac{zr'(z)}{r(z)} \geq -\frac{1}{2} \}$, $z_1 := \sup_z \{ \frac{zr'(z)}{r(z)} \geq -1 \}$, and $c_0 := \max\{r(z_0), r(z_1) + \frac{1}{2}\} \in (\frac{1}{2}, 1)$. Let $\delta \in (0, 1 - c_0)$ be any given constant. Suppose that the initialization $(p_0, q_0)$ satisfies $|p_0| \leq 1$ and $q_0 \in (c_0, 1 - \delta)$. Consider the reparameterized GD trajectory characterized in Eq. (7). We assume that for all $t \geq 0$ such that $q_t < 1$, we have $p_t \neq 0$. Then, there exists a time step $t_a = \mathcal{O}_{\delta,\phi}(\log(\eta^{-1}))$ such that for any $t \geq t_a$,*

$$\frac{q_t}{r(p_t)} = 1 + h(p_t)\eta + \mathcal{O}_{\delta,\phi}(\eta^2)$$

*where $h : \mathbb{R} \to \mathbb{R}$ is a function defined as*

$$h(p) := \begin{cases} -\frac{\phi(p)^2 r(p)}{pr'(p)} & \text{if } p \neq 0, \text{ and} \\ -\frac{1}{r''(0)} & \text{if } p = 0. \end{cases}$$

*Proof of Theorem C.2.* By Proposition C.11, there exists a time step $t_a^* = \mathcal{O}_{\delta,\ell}(\log(\eta^{-1}))$ which satisfies:

$$|s_{t_a^*} - 1| = \left| \frac{q_{t_a^*}}{r(p_{t_a^*})} - 1 \right| = \mathcal{O}_{\delta,\phi}(\eta).$$

By Lemma C.12, there exists a constant $D > 0$ which depends on $\delta, \phi$ such that if $|s_t - 1| = \mathcal{O}_{\delta,\phi}(\eta)$, then

$$|s_{t+1} - 1 - h(p_{t+1})\eta| \leq \left( 1 + \frac{2p_t r'(p_t)}{r(p_t)} \right) |s_t - 1 - h(p_t)\eta| + D\eta^2 p_t^2. \tag{19}$$

Hence, if $|s_t - 1| = \mathcal{O}_{\delta,\phi}(\eta)$ and $|s_t - 1 - h(p_t)\eta| \geq \left( -\frac{p_t r(p_t)}{r'(p_t)} \right) D\eta^2$, then

$$|s_{t+1} - 1 - h(p_{t+1})\eta| \leq \left( 1 + \frac{p_t r'(p_t)}{r(p_t)} \right) |s_t - 1 - h(p_t)\eta|. \tag{20}$$

For any $t \leq T$, we have $q_t < 1 - \frac{\delta}{2}$ so that if $|s_t - 1| = \mathcal{O}_{\delta,\phi}(\eta)$, then $r(p_t) \leq (1 + \mathcal{O}_{\delta,\phi}(\eta))q_t < 1 - \frac{\delta}{4}$ for small step size $\eta$. From Eq. (20) with $t = t_a^*$, we have either

$$|s_{t_a^*} - 1 - h(p_{t_a^*})\eta| < \left( -\frac{p_{t_a^*} r(p_{t_a^*})}{r'(p_{t_a^*})} \right) D\eta^2,$$

or

$$|s_{t_a^*+1} - 1 - h(p_{t_a^*+1})\eta| \leq \left( 1 + \frac{\hat{r}(1 - \frac{\delta}{4}) r'(\hat{r}(1 - \frac{\delta}{4}))}{(1 - \frac{\delta}{4})} \right) |s_{t_a^*} - 1 - h(p_{t_a^*})\eta|,$$

where we used Assumption C.2 (ii) and $|p_t| > \hat{r}(1 - \frac{\delta}{4})$. In the latter case, $|s_{t_a^*+1} - 1| = \mathcal{O}_{\delta,\phi}(\eta)$ continues to hold and we can again use Eq. (20) with $t = t_a^* + 1$. By repeating the analogous arguments, we can obtain the time step

$$t_a \leq t_a^* + \frac{\log\left( -\frac{D\eta^2}{r''(0)|s_{t_a^*} - 1 - h(p_{t_a^*})\eta|} \right)}{\log\left( 1 + \frac{\hat{r}(1 - \frac{\delta}{4}) r'(\hat{r}(1 - \frac{\delta}{4}))}{(1 - \frac{\delta}{4})} \right)} = \mathcal{O}_{\delta,\ell}(\log(\eta^{-1})),$$

which satisfies: either

$$|s_{t_a} - 1 - h(p_{t_a})\eta| < \left( -\frac{p_{t_a} r(p_{t_a})}{r'(p_{t_a})} \right) D\eta^2,$$

or

$$|s_{t_a} - 1 - h(p_{t_a})\eta| \leq \left( -\frac{1}{r''(0)} \right) D\eta^2 \leq \left( -\frac{p_{t_a} r(p_{t_a})}{r'(p_{t_a})} \right) D\eta^2 \leq \left( -\frac{4r(4)}{r'(4)} \right) D\eta^2,$$

where we used $|p_t| \leq 4$ from Lemma C.6 and $-\frac{zr(z)}{r'(z)} \geq -\frac{1}{r''(0)}$ for any $z$ by Assumption C.2 (iii).

By Eq. (19), if $|s_t - 1 - h(p_t)\eta| \leq \left(-\frac{4r(4)}{r'(4)}\right) D\eta^2$ is satisfied for any time step $t$, then

$$|s_{t+1} - 1 - h(p_{t+1})\eta| \leq \left(1 + \frac{2p_t r'(p_t)}{r(p_t)}\right)\left(-\frac{4r(4)}{r'(4)}\right) D\eta^2 + D\eta^2 p_t^2 \leq \left(-\frac{4r(4)}{r'(4)}\right) D\eta^2,$$

by $|p_t| \leq 4$ from Lemma C.6 and Assumption C.2 (iii).

Hence, by induction, we have the desired bound as following: for any $t \geq t_a$,

$$|s_t - 1 - h(p_t)\eta| \leq \left(-\frac{4r(4)}{r'(4)}\right) D\eta^2 = \mathcal{O}_{\delta,\ell}(\eta^2),$$

by $|p_t| \leq 4$ and Assumption C.2 (iii). □

## C.4 Proof of Theorem C.3

In this subsection, we prove Theorem C.3. We start by proving Lemma C.13 which provides a useful property of $h$ defined in Theorem C.2.

**Lemma C.13.** *Consider the function $h$ defined in Theorem C.2, given by*

$$h(p) := \begin{cases} -\frac{\phi(p)^2 r(p)}{pr'(p)} & \text{if } p \neq 0, \text{ and} \\ -\frac{1}{r''(0)} & \text{if } p = 0. \end{cases}$$

*Then, $h$ is a positive, even, and bounded twice continuously differentiable function.*

*Proof.* By Assumption C.1, $h$ is a positive, even function. Moreover, $h$ is continuous since $\lim_{p \to 0} h(p) = -\frac{1}{r''(0)} = h(0)$. Continuous function on a compact domain is bounded, so $h$ is bounded on the closed interval $[-1, 1]$. Note that $\phi(p)^2 \leq 1$, and $\left(-\frac{r(p)}{pr'(p)}\right)$ is positive, decreasing function on $p > 0$ by Assumption C.2 (ii). Hence, $h$ is bounded on $[1, \infty)$. Since $h$ is even, $h$ is bounded on $(-\infty, 1]$. Therefore, $h$ is a bounded on $\mathbb{R}$.

We finally prove that $h$ is twice continuously differentiable. Since $r$ is even and $C^4$ on $\mathbb{R}$, we can check that for any $p \neq 0$,

$$h'(p) = -\frac{2r(p)^2}{r'(p)} - \frac{\phi(p)^2}{p} + \frac{\phi(p)^2 r(p)(r'(p) + pr''(p))}{p^2 r'(p)^2},$$

and $h'(p) = 0$. Moreover, for any $p \neq 0$,

$$h''(p) = -6r(p) + \frac{2\phi(p)^2}{p^2} + \frac{\phi(p)^2 r''(p)}{pr'(p)} + \frac{\phi(p)^2 r(p) r^{(3)}(p)}{pr'(p)^2}$$
$$+ \frac{2r(p)^2(r'(p) + 2pr''(p))}{pr'(p)^2} - \frac{\phi(p)^2 r(p)(2r'(p) + pr''(p))}{p^3 r'(p)^2} - \frac{\phi(p)^2 r(p) r''(p)(r'(p) + 2pr''(p))}{p^2 r'(p)^3},$$

and

$$h''(0) = -1 - \frac{2\phi^{(3)}(0)}{3r''(0)} + \frac{r^{(4)}(0)}{3r''(0)^2}.$$

Since $\lim_{p \to 0} h''(p) = h''(0)$, we can conclude that $h$ is a twice continuously differentiable function. □

Now we give the proof of Theorem C.3, restated below for the sake of readability.

**Theorem C.3** (EoS regime, Phase II). *Under the same settings as in Theorem C.2, there exists a time step $t_b = \Omega((1 - q_0)\eta^{-1})$, such that $q_{t_b} \leq 1$ and $q_t > 1$ for any $t > t_b$. Moreover, the GD iterates $(p_t, q_t)$ converge to the point $(0, q^*)$ such that*

$$q^* = 1 - \frac{\eta}{r''(0)} + \mathcal{O}_{\delta,\phi}(\eta^2).$$

*Proof.* We first prove that there exists a time step $t \geq 0$ such that $q_t > 1$. Assume the contrary that $q_t \leq 1$ for all $t \geq 0$. Let $t_a$ be the time step obtained in Theorem C.2. Then for any $t \geq t_a$, we have

$$r(p_t) = (1 - h(p_t)\eta + \mathcal{O}_{\delta,\phi}(\eta^2))q_t \leq 1 - \frac{h(p_t)\eta}{2},$$

for small step size $\eta$. The function $g(p) := r(p) - 1 + \frac{h(p)\eta}{2}$ is even, continuous, and has the function value $g(0) = \frac{\eta}{2|r''(0)|} > 0$. Consequently, there exists a positive constant $\epsilon > 0$ such that $g(p) > 0$ for all $p \in (-\epsilon, \epsilon)$. Then, we have $|p_t| \geq \epsilon$ for all $t \geq t_a$, since $g(p_t) \leq 0$. This implies that for any $t \geq t_a$, it holds that

$$\frac{q_{t+1}}{q_t} = (1 - \eta\phi(p_t)^2)^{-2} \geq (1 - \eta\phi(\epsilon)^2)^{-2} > 1,$$

so $q_t$ grows exponentially, which results in the existence of a time step $t'_b \geq t_a$ such that $q_{t'_b} > 1$, a contradiction.

Therefore, there exists a time step $t_b$ such that $q_{t_b} \leq 1$ and $q_t > 1$ for any $t > t_b$, i.e., $q_t$ jumps across the value 1. This holds since the sequence $(q_t)$ is monotonically increasing. For any $t \leq t_b$, we have $q_{t+1} \leq q_t + \mathcal{O}(\eta)$ by Lemma C.6, and this implies that $t_b \geq \Omega((1 - q_0)\eta^{-1})$, as desired.

Lastly, we prove the convergence of GD iterates $(p_t, q_t)$. Let $t > t_b$ be given. Then, $q_t \geq q_{t_b+1} > 1$ and it holds that

$$\left|\frac{p_{t+1}}{p_t}\right| = \frac{2r(p_t)}{q_t} - 1 \leq \frac{2}{q_{t_b+1}} - 1 < 1.$$

Hence, $|p_t|$ is exponentially decreasing for $t > t_b$. Therefore, $p_t$ converges to 0 as $t \to \infty$. Since the sequence $(q_t)_{t=0}^{\infty}$ is monotonically increasing and bounded (due to Theorem C.2, it converges. Suppose that $(p_t, q_t)$ converges to the point $(0, q^*)$. By Theorem C.2, we can conclude that

$$\left|q^* - 1 + \frac{\eta}{r''(0)}\right| = \mathcal{O}_{\phi,\ell}(\eta^2),$$

which is the desired bound. $\qquad\square$

## C.5  Proof of Theorem C.4

In this subsection, we prove Theorem C.4. We first prove an important bound on the sharpness value provided by the Proposition C.14 stated below.

**Proposition C.14.** *For any* $(x, y) \in \mathbb{R}^2$ *with* $r(x) + xr'(x) \geq 0$*, the following inequality holds:*

$$(r(x) + xr'(x))y^2 \leq \lambda_{\max}(\nabla^2 \mathcal{L}(x, y)) \leq (r(x) + xr'(x))y^2 + \frac{4x^2 r(x)}{r(x) + xr'(x)},$$

*Proof.* Let $(x, y) \in \mathbb{R}^2$ be given. The loss Hessian at $(x, y)$ is

$$\nabla^2 \mathcal{L}(x, y) = \begin{bmatrix} (\phi(x)\phi''(x) + \phi'(x)^2)y^2 & 2\phi(x)\phi'(x)y \\ 2\phi(x)\phi'(x)y & \phi(x)^2 \end{bmatrix}.$$

Note that the largest absolute value of the eigenvalue of a symmetric matrix equals to its spectral norm. Since trace of the Hessian, $\text{tr}(\nabla^2 \mathcal{L}(x, y)) = (\phi(x)\phi''(x) + \phi'(x)^2)y^2 + \phi(x)^2 = (r(x) + xr'(x))y^2 + \phi(x)^2 \geq 0$, is non-negative, the spectral norm of the Hessian $\nabla^2 \mathcal{L}(x, y)$ equals to its largest eigenvalue. Hence, we have

$$\begin{aligned}
\lambda_{\max}(\nabla^2 \mathcal{L}(x, y)) &= \|\nabla^2 \mathcal{L}(x, y)\|_2 \\
&\geq \left\|\nabla^2 \mathcal{L}(x, y) \cdot \begin{bmatrix} 1 \\ 0 \end{bmatrix}\right\|_2 \\
&= \left[((\phi(x)\phi''(x) + \phi'(x)^2)y^2)^2 + (2\phi(x)\phi'(x)y)^2\right]^{\frac{1}{2}} \\
&\geq (\phi(x)\phi''(x) + \phi'(x)^2)y^2 \\
&= (r(x) + xr'(x))y^2,
\end{aligned}$$

which is the desired lower bound. We also note that for any matrix $\boldsymbol{A}$, the inequality $\|\boldsymbol{A}\|_2 \leq \|\boldsymbol{A}\|_F$ holds, where $\|\boldsymbol{A}\|_F$ is the Frobenius norm. Hence, we have

$$
\begin{aligned}
\lambda_{\max}(\nabla^2 \mathcal{L}(x,y)) &= \|\nabla^2 \mathcal{L}(x,y)\|_2 \\
&\leq \|\nabla^2 \mathcal{L}(x,y)\|_F \\
&= \left[ ((\phi(x)\phi''(x) + \phi'(x)^2)y^2)^2 + 2(2\phi(x)\phi'(x)y)^2 + \phi(x)^4 \right]^{\frac{1}{2}} \\
&= \left[ (r(x) + xr'(x))^2 y^4 + 8x^2 r(x)^2 y^2 + \phi(x)^4 \right]^{\frac{1}{2}} \\
&\leq \left[ \left( (r(x) + xr'(x))y^2 + \frac{4x^2 r(x)}{r(x) + xr'(x)} \right)^2 \right]^{\frac{1}{2}} \\
&= (r(x) + xr'(x))y^2 + \frac{4x^2 r(x)}{r(x) + xr'(x)},
\end{aligned}
$$

which is the desired upper bound. $\qquad\square$

Now we give the proof of Theorem C.4, restated below for the sake of readability.

**Theorem C.4** (progressive sharpening). *Under the same setting as in Theorem C.2, let $t_a$ denote the obtained time step. Define the function $\tilde{\lambda} : \mathbb{R}_{>0} \to \mathbb{R}$ given by*

$$
\tilde{\lambda}(q) := \begin{cases} \left( 1 + \frac{\hat{r}(q)r'(\hat{r}(q))}{q} \right) \frac{2}{\eta} & \text{if } q \leq 1, \text{ and} \\ \frac{2}{\eta} & \text{otherwise.} \end{cases}
$$

*Then, the sequence $\left( \tilde{\lambda}\left( q_t \right) \right)_{t=0}^{\infty}$ is monotonically increasing. For any $t \geq t_a$, the sharpness at GD iterate $(x_t, y_t)$ closely follows the sequence $\left( \tilde{\lambda}\left( q_t \right) \right)_{t=0}^{\infty}$ satisfying the following:*

$$
\lambda_{\max}(\nabla^2 \mathcal{L}(x_t, y_t)) = \tilde{\lambda}\left( q_t \right) + \mathcal{O}_\phi(1).
$$

*Proof.* By Theorem C.2 and since $h$ is a bounded function by Lemma C.13, we have $s_t = 1 + \mathcal{O}_\phi(\eta)$ for any $t \geq t_a$. Note that $r(p_t) = (1 + \mathcal{O}_{\delta,\phi}(\eta))q_t \geq (1 + \mathcal{O}_{\delta,\phi}(\eta))q_0 \geq c_0 \geq r(z_0)$ for small step size $\eta$, where $z_0 = \sup\{\frac{zr'(z)}{r(z)} \geq -\frac{1}{2}\}$. Consequently, it holds that $1 + \frac{2p_t r'(p_t)}{r(p_t)} \geq 0$, and this implies $1 + \frac{p_t r'(p_t)}{r(p_t)} \geq \frac{1}{2}$. By Proposition C.14, we can bound the sharpness $\lambda_t := \lambda_{\max}(\nabla^2 \mathcal{L}(x_t, y_t))$ by

$$
\left( 1 + \frac{p_t r'(p_t)}{r(p_t)} \right) \frac{2r(p_t)}{\eta q_t} \leq \lambda_t \leq \left( 1 + \frac{p_t r'(p_t)}{r(p_t)} \right) \frac{2r(p_t)}{\eta q_t} + 4p_t^2 \left( 1 + \frac{p_t r'(p_t)}{r(p_t)} \right)^{-1}.
$$

By Lemma C.6, $|p_t| \leq 4$ for any time step $t$. Moreover, since $1 + \frac{p_t r'(p_t)}{r(p_t)} \geq \frac{1}{2}$ holds for any $t \geq t_a$ with small step size $\eta$, we have $4p_t^2 \left( 1 + \frac{p_t r'(p_t)}{r(p_t)} \right)^{-1} = \mathcal{O}_\phi(1)$. Hence, for any $t \geq t_a$, it holds that

$$
\lambda_t = \left( 1 + \frac{p_t r'(p_t)}{r(p_t)} \right) \frac{2r(p_t)}{\eta q_t} + \mathcal{O}_\phi(1). \tag{21}
$$

For any $t \geq t_a$, we have $s_t = 1 + \mathcal{O}_\phi(\eta)$ and this implies $|s_t^{-1} - 1| = \mathcal{O}_\phi(\eta)$ and $|r(p_t) - q_t| = \mathcal{O}_\phi(\eta)$. For any $0 < q < 1$, we have

$$
\begin{aligned}
\frac{d}{dq}\left( \frac{\hat{r}(q)r'(\hat{r}(q))}{q} \right) &= \frac{\hat{r}'(q)(r'(\hat{r}(q)) + \hat{r}(q)r''(\hat{r}(q)))}{q} - \frac{\hat{r}(q)r'(\hat{r}(q))}{q^2} \\
&= \frac{1}{q}\left( 1 + \frac{\hat{r}(q)r''(\hat{r}(q))}{r'(\hat{r}(q))} \right) - \frac{\hat{r}(q)r'(\hat{r}(q))}{q^2},
\end{aligned}
$$

so that

$$
\lim_{q \to 1^-}\left( \frac{d}{dq}\left( \frac{\hat{r}(q)r'(\hat{r}(q))}{q} \right) \right) = \lim_{p \to 0^+}\left( 1 + \frac{pr''(p)}{r'(p)} \right) = 2.
$$

Therefore, $\frac{d}{dq}\left(\frac{\hat{r}(q)r'(\hat{r}(q))}{q}\right)$ is bounded on $[\frac{1}{4}, 1)$ and Taylor's theorem gives

$$\left|\frac{p_t r'(p_t)}{r(p_t)} - \frac{\hat{r}(q_t)r'(\hat{r}(q_t))}{q_t}\right| \le \mathcal{O}_\phi(|r(p_t) - q_t|) \le \mathcal{O}_\phi(\eta).$$

for any time step $t$ with $q_t < 1$. Hence, if $q_t < 1$, we have the following bound:

$$\left|\tilde{\lambda}(q_t) - \left(1 + \frac{p_t r'(p_t)}{r(p_t)}\right)\frac{2r(p_t)}{\eta q_t}\right| \le \left|1 - s_t^{-1} + \frac{\hat{r}(q_t)r'(\hat{r}(q_t))}{q_t} - \frac{p_t r'(p_t)}{r(p_t)}\right|\frac{2}{\eta} + \mathcal{O}_\phi(1) = \mathcal{O}_\phi(1),$$

$$(22)$$

where we used $\frac{p_r r'(p_t)}{q_t} = \frac{p_t r'(p_t)}{r(p_t)}(1 + \mathcal{O}_\phi(\eta)) = \frac{p_t r'(p_t)}{r(p_t)} + \mathcal{O}_\phi(\eta)$, since $|p_t r'(p_t)| \le r(p_t)$ for any $t \ge t_a$. Now let $t$ be any given time step with $q_t \ge 1$. Then, $r(p_t) = 1 - \mathcal{O}_\phi(\eta)$, and since $r(z) = 1 + r''(0)z^2 + \mathcal{O}_\phi(z^4)$ for small $z$, we have $|p_t| = \mathcal{O}_\phi(\sqrt{\eta})$. Hence,

$$\left|\tilde{\lambda}(q_t) - \left(1 + \frac{p_t r'(p_t)}{r(p_t)}\right)\frac{2r(p_t)}{\eta q_t}\right| \le \left|1 - s_t^{-1} - \frac{p_t r'(p_t)}{r(p_t)}\right|\frac{2}{\eta} + \mathcal{O}_\phi(1) = \mathcal{O}_\phi(1), \qquad (23)$$

for any $t$ with $q_t \ge 1$, where we again used $\frac{p_r r'(p_t)}{q_t} = \frac{p_t r'(p_t)}{r(p_t)}(1 + \mathcal{O}_\phi(\eta)) = \frac{p_t r'(p_t)}{r(p_t)} + \mathcal{O}_\phi(\eta)$. By Eqs. (21), (22), and (23), we can conclude that for any $t \ge t_a$, we have

$$\left|\lambda_t - \tilde{\lambda}(q_t)\right| \le \mathcal{O}_\phi(1),$$

the desired bound.

Finally, we can easily check that the sequence $(\tilde{\lambda}(q_t))_{t=0}^{\infty}$ is monotonically increasing, since $z \mapsto \frac{z r'(z)}{r(z)}$ is a decreasing function by Assumption C.2 (ii) and the sequence $(q_t)$ is monotonically increasing. $\qquad\square$

