# OpenReview forum: "Trajectory Alignment: Understanding the Edge of Stability Phenomenon via Bifurcation Theory"
_NeurIPS.cc/2023/Conference — NeurIPS 2023 poster_

### Official Review · Reviewer_Qm1U · 2023-07-05

**Soundness:** 4 excellent
**Presentation:** 4 excellent
**Contribution:** 3 good
**Rating:** 7
**Confidence:** 4

**Summary:**

This paper discovers and studies a new phenomenon that occurs when training neural nets on 1 data point with large enough learning rates.    The phenomenon concerns the end-stage evolution of the tuple (residual, sharpness), where the residual (defined as "prediction - target") is a scalar because there is only 1 data point.   If one fixes the sharpness and considers the evolution of the residual, and if one linearizes the network, then the dynamical system that governs the evolution of the residual has a stable period-2 orbit whose value depends on the fixed sharpness, i.e. residual $\to \pm$  function(sharpness).

The main finding of the paper is that the residual and sharpness rapidly evolve onto the curve "residual $=\pm$  function(sharpness)" and then remain on this curve for the end stage of training.  This finding is made empirically on some real architectures and is also proved rigorously in two highly simplified settings.  This is interesting because it is a richer characterization of the training dynamics than has been previously known (previous knowledge is limited to the behavior of the sharpness and says nothing about the residual).


**Strengths:**

- I think the identification of the trajectory alignment phenomenon is a solid contribution.
- It's great that the authors can rigorously establish this phenomenon in several simplified settings


**Weaknesses:**

- The main weakness is that the trajectory alignment phenomenon seems mostly limited to the setting of training on one data point.  The authors do briefly try to extend it to the setting of multiple data points, but the experimental evidence is not highly convincing (especially the $\ell_\infty$ plot) and I do not really see why the phenomenon _would_ be true.  For example, I do not see why the generalized sharpness metric (which is kind of like trace(Hessian) / n) would be the right generalization.

- Another weakness is that the EOS setting seems different than the one from Cohen et al '21.  In that paper, training with gradient flow makes the sharpness continually increase, whereas my understanding is that in this paper, training with gradient flow does not make the sharpness continually increase.  Is my understanding correct?  If so, I would prefer if the paper were clearer about the differences from Cohen et al '21, so as not to add confusion to the literature.


**Questions:**

- Could the authors please cite these two papers in the related work section?  https://arxiv.org/abs/1807.05031, https://arxiv.org/abs/2002.09572.  These papers gave evidence prior to Cohen et al '21, that the step size affects the sharpness along the optimization trajectory.

- I wonder if the reason why the trajectory alignment phenomenon occurs on wide networks but not narrow ones is that the linearization of the network is more accurate for wide networks.  Therefore, I wonder if one could also see trajectory alignment on narrow nets if one adopted the scaling trick from Chizat and Bach (https://arxiv.org/abs/1812.07956) which makes any network closer to its linearization.

- I believe Lemma 2.1(ii) has a typo, it should be r_hat(q)



**Limitations:**

Please see the second point in the 'weaknesses' section.

---

> ### Author Rebuttal · Authors · 2023-08-09
>
> We greatly appreciate your recognition and constructive feedback on our work. We have carefully considered your comments to enhance our manuscript. We have also conducted additional experiments, and the associated figures can be found in the attached PDF file in the "global" response (Response to All Reviewers). When referencing figures from the additional experiments, we will use the format "Figure (number) - Additional Experiments" to distinguish them from the figures in the manuscript.
>
> Regarding the specific comments in the review, we would like to address them as follows:
>
> >(W1) The trajectory alignment phenomenon seems mostly limited to the setting of training on one data point.
>
> We understand the reviewer's concerns regarding the multiple data points setting. Specifically, we initially claimed the trajectory alignment phenomenon throughout all settings, including the mean, $\ell_1$, $\ell_2$, and $\ell_\infty$ plot. In the revised manuscript, we will tone down and claim that trajectory alignment phenomenon is observed especially at the "mean" plot across wide range of setting.
>
> We want to emphasize that the trajectory alignment is clearly observed in the mean plot. Our experiments in Appendix A.4 depicted in Figures 13-16 consistently demonstrate trajectory alignment for large-width FC networks trained on CIFAR-10. We also conducted additional experiments on CNN trained on CIFAR-10, presented in Figure 1 - Additional Experiments, and the results also show trajectory alignment phenomenon. The empirical evidence consistently supports this phenomenon, although explaining it theoretically remains outside the scope of our work.
>
> Regarding the choice of (p,q) in the multiple data points setting, we choose (p,q) to set (p,q) = (0,1) for parameters corresponding to global minima and sharpness (defined as maximum eigenvalue of loss Hessian, rather than trace(Hessian) / n) equal to 2/(step size). As a conjecture, other careful choices of (p,q) may also exhibit trajectory alignment, warranting further investigation.
>
> Furthermore, we present an extension of Eq.(3) to the multiple data settings, assuming a network can be well approximated by its linearized model, yielding insights into the relationship between residual and sharpness. Specifically, under the linear approximation $f(X;\Theta_{t+1})\approx f(X;\Theta_t)+Df(X;\Theta_t)(\Theta_{t+1}-\Theta_t)$, we obtain
> $$f(X;\Theta_{t+1})-y \approx (f(X; \Theta_t)-y) - \frac{\eta}{n} K(X,X;\Theta_t) \ell'(f(X; \Theta_t)-y),$$
> where $K(X,X;\Theta)\in\mathbb{R}^{n\times n}$ is a Gram matrix with (i,j)-th entry given by $\langle \nabla_{\Theta} f(x_i;\Theta),\nabla_{\Theta}f(x_j;\Theta)\rangle$, also known as NTK [Jacot2018]. In case of n=1, $K(x_1,x_1;\Theta)=\lVert\nabla_{\Theta} f(x_1;\Theta)\rVert_2^2$ and this matches with Eq.(3). Understanding the trajectory alignment phenomenon in this context remains a topic for future work.
>
> >(W2) In [Cohen2021], training with gradient flow makes the sharpness continually increase, whereas in this paper, training with gradient flow does not make the sharpness continually increase.
>
> As the reviewer pointed out, our findings on progressive sharpening differ from those in [Cohen2021]. While [Cohen2021] observed an increase in sharpness during training in the gradient flow regime, our work reveals that sharpness increases while exhibiting oscillations in the EoS regime. This distinction could indeed be attributed to differences in the loss functions used in two studies. In the revised version, we will make this point clearer and provide a detailed discussion on the differences between our work and the original literature.
>
> >(Q1) Could the authors please cite [Jastrzebski2018] and [Jastrzebski2020] in the related work section?
>
> We acknowledge that [Jastrzebski2018] and [Jastrzebski2020] provided evidence of the step size's impact on sharpness along the optimization trajectory before [Cohen2021]. In the revised manuscript, we will include proper citations to these papers in the related work section.
>
> >(Q2) I wonder if one could also see trajectory alignment on narrow nets if one adopted the scaling trick from [Chizat2018] which makes any network closer to its linearization.
>
> Thank you for your important question and insightful suggestion. We also share the belief that the trajectory alignment phenomenon occurs on wide networks due to their close approximation to their linearized models, as we adopted a linear approximation of a network to obtain Eq.(3).
>
> To further investigate the relationship between the trajectory alignment and network linearization, we conducted additional experiments on narrow networks (plots are not provided due to the space constraint). As the reviewer suggested, we adopted the scaling trick from [Chizat2018] to bring the networks closer to their linearization. The empirical evidence clearly demonstrates that increasing the scaling factor induces the trajectory alignment phenomenon, even for narrow networks.
>
> We will include a detailed description of the additional experiments and their results in the revised manuscript. Additionally, we will provide a comprehensive discussion about the relation between the linearization of the network and the trajectory alignment phenomenon.
>
> Lastly, we further note that our theoretical results do not rely on linear approximation of networks; instead, we rigorously analyze the exact dynamics, making our theory applicable to a two-layer linear network with an arbitrary width.
>
> >(Q3) I believe Lemma 2.1(ii) has a typo, it should be $\hat{r}(q)$.
>
> Thank you for pointing out the typo. We apologize for the oversight, and we will correct Lemma 2.1(ii) in the revised manuscript.
>
> ---
> __Reference__
>
> [Chizat2018] https://arxiv.org/abs/1812.07956
>
> [Cohen2021] https://openreview.net/forum?id=jh-rTtvkGeM
>
> [Jastrzebski2018] https://arxiv.org/abs/1807.05031
>
> [Jastrzebski2020] https://arxiv.org/abs/2002.09572

---

> > ### Comment · Reviewer_Qm1U · 2023-08-17
> > **multiple data points**
> >
> > It may be worth trying to see whether the trajectory alignment holds for multiple datapoints if you consider (1) the largest eigenvalue of the NTK, and (2) the projection of the residuals onto the largest eigenvector of the NTK?

---

> > > ### Author Response · Authors · 2023-08-18
> > >
> > > We greatly appreciate your insightful suggestions. Based on your recommendations, we conducted additional experiments utilizing an alternative reparameterization. In this approach, we considered a canonical mapping given by:
> > > $$(p,q) = \left(v_1(X;\Theta)^\top (f(X; \Theta)-y), \frac{2n}{\eta \lambda_1(X; \Theta)}\right).$$
> > > Here, $\lambda_1(X;\Theta)$ and $v_1(X;\Theta)$ represent the largest eigenvalue and its corresponding normalized eigenvector of the NTK matrix $K(X, X; \Theta)\in \mathbb{R}^{n\times n}$, respectively. We observed that even with this reparameterization, the trajectory alignment phenomenon persists across different settings, including varying network width and depth trained on CIFAR-10. Notably, trajectories align along a curve qualitatively similar to the mean plot in Figure 14(d) of the appendix and converge to a point close to $(p,q) = (1,0)$.
> > >
> > > To understand this phenomenon, we consider the one-step update of the residual and approximated it as follows:
> > > $$f(X; \Theta_{t+1}) - y \approx (f(X; \Theta_t) - y) - \frac{\eta}{n} K(X, X; \Theta_t) \ell'(f(X; \Theta_t) - y).$$
> > > Assuming $v_1(X;\Theta_{t+1}) \approx v_1(X; \Theta_t)$, we can approximate the one-step update of $p_t$ as:
> > > $$p_{t+1} \approx p_t - \frac{2v_1(X;\Theta_t)^\top \ell'(f(X; \Theta_t) - y)}{q_t}.$$
> > > An interesting question arises concerning whether the term $v_1(X;\Theta_t)^\top \ell'(f(X; \Theta_t) - y)$ can be represented as a function of $p_t$. Under a stronger assumption, where $v_1(X; \Theta_t)$ is approximated by a fixed vector $v_1$ and the residual $(f(X; \Theta_t)-y)$ aligns with $v_1$, we find that $v_1(X;\Theta_t)^\top \ell'(f(X; \Theta_t)-y) \approx g(p_t)$, where $g(p) := v_1^\top \ell'(pv_1)$ for each $p\in \mathbb{R}$. This assumption aligns with the observation that the oscillation direction in the EoS regime corresponds to the eigenvector of the largest eigenvalue. This insight leads us to the conclusion that $p_{t+1} \approx p_t - \frac{2g(p_t)}{q_t}$, enabling us to potentially conduct bifurcation analysis on this map, analogous to the single data point case outlined in Section 2.3 of our manuscript. We deeply value your comment, as we believe this approach will pave the way for extending our theory to the multiple data setting in future work.

---

### Official Review · Reviewer_6YQZ · 2023-07-06

**Soundness:** 2 fair
**Presentation:** 2 fair
**Contribution:** 2 fair
**Rating:** 3
**Confidence:** 2

**Summary:**

This paper studies the Edge of Stability phenomenon in training neural networks. The authors demonstrate that, on some reparameterization of the loss function, for a small enough stepsize, the trajectories of gradient descent with different initializations align on a bifurcation diagram. They show this for two classes of functions:
(1) Two-layer fully connected linear neural networks trained on a single data point with a convex and Lipschitz loss.
(2) Single-neuron nonlinear network trained on a single data point with the squared loss.

**Strengths:**

1. The analysis presented by the authors is novel, and it improves upon prior work in some settings (e.g. tightening the gap on sharpness upper and lower bounds from $\mathcal{O}(\eta)$ to $\mathcal{O}(\eta^3)$ for two-layer linear neural networks).
2. The paper imports a lot of ideas from the literature on control theory to studying the optimization of neural nets. This might be useful more generally.

**Weaknesses:**

1. Why is the whole "trajectory alignment" phenomenon is interesting in the first place? What new things does it tell us?
2. It seems very obvious that for high-dimensional, deep networks, gradient descent doesn't converge to the same minimizers for different initializations. Is this whole phenomenon a product of small $d$ and toy examples? You fix the Xavier initialization for all experiments, and as far as I can tell the widest network used has width 256. This is a far cry from modern neural networks.
3. The conditions you have on the initialization in Thm 4.2 are very restrictive. Do they hold in practice?

**Questions:**

See the weaknesses section.

**Limitations:**

N/A.

---

> ### Author Rebuttal · Authors · 2023-08-09
>
> We appreciate your valuable feedback on our paper. We have carefully considered your comments and improved our manuscript accordingly. We have also conducted additional experiments, with corresponding figures available in the attached PDF files in the "global" response (Response to All Reviewers). When citing figures from these additional experiments, we will use the format "Figure (number) - Additional Experiments" to differentiate them from the figures in our manuscript. We hope that the reviewer will find our response satisfactory. We kindly ask for your consideration in adjusting the review score, as we believe we have effectively addressed your primary concerns.
>
> Regarding the specific review comments, we would like to respond as follows:
>
> >(W1) Why is the whole "trajectory alignment" phenomenon interesting in the first place?
>
> The trajectory alignment phenomenon holds significant interest as it sheds light on the training dynamics of neural network optimization using GD with large step sizes, particularly in the Edge of Stability (EoS) regime. This phenomenon is intriguing because prior understanding in the EoS regime primarily focused on the behavior of sharpness alone. Our work introduces a novel observation where reparameterized GD trajectories consistently align on a specific bifurcation diagram regardless of their initializations. This discovery is notable due to the intricate and non-convex nature of neural network optimization, where the algorithm trajectory is heavily influenced by initialization choices. Understanding trajectory alignment has the potential to advance our fundamental understanding of optimization processes in neural networks, offering insights into improved optimization strategies and landscape properties.
>
> Furthermore, the existing theoretical understanding of the EoS phenomenon remains limited, with few works rigorously examining the limiting sharpness, even in simplified models. Our work goes beyond this by providing a rigorous characterization of the oscillating dynamics and the evolution of sharpness for a broader range of networks.
>
> >(W2) Is this whole phenomenon a product of small $d$ and toy examples?
>
> We would like to clarify that the trajectory alignment phenomenon is not a product of small data dimension $d$ or toy examples. Our empirical observations consistently demonstrate its presence across higher data dimensions and wider networks. In fact, this phenomenon becomes more pronounced as network width increases.
>
> To illustrate this, in Figure 3, we show that the trajectory alignment phenomenon occurs in wide networks with a width of 256, but not as clearly in narrower networks with a width of 64. Furthermore, we conducted experiments on fully connected (FC) networks trained on the CIFAR-10 dataset, employing widths of 512 and 1024. The results, displayed in Figures 14, 15, and 16 in Appendix A.4, indicate the trajectory alignment phenomenon in large-width FC networks trained on a real-world dataset.
>
> For additional validation, we conducted additional experiments on convolutional neural networks (CNNs) trained on CIFAR-10, and the corresponding results are presented in Figure 1 - Additional Experiments. We observe the trajectory alignment phenomenon in wide convolutional networks trained on a real-world dataset. Therefore, the trajectory alignment phenomenon is not confined to small dimensions or toy examples.
>
> The use of Xavier initialization was maintained consistently across experiments to ensure fair comparisons and minimize potential biases originating from initialization. It's important to note that the presence of the trajectory alignment phenomenon is not related to the initialization method. Rather, it emerges as a characteristic behavior during optimization using GD with "large" step sizes.
>
> In summary, our empirical evidence robustly demonstrates that the trajectory alignment phenomenon occurs in various experimental settings that include wide FC networks, CNNs (with widths of up to 1024) trained on a real-world dataset, and high data dimensions such as $d=32\times 32\times 3=3072$.
>
> >(W3) The conditions you have on the initialization in Theorem 4.2 are very restrictive.
>
> We want to clarify that the conditions on the initialization in Theorem 4.2 are not as restrictive as they might seem. For simplicity, let us focus on the case of log-cosh loss.
>
> In Theorem 4.2, the conditions on the initialization $(p_0, q_0)$ are satisfied when $\lvert p_0 \rvert \le 1$ and $q_0 \in (\frac{3}{4}, 1)$. Although we assumed the bound $\lvert p_0 \rvert \le 1$ for simplicity, our proof extends to the assumption $\lvert p_0 \rvert \le K$ for any positive constant $K$, modulo some changes in numerical constants. Hence, the only essential assumption is that $q_0 \in (\frac{3}{4}, 1)$.  Notably, Theorem 4.1 already covers the range $q_0 > 1$, leaving out only the case of $0 < q_0 < \frac{3}{4}$.
>
> Empirically, we have observed that when $q_0$ satisfies $0 < q_0 < \frac{3}{4}$, $q_t$ increases during early dynamics and eventually reaches the regime $\frac{3}{4} < q_t < 1$, which is covered by our theory. We conducted additional experiments to support this observation. The results, depicted in Figure 2 - Additional Experiments, demonstrate that for small step sizes, GD trajectories initialized with $q_0<1$ eventually reach the set $S_A := [-1, 1]\times (\frac{3}{4},1)$ in most cases. This further emphasizes that the assumption in Theorem 4.2 encompasses a wide range of scenarios.

---

### Official Review · Reviewer_ft7H · 2023-07-27

**Soundness:** 4 excellent
**Presentation:** 4 excellent
**Contribution:** 3 good
**Rating:** 6
**Confidence:** 3

**Summary:**

This paper investigates a trajectory alignment phenomenon during the edge of stability (EoS) regime in gradient descent (GD): the trajectories from different GD dynamics tend to align on a bifurcation diagram. The paper first empirically observes the trajectory alignment in deep neural network training in a canonically reparametrized space that can reveal the bifurcation. With a problem setting of a two-layer linear neural network and single neuron nonlinear network training on a single data point, the alignment is theoretically analyzed, explaining the GD trajectories in the gradient flow regime, two phases of the EoS regime, and the progressive sharpening.

**Strengths:**

- Based on simple models, this paper clearly characterizes the gradient flow regime, two phases of the EoS regime, and progressive sharpening using the bifurcation theory. This seems a novel and original way to analyze the phenomenon. The introduction of canonical reparametrization greatly helps capture the crux of the EoS phenomenon. This work will provide useful insight to study the dynamics in the EoS regime of neural network training, which is recently getting a huge interest in the community.
- This paper is in general very well written. The theories seem to be developed in a mathematically sound manner.


**Weaknesses:**

- The claim that the trajectory alignment on a bifurcation diagram is independent of network architecture and training data seems somewhat excessive. It does not corroborate well with some experimental results in the main text and appendix (see Figures 5, 10, 13, 14, 15), which shows that the degree of alignment depends greatly on the width size and the number of data. Also, the analyses are only performed for specific network architecture and loss function combinations.
- The theory does not cover some widely used losses, e.g., the cross-entropy loss. Is the loss function necessary to be symmetric for this alignment to occur?


**Questions:**

In addition to the questions posed in the **Weaknesses** section, I have the following questions:
- Is any further analysis possible for the period-4 (or more) oscillation shown in Figure 1 (b) and (c)?
- Can the EoS for GD with momentum also be explained by this theory?
- How do the experimental conditions differ for each color in Figure 2?
- In Figure 5 of the appendix, why there seems to be more alignment to the bifurcation diagram for the mean and $L_\infty$-norm cases than the $L_1$, $L_2$-norm cases?
- What happens if $q_0 < \frac{1}{2}$ in line 221?
- How is the sharpness anticipation for progressive sharpening in **Theorem 4.4** accurate for neural networks larger than those in Figure 1 (a)?
- Can the progressive sharpening also be interpreted in the bifurcation diagram?


**Limitations:**

Some limitations of the current results are discussed in **Conclusion**. For other possible limitations, please refer to the weaknesses and questions above.

---

> ### Author Rebuttal · Authors · 2023-08-09
>
> Thank you for recognizing our work and providing valuable feedback. We've carefully incorporated your suggestions to improve our manuscript. Additional experiments and figures can be found in the attached PDF files in  the "global" response, referenced as "Figure (number) - Additional Experiments."
>
> Here's how we've addressed your specific comments:
>
> >(W1) The claim that the trajectory alignment on a bifurcation diagram is independent of network architecture and training data seems excessive.
>
> We acknowledge that our initial claim was excessive. We will revise our claim to clarify that reparameterized GD trajectories align on a specific bifurcation diagram __independent of initialization__, when the EoS phenomenon occurs.
>
> >(W2) The theory does not cover cross-entropy loss. Is the loss function necessary to be symmetric?
>
> We want to clarify that cross-entropy loss falls outside the scope of our work's main focus. Under cross-entropy loss and GD with a large step size, the sharpness typically decreases towards the end of training, going far below 2/(step size). This phenomenon has been empirically observed and theoretically investigated by [Cohen2021] and [Wu2023]. In contrast, our work concentrates on scenarios where sharpness saturates near 2/(step size), and the trajectory alignment phenomenon occurs in this setting.
>
> The symmetry assumption on loss is unnecessary for the trajectory alignment phenomenon. While we assume a symmetric loss in our theoretical analysis for simplicity, the phenomenon can also arise in an asymmetric setting. As demonstrated in Figure 1(c), we still observe trajectory alignment on an asymmetric bifurcation diagram when using ELU activation.
>
> >(Q1) period-4 (or more) oscillation shown in Figure 1 (b) and (c)
>
> Analyzing higher-order oscillations goes beyond our current scope. We want to clarify that we mainly focus on convex Lipschitz losses, where period-4 (or more) oscillations do not occur as discussed in Line 154. The occurrence of higher-order oscillations depends on the specific model and remains an interesting future direction.
>
> >(Q2) EoS for GD with momentum
>
> Explaining the EoS for GD with momentum using our current theory is beyond the scope of our work. Analyzing the dynamics of GD with momentum is technically challenging, and to the best of our knowledge, there is no existing work that theoretically investigates EoS for this variant of GD.
>
> We conducted additional experiments using a toy model $(x,y)\mapsto \log(\cosh(xy))$ to examine the trajectories of GD with both Polyak and Nesterov momentum (plots are not provided due to the space constraint). We observe that the behavior of GD with momentum is qualitatively different from classical GD. For example, GD with Polyak momentum under different initializations does not necessarily converge to the minimum of same sharpness, but we do observe some level of consistency in Nesterov. These observations highlight the need for further investigation to understand GD with momentum in the EoS regime.
>
> >(Q3) color used in Figure 2
>
> In the figure, each color represents an independent run of GD with different initializations. In the revised manuscript, we will enhance the figure captions to provide a clear explanation of the color used.
>
> >(Q4) Why there seems to be more alignment for the mean and $L_\infty$-norm cases than the $L_1$,$L_2$-norm cases?
>
> In Figure 5 of Appendix, the trajectory alignment phenomenon appears more pronounced in the mean plot than the $L_\infty$ plot. However, the underlying reasons remain currently unclear. In our experiments, we explored different reparameterization options, including mean, $L_1$, $L_2$, and $L_\infty$ norms. Notably, the consistent occurrence of trajectory alignment in the mean plot across various settings is depicted in Figures 11-16 of Appendix. Understanding this phenomenon would be an intriguing future work.
>
> >(Q5) What happens if q_0 < 1/2 in line 221?
>
> In the case where 0 < q_0 < 1/2, the sequence (q_t) may not exhibit a monotonic increase. However, empirical observations indicate that q_t increase over the long run and eventually enters a regime where q_t > 1/2.
>
> We conducted additional experiments to further investigate the behavior of GD trajectories initialized with q_0 < 1/2 under log-cosh loss. The results, presented in Figure 2 - Additional Experiments, demonstrate that in most cases, these trajectories eventually reach the set $S_A:=[-1,1]\times (\frac{3}{4}, 1)$. Note that the assumptions in Theorem 4.2 is satisfied when $(p_0, q_0)\in S_A$. This indicates that the trajectory alignment phenomenon still occurs (in the long run) when q_0 < 1/2.
>
> >(Q6) Is the sharpness anticipation in Theorem 4.4 accurate for large networks?
>
> Yes, the sharpness anticipation is accurate for large networks. We conducted additional experiments on two-layer linear networks with widths of 256 and 1024. The results, illustrated in Figure 3 - Additional Experiments, provide strong evidence that the sharpness prediction derived from Theorem 4.4 effectively approximates the actual sharpness values.
>
> >(Q7) Can progressive sharpening be interpreted in the bifurcation diagram?
>
> Certainly, the progressive sharpening phenomenon can indeed be interpreted within the context of the bifurcation diagram. The relationship between progressive sharpening and the bifurcation diagram is established through our theorems: Theorem 4.4 establishes that as the GD trajectory closely follows the bifurcation diagram, the sharpness increases during the training process. In particular, $\tilde{\lambda}(q)$ represents the sharpness value on the bifurcation diagram. Consequently, as the sequence $(\tilde{\lambda}(q_t))_{t=0}^\infty$ increases, the sharpness also increases since $\tilde{\lambda}(q_t)$ well-approximates the actual sharpness when the trajectory alignment phenomenon occurs.
>
> ---
> __Reference__
>
> [Cohen2021] https://openreview.net/forum?id=jh-rTtvkGeM
>
> [Wu2023] https://arxiv.org/abs/2305.11788

---

> > ### Comment · Reviewer_ft7H · 2023-08-15
> >
> > I am grateful to the authors who faithfully answered my concerns and questions. Several questions have been answered, but some unresolved issues require further investigation before the theory is convincing in more general settings. So I will keep my score.

---

> > > ### Author Response · Authors · 2023-08-16
> > >
> > > Thank you for taking the time to review our response. If there are any remaining concerns or unresolved issues, please let us know. We will do our best to address the concerns and will proceed with the necessary revisions accordingly.

---

> > > > ### Comment · Reviewer_ft7H · 2023-08-16
> > > >
> > > > In my opinion, a more sophisticated analysis is necessary for the case of training on multiple data points. However, this expansion might be difficult due to the short discussion period.

---

> > > > > ### Author Response · Authors · 2023-08-17
> > > > >
> > > > > We appreciate your feedback and recognize the need for more rigorous analysis in the case of training on multiple data points. While our current theory does not cover this setting, we have taken an initial step to explore it by extending Eq.(3) to the multiple data setting. This extension assumes that a network can be well approximated by its linearized model, providing insights into the relationship between residuals and sharpness.
> > > > >
> > > > > Specifically, under the linear approximation $f(X; \Theta_{t+1}) \approx f(X; \Theta_t) + Df(X; \Theta_t) (\Theta_{t+1} - \Theta_t)$, we can obtain
> > > > > $$f(X; \Theta_{t+1}) - y \approx (f(X; \Theta_t) - y) - \frac{\eta}{n} K(X, X; \Theta_t) \ell'(f(X; \Theta_t) - y),$$
> > > > > where $K(X, X; \Theta)\in \mathbb{R}^{n\times n}$ is a Gram matrix where (i,j)-th entry is given by $\langle \nabla_{\Theta} f(x_i; \Theta), \nabla_{\Theta} f(x_j; \Theta) \rangle$, also known as NTK [Jacot2018]. In the case of n=1, $K(x_1, x_1; \Theta) = \lVert \nabla_{\Theta} f(x_1; \Theta) \rVert_2^2$ and the expression matches Eq.(3) in our manuscript. Understanding the trajectory alignment phenomenon in this context remains a topic for future work.
> > > > >
> > > > > ---
> > > > >
> > > > > __Reference__
> > > > >
> > > > > [Jacot2018] https://arxiv.org/abs/1806.07572

---

> > > > > > ### Comment · Reviewer_ft7H · 2023-08-18
> > > > > >
> > > > > > I appreciate the response. The [**Reviewer Qm1U**'s additional comment](https://openreview.net/forum?id=PnJaA0A8Lr&noteId=RiUngFp3S7) may be worth studying further.

---

> > > > > > > ### Author Response · Authors · 2023-08-18
> > > > > > >
> > > > > > > We appreciate your suggestion. Following Reviewer Qm1U's [additional comment](https://openreview.net/forum?id=PnJaA0A8Lr&noteId=RiUngFp3S7), we conducted further experiments and observed that the trajectory alignment phenomenon also persists in this reparameterization. For more details, please refer to [our response](https://openreview.net/forum?id=PnJaA0A8Lr&noteId=GJGePFLQNp) to Reviewer Qm1U's comment.

---

### Official Review · Reviewer_mAcJ · 2023-07-31

**Soundness:** 2 fair
**Presentation:** 2 fair
**Contribution:** 2 fair
**Rating:** 5
**Confidence:** 3

**Summary:**

This paper studies the edge of stability phenomenon and proposes a "trajectory alignment" phenomenon, which states that with different learning rates and different initializations, if one plots the residuals and the inverse squared gradient norms, these lie on a fixed curve. The paper supports this theory with experiments on small networks with log-cosh loss and rigorously proves it for a two-layer linear net and a single neuron nonlinear net, both trained on a single datapoint.

**Strengths:**

- While the bifurcation analysis of GD at EoS is not new, this paper takes this analysis further than previous papers and rigorously characterizes the limiting curve for toy models trained on a single datapoint.

- The paper also provides fine-grained analyses for each of the toy cases it considers including precise predictions for the limiting sharpness which improves over previous work.

**Weaknesses:**

- The abstract and the introduction state that GD trajectories (after a proper reparameterization) align on a specific bifurcation diagram. However, the (p,q) variables in the paper are only a reparameterization in the toy case when $\theta$ has dimension 2. In the general case (e.g. in section 3) it is not fair to call this a reparameterization as the variables (p,q) do not determine the GD trajectory.

- The abstract claims that the paper establishes the trajectory alignment phenomenon through empirical studies. However, the experiments in the paper are extremely limited. The experiments in figure 3 use a single datapoint equal to $e_1 \in \mathbb{R}^{10}$ and the non-standard log-cosh loss function, while the experiments in figure 4 use 10 datapoints sampled from a Gaussian distribution in $\mathbb{R}^{10}$. While these settings are certainly more difficult to analyze than the settings in the rest of the paper, they are still very much toy settings. It would be interesting to see if similar phenomena occur on more realistic datasets/models with more realistic loss functions (e.g. square loss or cross entropy loss).

- The paper states a couple of times that the bifurcation diagram is determined solely by the loss function, independent of the network architecture, training data, and step size. However, this strong claim is not demonstrated outside of the toy setting of a single datapoint. Figure 4 does suggest that in the general case it may be independent of the learning rate but there is no evidence for the full claim.

- It seems that all of the results in terms of the "canonical parameterization" in eq (2) are specific to assumption 2.4. In particular, the results in section 5 for square loss require defining a different parameterization (definition 5.2) which challenges the generality of the results in the rest of the paper. As the original EoS phenomenon as described in Cohen et al. was observed for square/logistic/cross entropy losses, it is unclear whether the phenomenon being studied is the same as the phenomenon observed in Cohen et al.

Minor Comments:
- There seems to be a clear separation of colors in Figures 2, 3, 4 but as far as I can tell these colors are never defined. Also, increasing the size of the datapoints might help with visibility, especially in Figure 3.
- When discussing previous EoS papers with bifurcation diagrams, the paper should cite Chen et al. "Beyond the Edge of Stability via Two-step Gradient Updates" (it is in the list of references but not in the related work)

**Questions:**

- Why did the authors chose to parameterize $q$ in terms of the inverse squared gradient norm instead of simply the squared gradient norm? At first glance, the reciprocal $1/q$ seems like an easier object to work with?
- How is the analysis in section 5 for square loss different from the analyses in the rest of the paper?

**Limitations:**

The authors have adequately addressed the limitations of this work.

---

> ### Author Rebuttal · Authors · 2023-08-09
>
> We appreciate your recognition and feedback on our work. We've integrated your suggestions to enhance our manuscript. For additional experiments and figures, please refer to the attached PDF files in the "global" response, which we reference as "Figure (number) - Additional Experiments."
>
> We are addressing your specific comments as follows:
>
> >(W1) reparameterization do not determine the GD trajectory
>
> We understand your concern that the reparameterization may not fully determine the GD trajectory. Our use of the the term "reparameterization" was intended to encompass not only bijections between coordinates but also surjections, such as projections to a subspace. We will clarify the term "reparameterization" in the revised manuscript. If you have suggestions for alternative terms that better represent this concept, we would appreciate your input and consider it for the revision.
>
> >(W2) realistic datasets/models with realistic loss functions
>
> We have indeed observed the trajectory alignment phenomenon in more realistic settings. In our experiments, we trained large width FC networks on CIFAR-10 and observed trajectory alignment, as illustrated in Appendix A.4 and Figures 13-16. We further conducted additional experiments on CNNs trained on CIFAR-10, as shown in Figure 1 - Additional Experiments. The results consistently demonstrate the trajectory alignment phenomenon when training wide CNNs on a real-world dataset.
>
> Regarding the loss functions, extending the results to squared loss is an important future direction. Our work in Section 5 provides an initial step towards understanding trajectory alignment that arises on nonlinear networks trained with squared loss.
>
> However, we want to clarify that cross-entropy loss in fact falls outside the scope of our work's focus. Under cross-entropy loss and GD with a large step size, the sharpness typically decreases towards the end of training, going far below 2/(step size). This phenomenon has been empirically observed and theoretically investigated by [Cohen2021] and [Wu2023]. In contrast, our work concentrates on scenarios where sharpness saturates near 2/(step size), and the trajectory alignment phenomenon occurs in this setting.
>
> >(W3) The claim that bifurcation diagram is independent of the network, data, and step size is too strong.
>
> We acknowledge the reviewer's concern on our initial claim. We will revise our claim to clarify that different GD trajectories (after a proper reparameterization) align on a specific bifurcation diagram __independent of initialization__.
>
> >(W4) It seems that all of the results in terms of the "canonical parameterization" ...  observed in [Cohen2021].
>
> Indeed, the canonical reparameterization applies to a class of convex Lipschitz losses, hence excluding square/logistic/cross-entropy losses. Extending our finding to squared loss is an important future direction. However, cross-entropy loss falls outside the scope of our interest, as explained in our response to (W2).
>
> Furthermore, we want to clarify that our primary focus is to study the EoS phenomenon observed by [Cohen2021]. This phenomenon refers to the oscillating behavior exhibited by GD with a large step size, causing the sharpness to saturate near 2/(step size). Our work is centered around characterizing this phenomenon, as the trajectory alignment phenomenon captures the oscillating dynamics and the sharpness approaching 2/(step size).
>
> >(W5) colors in figures
>
> The distinct colors used in the figures indicate independent runs of GD with varying initializations. In the revised manuscript, we will enhance the figure captions to provide a explanation of the color scheme and increase the size of the markers in our plots for better visibility.
>
> >(W6) The paper should cite [Chen2023]
>
> We appreciate your suggestion regarding [Chen2023]. We will include discussions on their study in the related works section of our revised manuscript. Specifically, [Chen2023] observed that in specific settings, GD with a large step size converges to its period-2 orbit, which is a fixed point of a two-step update. In contrast, both [Zhu2023] and our work focus on the phenomenon where GD exhibits oscillatory behavior, eventually converging to a fixed point of the one-step update. Indeed, [Chen2023] studied the EoS phenomenon but did not capture that GD trajectories exhibit bifurcation-like oscillation. We will ensure that these differences are properly discussed in the revised manuscript.
>
> >(Q1) Why did the authors chose $q$, but not $1/q$?
>
> We chose the inverse squared gradient norm for $q$ mainly for better visualization. This choice allows us to represent the trajectory alignment phenomenon more clearly within the range of $q\in (0,1)$. Although using the reciprocal $1/q$ is an option, working directly with $q$ offered clearer and more interpretable visualizations for our experiments and analysis.
>
> >(Q2) How is the analysis in section 5 for square loss different from the rest?
>
> The theoretical analysis in Section 5 for squared loss is similar to the analysis in Section 4. This similarity arises from the resemblance between Eq.(7) and Eq.(6).
>
> However, there are some differences in the specifics. For example, Theorem C.3 proves that the limiting sharpness is $\frac{2}{\eta}-\frac{2}{\lvert r''(0) \rvert} + O(\eta)$, while Theorem 4.3 proves that the limiting sharpness is $\frac{2}{\eta} - \frac{\eta}{\lvert r''(0) \rvert} + O(\eta^3)$. The error term is $O(\eta)$ in Theorem C.3 and $O(\eta^3)$ in Theorem 4.3. This difference arises because in Eq. (7), the one-step update of $q$ is $q_{t+1} = (1+O(\eta))q_t$, while in Eq. (6), it is $q_{t+1} = (1+O(\eta^2)) q_t$. Nonetheless, the main ideas of the theoretical analysis used in both sections remain the same.
>
> ___
> __Reference__
>
> [Chen2023] https://openreview.net/forum?id=AvwlrX9AQr
>
> [Cohen2021] https://openreview.net/forum?id=jh-rTtvkGeM
>
> [Wu2023] https://arxiv.org/abs/2305.11788
>
> [Zhu2023] https://openreview.net/forum?id=p7EagBsMAEO

---

> > ### Comment · Reviewer_mAcJ · 2023-08-15
> >
> > Thank you for the clarifications. I have updated my score accordingly.
> >
> > Regarding the term “reparameterization,” I agree it doesn’t have to be a bijection (e.g. you can parameterize a matrix by the outer product of two vectors to enforce a rank 1 constraint), but the statistics in the paper don’t constitute a reparameterization because there is no obvious map from these statistics to the original parameter space. One possibility would be something like “the gradient norm and various statistics of the residuals align on a curve independent of initialization” but there are many ways of rephrasing this that would work.

---

> > > ### Author Response · Authors · 2023-08-15
> > >
> > > We appreciate your careful reading of our responses and your valuable suggestion. We would also like to thank you for raising the review score.
> > >
> > > We acknowledge your point and agree that there is no obvious mapping from statistics of a reparameterization to the original parameter space. We will consider alternative expressions (e.g., canonical mapping) that more accurately capture the nature of the relationship and avoid any potential confusion in the revised manuscript.

---

### Author Rebuttal · Authors · 2023-08-09

Dear Reviewers, thank you for your valuable feedback and insightful comments. We have carefully considered all your questions, concerns, and suggestions, and we greatly appreciate the time and effort you have invested in reviewing our work.

We did our best to address all of your comments and will make the necessary revisions to improve the clarity, rigor, and validity of our manuscript. We kindly ask for your consideration in adjusting the review score if you find that our response has effectively addressed your concerns.

We are delighted to see that our novel characterization of the EoS phenomenon has been recognized by Reviewers Qm1U and ft7H. We extend our sincere gratitude to Reviewers Qm1U, ft7H, and mAcJ for their positive comments on our theoretical establishment of the trajectory alignment phenomenon.

In the revised manuscript, we will make the following major modifications.

- In abstract and main text, we initially claimed "When the EoS phenomenon occurs, different GD trajectories (after a proper reparameterization) align on a specific bifurcation diagram determined solely by the loss function, __independent of the network architecture, training data, and step size__." We will revise this claim to "When the EoS phenomenon occurs, different GD trajectories (after a proper reparameterization) align on a specific bifurcation diagram __independent of initialization__."

- We will replace __Figure 4__ (experiments on synthetic Gaussian dataset) in the main text by __Figure 13 in Appendix__ and __Figure 1 of the attached PDF file__, which demonstrates the trajectory alignment phenomenon of GD trained on a __real-world dataset (CIFAR-10) using FC networks and CNNs__.

Furthermore, we conducted additional experiments to address the reviewers' concerns and the results are presented in Figures 1-3 of the attached PDF file. Below, we provide a detailed description of each figure:

-  __Figure 1: Training CNN on CIFAR-10.__ To demonstrate the trajectory alignment phenomenon in more realistic settings, we trained CNN of width 500 and depth 3 on CIFAR-10, and observe the trajectory alignment phenomenon.

- __Figure 2: Assumptions on initialization in Theorem 4.2.__ To verify that the assumptions made in Theorem 4.2 covers wide range of cases, we considered log-cosh loss and investigated whether GD iterates $(p_t, q_t)$ with different initializations $(p_0, q_0)$ eventually fall into $S_A := [−1, 1] \times (\frac{3}{4} , 1)$ within $10^5$ iterations, under different step sizes ($\eta = 0.1, 0.01$). Note that once the iterate enters $S_A$, the assumption in Theorem 4.2 is satisfied and the theorem can be applied thereafter. Figure 2 shows that for small step sizes, GD iterates with initialization in the entire range of $q_0 \in (0, 1)$ eventually reach $S_A$ in most cases.

- __Figure 3: Sharpness value closely follows $\tilde{\lambda}(q_t)$ predicted by Theorem 4.4 for large width networks.__ The sharpness prediction $\tilde{\lambda}(q_t)$ for progressive sharpening in Theorem 4.4 was accurate for single-neuron case as shown in Figure 1(a) of the main text. We conducted additional experiments on two layer linear network with large widths (256 and 1024) and observe that our sharpness prediction is accurate regardless of the width.

We hope that these additional experiments further strengthen our contributions.

---

### Decision · Program_Chairs · 2023-09-21

**Decision:**

Accept (poster)

**Comment:**

The paper studies the edge of stability phenomenon in training of deep neural networks. The main claim is that different GD trajectories — after proper reparameterization — align on a specific bifurcation diagram independent of initialization.

**Weaknesses**

- One of the major limitations of the paper, as echoed by reviewers, is the experimental scope. The evidence presented mostly pertains to toy models and a singular data point, making the universal applicability of the findings questionable.
- Another concern lies in the strong claims made regarding the independence of the bifurcation diagram from multiple factors like network architecture and training data. These claims have not been sufficiently demonstrated across broader settings, and the claim was revised during discussions.
- Reviewers also raised questions about the generalizability of the reparameterization variables (p, q) and their utility in larger, more complex scenarios, beyond toy models.
- Most of the results and theory focuses on a perhaps too contrived case of training on a single data point. While the Authors have added additional results concerning training on more data points, it is still fully clear to what extend trajectory alignment also holds in this more realistic setting.

**Strengths**

- The concept of trajectory alignment is a solid and novel contribution that will potentially bring clarity to the field.
- The theoretical contributions such as using control theory or defining the canonical reparametrization were also appreciated. The trajectory alignment is proven rigorously. There is an argument to be made that the trajectory alignment is expected or plausible given results in prior work, however the paper makes concrete predictions (in the simple case) for the exact shape of the trajectory.

Although the paper has limitations, particularly concerning its experimental scope and the strong claims made, its theoretical contributions outweight the weaknesses. Based on this, it is my pleasure to recommend acceptance. I would like to ask the Authors to focus in the camera ready version on clarification of the main claim, and bringing in results on training involving a full dataset.